# Multivariate Distributional Reinforcement Learning Using Sliced Divergences

**Baptiste Debes** [1]    **Tinne Tuytelaars** [1]

## Abstract

Distributional reinforcement learning (DRL) models the full return distribution rather than expectations, but extending it to multivariate settings remains challenging. Many common metrics do not naturally generalize beyond one dimension or lose computational tractability, and the multivariate case introduces additional difficulties such as general matrix discounting, for which no contraction results are available. We introduce Sliced Distributional Reinforcement Learning (SDRL), which lifts tractable one-dimensional divergences to multivariate return distributions via projections. We prove Bellman contraction for uniform slicing under shared scalar discounting, and introduce a maximum-slicing variant with contraction under general dense discount matrices. SDRL supports a broad class of base divergences; we analyze Wasserstein, Cramér, and Maximum Mean Discrepancy (MMD), and characterize which SDRL variants suit the standard single-sample Bellman update used in distributional RL. We evaluate SDRL on a toy chain problem and a gridworld image-based environment as well as a subset of Atari games. Code is available at https://github.com/BaptisteDebes/SlicedDistributionalRL

## 1. Introduction

Distributional reinforcement learning (DRL) models full return distributions rather than expectations, with strong empirical (Dabney et al., 2018b;a; Barth-Maron et al., 2018; Hessel et al., 2018) and theoretical support (Lyle et al., 2019; Rowland et al., 2018; 2019a), building on the foundational perspective of Bellemare et al. (2017a; 2023b). In practice, DRL hinges on two coupled design choices: how to measure

---
[1]Department of Electrical Engineering (ESAT), KU Leuven, Leuven, Belgium. Correspondence to: Baptiste Debes <baptiste.debes@kuleuven.be>.

*Proceedings of the $43^{rd}$ International Conference on Machine Learning*, Seoul, South Korea. PMLR 306, 2026. Copyright 2026 by the author(s).

distributional discrepancy and the critic's parameterization (Rowland et al., 2019a). In the univariate case, several tractable combinations are available. For instance, the categorical parameterization paired with the KL divergence yields efficient updates (Bellemare et al., 2017a), while Wasserstein-based discrepancies admit efficient estimators when combined with quantile representations (Dabney et al., 2018a). This tractability is largely lost in the multivariate setting: categorical grids grow combinatorially, quantile parameterizations do not scale, and Wasserstein estimation becomes prohibitively costly, typically $\mathcal{O}(n^3 \log n)$ for general optimal transport solvers (Genevay et al., 2018).

A classical approach to high-dimensional distribution comparison is *slicing*, which represents multivariate distributions through their one-dimensional projections and aggregates discrepancies across directions. This idea underlies the framework of *Sliced Probability Divergences* (SPDs) (Rabin et al., 2011; Bonneel et al., 2015; Nadjahi et al., 2020), where distributions are projected onto directions on the unit sphere, one-dimensional discrepancies are computed, and the results are aggregated. This projection–aggregation mechanism reduces multivariate comparison to a collection of tractable univariate problems, enabling the use of base divergences with efficient one-dimensional estimators.

Building on this slicing principle, we introduce *Sliced Distributional Reinforcement Learning* (SDRL), a distributional reinforcement learning framework that leverages tractable one-dimensional projections to efficiently compare multivariate return distributions. Our approach lifts base divergences with efficient one-dimensional estimators, such as Wasserstein or Cramér, to the multivariate setting.

Crucially, moving beyond scalar-valued returns introduces additional degrees of freedom, notably through more general forms of discounting. While such forms of discounting naturally arise in a variety of settings, they have so far received limited theoretical treatment in distributional reinforcement learning. Beyond uniform slicing, we consider max slicing (Deshpande et al., 2019) as a tool to obtain contraction guarantees in settings beyond the standard scalar-discounted case. Max slicing replaces aggregation over random directions with an optimization over the most discriminative direction. One example is the problem of Distributional Sobolev Temporal Difference, for which a concrete instan-

tiation based on max slicing and MMD has been proposed (Debes & Tuytelaars, 2026). Here we develop the general framework underlying such constructions.

**Contributions.** We introduce **Sliced Distributional RL (SDRL)**, a general approach to multivariate return distributions based on sliced divergences, and establish Bellman contraction under shared scalar discounting. We further introduce a **Max-Sliced** variant (MSDRL) and establish contraction guarantees for *matrix-discounted* multivariate Bellman updates, a setting not covered by existing contraction analyses in distributional RL.

## 2. Background and Related Works

In the *expected* reinforcement learning framework, an agent interacts with an environment modeled as a Markov decision process (MDP) $(\mathcal{S}, \mathcal{A}, P, R, \{\Gamma(s,a)\})$, where rewards may be *d-dimensional* ($R_t \in \mathbb{R}^d$, $d \geq 1$). Here $\Gamma(s,a) \in \mathbb{R}^{d \times d}$ denotes a (possibly dense) state–action dependent discount–mixing matrix. Given a policy $\pi(a|s)$, the agent seeks to maximize the expected discounted return

$$Q^\pi(s,a) =$$
$$\mathbb{E}\left[\sum_{t=0}^\infty \Big(\prod_{k=1}^t \Gamma(S_k, A_k)\Big) R_t \,\Big|\, S_0 = s, \, A_0 = a\right] \quad (1)$$

with the convention $\prod_{k=1}^0 \Gamma(S_k, A_k) = I_d$. Classical RL methods focus on estimating $Q^\pi(s,a)$, the *expectation* of the return distribution (componentwise when $d > 1$).

The *distributional* perspective (Bellemare et al., 2017a), originally developed for scalar rewards with scalar discounting, can be applied here as well: it models the full return random variable

$$Z^\pi(s,a) \;=\; \sum_{t=0}^\infty \Big(\prod_{k=1}^t \Gamma(S_k, A_k)\Big) R_t, \quad (2)$$

whose expectation recovers $Q^\pi(s,a) = \mathbb{E}[Z^\pi(s,a)]$ (componentwise when $d > 1$). This viewpoint leads to the *distributional Bellman operator* with state–action dependent matrix discount. Let $S' \sim P(\cdot \mid s,a)$ and $A' \sim \pi(\cdot \mid S')$ be the next state and action; we adopt this sampling convention for all subsequent definitions of the Bellman operator. The operator is then defined as

$$(\mathcal{T}^\pi Z)(s,a) \stackrel{D}{=} R(s,a) + \Gamma(s,a)\, Z(S', A'), \quad (3)$$

where $\stackrel{D}{=}$ denotes equality in distribution.

**Special cases.**

1. **Classical distributional RL:** $d = 1$, $\Gamma(s,a) = \gamma \in [0,1)$ (Bellemare et al., 2017a).

2. **Multivariate with shared scalar discount:** $d > 1$, $\Gamma(s,a) = \gamma I_d$ (Zhang et al., 2021; Wiltzer et al., 2024a).

3. **General constant matrix:** $\Gamma(s,a) \equiv \Gamma$, where $\Gamma$ may be possibly dense; e.g., multi-horizon return modeling with $\Gamma = \mathrm{diag}(\gamma_1, \ldots, \gamma_d)$ that represent returns at different horizons via multiple discount factors (Fedus et al., 2019).

4. **State-action dependent general matrix:** $\Gamma(s,a)$ is a general (possibly dense) matrix (Debes & Tuytelaars, 2026).

### 2.1. Related works.

**Wasserstein.** Wasserstein metrics provide a *natural* divergence for distributional reinforcement learning, and in the scalar setting the distributional Bellman operator is contractive under $\mathbf{W}_p$ (Bellemare et al., 2017a). However, in high dimensions Wasserstein distances suffer from dimension-dependent statistical complexity (Fournier & Guillin, 2015). Motivated by these contraction properties, Freirich et al. (2019) reinterpret the distributional Bellman equation as an adversarial learning problem, using a learned discriminator to approximate $\mathbf{W}_1$. In practice, such discriminators can suffer from Lipschitz violations, finite-sample bias, and optimization error, yielding objectives that may deviate substantially from true optimal-transport distances (Mallasto et al., 2019; Stanczuk et al., 2021), thus weakening contraction claims that presume exact $\mathbf{W}_1$.

**MMD.** Moment matching with MMD was explored in the univariate case (Nguyen-Tang et al., 2021) and later extended to multivariate returns (Zhang et al., 2021). In the multivariate setting, contractivity results are available only for a narrow class of kernels (Wiltzer et al., 2024a), and identifying a kernel that is both empirically strong and contractive remains challenging (Killingberg & Langseth, 2023). Consequently, practitioners often resort to mixtures of Gaussian kernels despite their lack of contractivity guarantees in the multivariate setting.

*Synthesis.* Taken together, existing approaches suffer from at least one of three limitations: **(1)** performant methods may not be contractive, **(2)** theoretical guarantees do not extend to the general anisotropic discount setting we target, or **(3)** the estimation is too loose to support contraction claims (adversarial $\mathbf{W}_1$). This gap motivates our sliced approach.

## 3. SDRL: Distributional RL via Sliced Probability Divergences

### 3.1. Sliced Probability Divergences

**Slicing a base divergence.** Let $\Delta : \mathcal{P}(\mathbb{R}) \times \mathcal{P}(\mathbb{R}) \to \mathbb{R}_+ \cup \{\infty\}$ be a divergence on one–dimensional probability laws. Fix $p \geq 1$ and let $\sigma$ denote the uniform measure on

$\mathbb{S}^{d-1}$. For a unit direction $\theta \in \mathbb{S}^{d-1}$, let $P_\theta : \mathbb{R}^d \to \mathbb{R}$ denote the linear projection $P_\theta(x) = \langle \theta, x \rangle$, and write $(P_\theta)_{\#}\mu$ for the pushforward of $\mu$ by $P_\theta$. Given $\mu, \nu \in \mathcal{P}(\mathbb{R}^d)$, the associated sliced probability divergence (SPD) is

$$\mathbf{S}\Delta_p^p(\mu, \nu) \;=\; \int_{\mathbb{S}^{d-1}} \Delta^p\big((P_\theta)_{\#}\mu, (P_\theta)_{\#}\nu\big) \, d\sigma(\theta). \quad (4)$$

This lifts $\Delta$ to multivariate laws by averaging the one-dimensional divergence over random projection directions (Nadjahi et al., 2020).

**Monte Carlo approximation.** In practice, this integral is estimated via Monte Carlo sampling by drawing $L$ i.i.d. directions $\{\theta_i\}_{i=1}^L \sim \sigma$ and computing

$$\widehat{\mathbf{S}\Delta}_p^p(\mu, \nu) \;=\; \frac{1}{L} \sum_{i=1}^L \Delta^p\big((P_{\theta_i})_{\#}\mu, (P_{\theta_i})_{\#}\nu\big). \quad (5)$$

Each projected subproblem is independent, so the $L$ evaluations can be carried out in parallel.

**Sliced Wasserstein distance.** Among sliced probability divergences, the most widely used instance is the *sliced Wasserstein distance* (SWD) (Rabin et al., 2011; Bonneel et al., 2015), where the base divergence is chosen as $\Delta = \mathbf{W}_p$. For $\mu, \nu \in \mathcal{P}(\mathbb{R}^d)$ and $p \geq 1$,

$$\mathbf{SW}_p^p(\mu, \nu) \;=\; \int_{\mathbb{S}^{d-1}} \mathbf{W}_p^p\big((P_\theta)_{\#}\mu, (P_\theta)_{\#}\nu\big) \, d\sigma(\theta), \quad (6)$$

which reduces the high-dimensional Wasserstein problem to an average of one-dimensional Wasserstein distances between the projected pushforwards $(P_\theta)_{\#}\mu$ and $(P_\theta)_{\#}\nu$. For two sets of $n$ samples, computing each one-dimensional $\mathbf{W}_p$ reduces to sorting the projected samples, at cost $\mathcal{O}(n \log n)$ per direction; hence the Monte Carlo estimator with $L$ directions costs $\mathcal{O}(L n \log n)$ overall. This contrasts with computing exact OT in $\mathbb{R}^d$ between two sets of $n$ samples, which can scale as $\mathcal{O}(n^3 \log n)$ for standard exact solvers (Genevay et al., 2019). Further details on properties and the estimator are provided in Appendix A.1.

**Sliced Cramér distance.** In one dimension, a natural family of discrepancies is given by $\ell_p$ distances between cumulative distribution functions:

$$\ell_p^p(\mu, \nu) := \int_{\mathbb{R}} \big|F_\mu(u) - F_\nu(u)\big|^p \, du, \quad (7)$$

where $F_\mu$ and $F_\nu$ are the univariate CDFs of $\mu, \nu$. The special case $p = 2$ is the *Cramér distance* $\mathbf{C}_2 := \ell_2^2$ (Bellemare et al., 2017b). The *sliced Cramér* distance lifts this divergence to $\mathbb{R}^d$ via random projections:

$$\mathbf{SC}_2^2(\mu, \nu) \;=\; \int_{\mathbb{S}^{d-1}} \ell_2^2\big((P_\theta)_{\#}\mu, (P_\theta)_{\#}\nu\big) \, d\sigma(\theta). \quad (8)$$

This distance is also referred to as the *Cramér–Wold distance* and has been investigated in machine learning (Knop et al., 2020; Kolouri et al., 2020). Its estimator has the same complexity as sliced Wasserstein, since each one-dimensional Cramér evaluation costs $\mathcal{O}(n \log n)$. The use of the Cramér distance in distributional RL has been explored in prior work (Rowland et al., 2018; Lhéritier & Bondoux, 2021; Théate et al., 2023). See Appendix A.2 for further properties and our estimator.

**Sliced MMD.** Another tractable base divergence is the *Maximum Mean Discrepancy* (MMD) (Gretton et al., 2012), which has already been explored in distributional RL (Nguyen et al., 2020; Killingberg & Langseth, 2023; Wiltzer et al., 2024b). For laws $\mu, \nu \in \mathcal{P}(\mathbb{R}^d)$ and a kernel $k$, the squared MMD is

$$\begin{aligned}\mathbf{MMD}_k^2(\mu, \nu) = \mathbb{E}_{x,x'\sim\mu}[k(x,x')] + \mathbb{E}_{y,y'\sim\nu}[k(y,y')] \\ - 2\,\mathbb{E}_{x\sim\mu,\,y\sim\nu}[k(x,y)]. \quad (9)\end{aligned}$$

Lifting this discrepancy through random projections yields the *sliced MMD*: for $\mu, \nu \in \mathcal{P}(\mathbb{R}^d)$,

$$\begin{aligned}\mathbf{SMMD}_k^2(\mu, \nu) = \\ \int_{\mathbb{S}^{d-1}} \mathbf{MMD}_k^2\big((P_\theta)_{\#}\mu, (P_\theta)_{\#}\nu\big) \, d\sigma(\theta)\end{aligned} \quad (10)$$

Sliced MMD was first introduced in (Nadjahi et al., 2020). For two sets of $n$ samples, the base MMD estimator costs $\mathcal{O}(n^2)$ per projection, yielding an overall sliced estimator complexity of $\mathcal{O}(L n^2)$. More details on the properties of MMD and the estimator we use are provided in Appendix A.3.

### 3.2. Max Sliced Probability Divergences

Max slicing was proposed in (Deshpande et al., 2019) to address the potential inefficiency of uniform random slicing, where many directions may be required to capture the discrepancy between two distributions. It does so by learning the most discriminative projection direction, along which the projected one-dimensional divergence is largest. It is defined as

$$\mathbf{MS}\Delta(\mu, \nu) = \sup_{\theta \in \mathbb{S}^{d-1}} \Delta\big((P_\theta)_{\#}\mu, (P_\theta)_{\#}\nu\big). \quad (11)$$

This framework was originally proposed for $\Delta = \mathbf{W}_p$, yielding the *max–sliced Wasserstein distance* $\mathbf{MSW}_p$ (Deshpande et al., 2019). As discussed in Section 4, we show that max slicing can provide stronger contraction guarantees than uniform slicing.

**Estimation.** Since the supremum in the definition of max–sliced divergences cannot, in general, be computed exactly, it is typically approximated by iterative optimization of

the projection direction on the unit sphere. At each step a gradient ascent update on the divergence is followed by renormalization onto the unit sphere, and the final direction defines the empirical estimate. The full procedure is outlined in Algorithm 2.

### 3.3. Problem setting and algorithmic approach

We focus on *multivariate distributional policy evaluation*, estimating the full return law under a fixed policy $\pi$. We wish to model the *joint vector of multivariate returns* in order to capture their correlations and higher-order structure, rather than only marginal statistics. Let $d > 1$ and $\mathcal{X} = \mathbb{R}^d$. For any policy $\pi(\cdot \,|\, s)$, let $\mu^\pi(s, a) \in \mathcal{P}(\mathcal{X})$ denote the law of the multivariate return $Z^\pi(s, a)$. The distributional Bellman operator $\mathcal{T}^\pi$ relates return laws across state–action pairs via

$$
(\mathcal{T}^\pi \mu)(s, a) := \int_{\mathcal{S}} \int_{\mathcal{A}} \int_{\mathcal{X}} (f_{\Gamma(s,a),r})_{\#} \mu(s', a') \tag{12}
$$
$$
\times R(dr \,|\, s, a) \, \pi(da' \,|\, s') \, P(ds' \,|\, s, a).
$$

where $f_{\Gamma(s,a),r}(z) = r + \Gamma(s, a) z$ for $z \in \mathbb{R}^d$ and $\Gamma(s, a) \in \mathbb{R}^{d \times d}$ is a (possibly dense) discount–mixing matrix. The target in policy evaluation is the fixed point $\mu^\pi$ of $\mathcal{T}^\pi$, so $\mathcal{T}^\pi \mu^\pi = \mu^\pi$.

**Algorithmic approach** Following the particle-based critic paradigm (Nguyen-Tang et al., 2021; Zhang et al., 2021), we approximate $\mu^\pi(s, a)$ by finite sets of particles and optimize them directly. Given a transition $(s, a, r, s')$ and next action $a' \sim \pi(\cdot|s')$, we produce a set of $N$ predicted particles $\{z_i = Z_\phi(s, a, i)\}_{i=1}^N$ and a set of $N$ target particles $\{\hat{z}_j = r + \Gamma(s, a) Z_\phi(s', a', j)\}_{j=1}^N$. Their discrepancy is measured by a sliced probability divergence with base $\Delta$, using either $L$ random projections or a single optimized direction (max–sliced), and minimizing it w.r.t. $\phi$ yields a distributional TD update toward the matrix-discounted target (Algorithm 1).

## 4. Theoretical results

In this section, we provide theoretical foundations for multivariate distributional RL with sliced divergences, based on a *supremum divergence* and sufficient conditions for contraction of the distributional Bellman operator.

**Definition 1** (Supremum divergence). Let $\mathcal{D}$ be a divergence on $\mathcal{P}(\mathbb{R}^d)$ and let $\mu, \nu : \mathcal{S} \times \mathcal{A} \to \mathcal{P}(\mathbb{R}^d)$. The *supremum divergence* is defined as

$$
\overline{\mathcal{D}}(\mu, \nu) := \sup_{(s,a) \in \mathcal{S} \times \mathcal{A}} \mathcal{D}\big(\mu(s, a), \nu(s, a)\big). \tag{13}
$$

**Standing notation.** Throughout this section, returns are $\mathbb{R}^d$–valued and we write $\eta(s, a) \in \mathcal{P}(\mathbb{R}^d)$ for the return distribution at $(s, a)$. We write $\Delta$ for a base divergence on

---

**Algorithm 1** Distributional policy evaluation with sliced divergence

---

**Input:** number of particles $N$; base divergence $\Delta$ and order $p$; discount matrix $\Gamma$
**Input:** either projection count $L$ or a projection direction $\theta$
**Input:** sample transition $(s, a, r, s')$; policy $\pi$; model parameters $\phi$ (and target $\phi^-$)
$a' \sim \pi(\cdot \,|\, s')$
**for** $i = 1$ **to** $N$ **do**
    *Predicted particle*
    $z_i \leftarrow Z_\phi(s, a, i)$
    *Target particle*
    $\hat{z}_i \leftarrow r + \Gamma\, Z_{\phi^-}(s', a', i)$
**end for**
**Choose projection set** $\Theta$
**if** a direction $\theta$ is provided (max setting) **then**
    $\Theta \leftarrow \{\theta\}$
**else**
    draw $\{\theta_\ell\}_{\ell=1}^L \sim \mathrm{Unif}(\mathbb{S}^{d-1})$
    $\Theta \leftarrow \{\theta_\ell\}_{\ell=1}^L$
**end if**
**Monte Carlo estimator over projections**
*Using $P_{\theta'}(x) = \langle \theta', x \rangle$*
$S \leftarrow \frac{1}{|\Theta|} \sum_{\theta' \in \Theta} \Big[ \Delta\big(\{\langle \theta', z_i \rangle\}_{i=1}^N, \ \{\langle \theta', \hat{z}_i \rangle\}_{i=1}^N\big) \Big]^p$
**Output:** $S\Delta_p^p(\{z_i\}_{i=1}^N, \{\hat{z}_i\}_{i=1}^N)$

---

$\mathcal{P}(\mathbb{R})$, with lifted divergences $\mathbf{S}\Delta_p$ and $\mathbf{MS}\Delta$, and use $\overline{\mathcal{D}}$ for the supremum lift over $(s, a)$.

We focus on the following questions:

1. **Metric property:** When do $\overline{\mathbf{S}\Delta_p}$ and $\overline{\mathbf{MS}\Delta}$ induce metrics on $\mathcal{P}(\mathbb{R}^d)^{\mathcal{S} \times \mathcal{A}}$?

2. **Contraction property:** Under what conditions on $\Delta$ and on the discount structure $\Gamma$ does $\mathcal{T}^\pi$ contract in $\overline{\mathbf{S}\Delta_p}$ or $\overline{\mathbf{MS}\Delta}$?

3. **Sample complexity:** How does the estimation error of the sliced and max–sliced divergences scale with the number of samples, and do they avoid the curse of dimensionality?

4. **Stochastic training suitability:** With the widely used bootstrap update that instantiates the Bellman target from a single sampled successor $(s', a')$, when do $\mathbf{S}\Delta_p$ and $\mathbf{MS}\Delta$ remain compatible with this training setup?

### 4.1. Metric property

It is known that uniform slicing preserves the metric property of a base divergence (Nadjahi et al., 2020). Similarly,

Deshpande et al. (2019) established that max–sliced Wasserstein is a metric; more generally, max–slicing preserves metricity whenever the base divergence is a metric on $\mathcal{P}(\mathbb{R})$. Finally, supremum lifting over $(s,a)$ preserves metricity. These results are summarized in Theorem 1, with full proofs provided in Appendix B.

**Theorem 1.** *Assume $\Delta$ is a metric on $\mathcal{P}(\mathbb{R})$ and fix $p \in [1, \infty)$. Then: (i) $\mathbf{S}\Delta_p$ is a metric on $\mathcal{P}(\mathbb{R}^d)$; (ii) $\mathbf{MS}\Delta$ is a metric on $\mathcal{P}(\mathbb{R}^d)$; and (iii) the sup–lifts $\overline{\mathbf{S}\Delta_p}$ and $\overline{\mathbf{MS}\Delta}$ are metrics on $\mathcal{P}(\mathbb{R}^d)^{\mathcal{S} \times \mathcal{A}}$.*

### 4.2. Contraction property

Let $\mathcal{D}$ be any divergence on $\mathcal{P}(\mathbb{R}^d)$. We say that $\mathcal{T}$ is a $\kappa$–contraction in $\overline{\mathcal{D}}$ if there exists $\kappa \in [0,1)$ such that, for all $\eta_1, \eta_2 : \mathcal{S} \times \mathcal{A} \to \mathcal{P}(\mathbb{R}^d)$,

$$\overline{\mathcal{D}}(\mathcal{T}\eta_1, \mathcal{T}\eta_2) \leq \kappa \overline{\mathcal{D}}(\eta_1, \eta_2). \qquad (14)$$

**Univariate contraction**  We recall *sufficient* conditions under which the univariate distributional Bellman operator $\mathcal{T}^\pi$ is a $c(\gamma)$–contraction with respect to $\overline{\mathcal{D}}$ (Proposition 1). This slightly generalizes a result from (Bellemare et al., 2023a). The proof is in Appendix C.1.

**Proposition 1** (Univariate Bellman contraction). *Let $\Delta$ be a metric on $\mathcal{P}(\mathbb{R})$. For $t \in \mathbb{R}$, let $T_t(x) = x + t$ denote translation, and for $\gamma \in (0,1)$ let $S_\gamma(x) = \gamma x$ denote scaling. Suppose $\Delta$ satisfies:*

**(T)** Translation nonexpansion: $\Delta((T_t)_{\#}\mu, (T_t)_{\#}\nu) \leq \Delta(\mu, \nu)$ for all $t \in \mathbb{R}$.

**(S)** Scale contraction: *there exists a nondecreasing function $c : [0, \infty) \to [0, \infty)$ such that for every $s \in [0,1]$,*

$$\Delta((S_s)_{\#}\mu, (S_s)_{\#}\nu) \leq c(s)\, \Delta(\mu, \nu),$$

*with $c(s) \leq 1$ for all $s \in [0,1]$ and $c(s) < 1$ for all $s \in [0,1)$.*

**($\mathbf{M}_p$)** Mixture $p$–convexity: *for some $p \in [1, \infty)$, any probability measure $\rho$ and measurable families $(\mu_c), (\nu_c) \subset \mathcal{P}(\mathbb{R})$,*

$$\Delta\left(\int \mu_c \, d\rho, \int \nu_c \, d\rho\right) \leq \left(\int \Delta(\mu_c, \nu_c)^p \, d\rho\right)^{1/p}.$$

*Then the Bellman operator $\mathcal{T}^\pi$ is a $c(\gamma)$–contraction:*

$$\overline{\Delta}(\mathcal{T}^\pi\eta_1, \mathcal{T}^\pi\eta_2) \leq c(\gamma)\,\overline{\Delta}(\eta_1, \eta_2).$$

**Uniform slicing**  We now turn to the main multivariate contraction result. We start from the canonical setting of vector-valued returns in $\mathbb{R}^d$ with $d > 1$, where the Bellman update uses the shared scalar discount from Section 2. This is the

standard multivariate formulation considered in (Freirich et al., 2019; Zhang et al., 2021). Given $X' \sim \eta(S', A')$, the corresponding distributional Bellman update is

$$(\mathcal{T}^\pi\eta)(s,a) = \text{Law}(R(s,a) + \gamma I_d X'). \qquad (15)$$

The key observation is that the univariate sufficient conditions **(T)**, **(S)**, **($\mathbf{M}_p$)** from Proposition 1 lift directly to the sliced objective, so (15) contracts in the sup–sliced divergence with the same factor $c(\gamma)$ as in the univariate case. This is stated in Theorem 2; the full proof is in Appendix C.3.

**Theorem 2** (Bellman contraction under uniform slicing). *Fix $p \in [1, \infty)$. If a base divergence $\Delta$ satisfies **(T)**, **(S)** at some $\gamma \in (0,1)$ with $c(\gamma) < 1$, and **($\mathbf{M}_p$)**, then the Bellman operator $\mathcal{T}^\pi$ with isotropic discounting is a $c(\gamma)$–contraction with respect to the sup–sliced divergence:*

$$\overline{\mathbf{S}\Delta_p}(\mathcal{T}^\pi\eta_1, \mathcal{T}^\pi\eta_2) \leq c(\gamma)\,\overline{\mathbf{S}\Delta_p}(\eta_1, \eta_2).$$

**Max–slicing**  We now move beyond shared scalar discounting and consider anisotropic multivariate Bellman updates of the form

$$(\mathcal{T}^\pi\eta)(s,a) = \text{Law}(R(s,a) + \Gamma(s,a) X'), \qquad (16)$$

Our goal is the *norm–only* contraction form **(A)** for (16), for some nondecreasing function $c$:

**(A)**
$$\overline{\mathcal{D}}(\mathcal{T}^\pi\eta_1, \mathcal{T}^\pi\eta_2) \leq c(\bar{L})\,\overline{\mathcal{D}}(\eta_1, \eta_2),$$
$$\bar{L} := \sup_{(s,a) \in \mathcal{S} \times \mathcal{A}} \|\Gamma(s,a)\|_{\text{op}}.$$

This deliberately discards any dependence on the geometry of $\Gamma$ beyond its operator norm. Our motivation for doing so is twofold. First, this is the exact type of criterion obtained when using Wasserstein as divergence (Debes & Tuytelaars, 2026). Second, it yields a simple norm–based criterion that is already useful in practice (Debes & Tuytelaars, 2026)).

We may wonder whether **(A)** is already known to hold for multivariate distributional RL under divergences that both admit tractable estimators and come with multivariate contraction results. To the best of our knowledge, there are currently two such options: MMD with contraction–admitting kernels (Nguyen-Tang et al., 2021) and uniform sliced divergences. The next proposition shows that, for both, **(A)** can fail in general; full constructions are given in Appendix C.4.

**Proposition 2** (Non-contraction under matrix discounting). *Fix $\bar{L} \in (0,1)$. There exist $d > 1$ and a fixed matrix $\Gamma \in \mathbb{R}^{d \times d}$ with $\|\Gamma\|_{\text{op}} \leq \bar{L}$, and return distributions $\eta_1, \eta_2$ such that the operator induced by (16) fails to satisfy **(A)** for (a) MMD with kernels from (Nguyen et al., 2020; Killingberg & Langseth, 2023) and (b) uniform sliced divergences.*

We now turn to a sufficient condition under which **(A)** *does* hold. Theorem 3 shows that max–slicing yields **(A)** for (16) when the base divergence satisfies the univariate sufficient conditions of Proposition 1. The proof is given in Appendix C.5.

**Theorem 3** (Bellman contraction under max-slicing). *Assume $\Delta$ satisfies* **(T)**, **(S)**, *and* **(M$_p$)**. *Fix $d > 1$ and consider $\mathcal{T}^\pi$ associated with (16). Then $\mathcal{T}^\pi$ satisfies* **(A)** *with $\mathcal{D} = \overline{\mathbf{MS\Delta}}$.*

### 4.3. Sample complexity

We summarize sample–complexity bounds for uniform slicing and max–slicing; proofs are deferred to Appendix E.

**Uniform slicing.** As shown by Nadjahi et al. (2020) and restated in Proposition 3, uniform slicing inherits the one–dimensional rate of the base divergence, without additional dependence on the ambient dimension.

**Proposition 3** (Uniform slicing inherits one-dimensional rates). *Fix $p \in [1, \infty)$. Let $\Delta$ be a divergence on $\mathcal{P}(\mathbb{R})$ and assume there exists a function $\alpha(p, n) \geq 0$ such that for every $\mu \in \mathcal{P}(\mathbb{R})$ with empirical $\hat{\mu}_n$ we have $\mathbb{E}\big[\Delta(\hat{\mu}_n, \mu)^p\big] \leq \alpha(p, n)$. Then for any $\mu \in \mathcal{P}(\mathbb{R}^d)$ with empirical $\hat{\mu}_n$,*

$$\mathbb{E}\big|\mathbf{S\Delta}_p^p(\hat{\mu}_n, \mu)\big| \leq \alpha(p, n).$$

**Max–slicing.** For a class of base divergences, Proposition 4 gives a finite–sample bound for $\mathbf{MS\Delta}$ under a bounded–support assumption, which is natural in RL when returns are assumed bounded. Depending on the base divergence $\Delta$, the resulting rate scales like $(d \log n/n)^{\alpha/2}$ and therefore avoids the curse of dimensionality (polynomial dependence on $d$).

**Proposition 4** (Max-slicing sample complexity). *Assume $\mathrm{diam}(\mathrm{supp}\,\mu) \leq D$ and let $\hat{\mu}_n$ be the empirical law of $n$ i.i.d. samples from $\mu$. Let $\Delta$ be a divergence on $\mathcal{P}(\mathbb{R})$ and assume that for any one–dimensional laws $\mu_1, \nu_1$ supported on an interval of length $\leq D$ there exist $\alpha \in (0, 1]$, $\beta \geq 0$, and $L > 0$ such that $\Delta(\mu_1, \nu_1) \leq L D^\beta \|F_{\mu_1} - F_{\nu_1}\|_\infty^\alpha$. Then*

$$\mathbb{E}\,\mathbf{MS\Delta}(\hat{\mu}_n, \mu) = O\Big(D^\beta \big(\tfrac{d \log n}{n}\big)^{\alpha/2}\Big).$$

### 4.4. Suitability for stochastic training.

A general property that often matters for stochastic optimization is whether *sample-based* gradients match the gradient of the *population* objective. Let $\mu \in \mathcal{P}(\mathbb{R}^d)$ denote a data law and let $\nu_\phi \in \mathcal{P}(\mathbb{R}^d)$ be a parametric model. Writing $\hat{\mu}_m$ for the empirical measure of $m$ i.i.d. samples from $\mu$, we say that a divergence $\mathcal{D}$ satisfies the *unbiased sample gradient* property if

$$\mathbb{E}_{X_{1:m} \sim \mu}[\nabla_\phi \mathcal{D}(\hat{\mu}_m, \nu_\phi)] = \nabla_\phi \mathcal{D}(\mu, \nu_\phi). \qquad \text{(U)}$$

When **(U)** fails, stochastic gradients can be biased and, in some cases, the minimizers of the expected sample objective and the population objective may differ (Bellemare et al., 2017b).

**Why this matters in distributional RL.** In distributional RL, the Bellman target is the *mixture* law induced by transition and policy randomness, but a standard TD update typically instantiates that target from a *single* sampled successor $(S', A')$. As a result, training implicitly optimizes an *expected sample loss* rather than the population loss against the true mixture target. Wasserstein provides a canonical illustration where **(U)** fails (Proposition 5 in Bellemare et al. (2017a)), which has motivated alternatives such as quantile-regression based objectives (Dabney et al., 2018b; Singh et al., 2022). In contrast, the Cramér distance and $\mathrm{MMD}^2$ are well-known examples that do satisfy **(U)** (Bellemare et al., 2017b; Bińkowski et al., 2018). This motivates asking, for each divergence used in our framework, whether it remains compatible with one-sample Bellman bootstrapping in the sense of **(U)**.

**Uniform slicing.** As shown in Proposition 5, uniform slicing preserves **(U)** for the *powered* sliced objective; see Appendix D for the proof.

**Proposition 5** (Uniform slicing preserves **(U)**). *Fix $p \in [1, \infty)$. Assume that $\Delta^p$ satisfies* **(U)** *on $\mathcal{P}(\mathbb{R})$. Then the uniformly sliced powered objective $\mathbf{S\Delta}_p^p$ also satisfies* **(U)** *on $\mathcal{P}(\mathbb{R}^d)$: for any $\mu \in \mathcal{P}(\mathbb{R}^d)$, any model $\nu_\phi \in \mathcal{P}(\mathbb{R}^d)$, and any $m \geq 1$,*

$$\mathbb{E}_{X_{1:m} \sim \mu}\big[\nabla_\phi \mathbf{S\Delta}_p^p(\hat{\mu}_m, \nu_\phi)\big] = \nabla_\phi \mathbf{S\Delta}_p^p(\mu, \nu_\phi).$$

**Max-slicing.** Unlike uniform slicing, max-slicing introduces an additional maximization step over directions, and this changes the behavior of sample-based objectives: the direction selected by the max can depend on the same sample used to evaluate the loss. As a result, the unbiased sample gradient property **(U)** can fail for max-sliced objectives, even when it holds direction-wise. This means that max-slicing is generally not compatible with the usual one-sample Bellman bootstrapping setup used in distributional RL. See Appendix D.

**Proposition 6** (Max-slicing can violate **(U)**). *Fix $p \in [1, \infty)$. In general, the max-sliced objective $\mathbf{MS\Delta}$ does not satisfy* **(U)***: there exist $d > 1$, a data law $\mu \in \mathcal{P}(\mathbb{R}^d)$, a parametric family $(\nu_\phi)_\phi$, and $m \geq 1$ such that*

$$\mathbb{E}_{X_{1:m} \sim \mu}[\nabla_\phi \mathbf{MS\Delta}^p(\hat{\mu}_m, \nu_\phi)] \neq \nabla_\phi \mathbf{MS\Delta}^p(\mu, \nu_\phi).$$

### 4.5. Instantiations

Our results are agnostic to the choice of base divergence $\Delta$ on $\mathcal{P}(\mathbb{R})$, provided it is a metric and satisfies **(T)**, **(S)**, and **(M$_p$)**: the corresponding sliced and max–sliced objectives

inherit metricity and yield Bellman contraction via Theorems 2 and 3. In this paper we instantiate $\Delta$ with $\mathbf{W}_p$, the Cramér distance $\mathbf{C}_2$, and $\mathbf{MMD}_k$. For $\mathbf{MMD}$, the conclusions extend to any kernel $k$ for which $\mathbf{MMD}_k$ is a metric and admits a scale bound of the form **(S)**; we use the multiquadric kernel (Killingberg & Langseth, 2023), defined as $k_h(x, y) = -\sqrt{1 + h^2\|x - y\|^2}$, as a representative example, and use this kernel for all experiments reported in the subsequent section. A complete summary of instantiations (contraction factors and sample–complexity rates) is given in Appendix F, Table 1.

# 5. Experiments

**Chain environment.** We validate sliced divergences for *multivariate* policy evaluation on a tabular chain MDP adapted from Rowland et al. (2019a); Nguyen-Tang et al. (2021). The state space is a length-$K$ chain with stochastic transitions (Fig. 1). Rewards are positional and defined as $R_{t+1} = e_{S_{t+1}} \in \mathbb{R}^K$, the one-hot embedding of the entered next state. Appendix I provides full details.

We compare two bootstrapping regimes: (i) *standard one-sample distributional TD*, which instantiates the Bellman target from a single successor, and (ii) a *near-exact* variant that explicitly constructs the transition-mixture target using the known dynamics (Appendix I.4). This controlled comparison isolates the effect of the unbiased sample-gradient property **(U)**: in Fig. 2, near-exact TD is largely insensitive to gradient bias, whereas under one-sample TD the objectives that violate **(U)** degrade substantially (notably the Wasserstein-based and max-sliced variants). Performance is measured by the empirical $\widehat{\mathbf{W}}_2$ between Monte Carlo returns and the learned critic distribution.

**Pixel-based environments.** We next evaluate the same objectives on pixel observations. All pixel-based experiments use PQN (Gallici et al., 2024) as a common backbone. Its fully vectorized implementation and simplified training pipeline (notably, no target network) make it substantially faster, which lets us compare many objectives and variants under a fixed compute budget.

We consider two complementary benchmarks. **Maze** follows the maze environments of Zhang et al. (2021), which provide pixel observations and vector-valued rewards, and is used for *policy evaluation*. We assess distributional accuracy via the empirical $\widehat{\mathbf{W}}_2$ between Monte Carlo return samples and the return distribution predicted by the critic. We include max-sliced variants for completeness: unlike the chain environment, exact TD is not available here, making this a realistic setting where the gap between objectives satisfying **(U)** and those that do not is practically meaningful. Figure 4 shows the environment and a representative example where the critic closely matches the Monte Carlo return

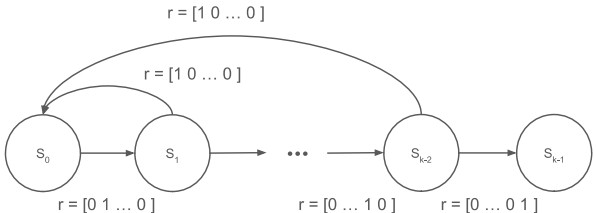

*Figure 1.* Chain policy-evaluation MDP: states $s_0 \to s_{K-1}$ (terminal). From any nonterminal $s_i$, action `fwd` moves to $s_{i+1}$ with probability 0.9 and resets to $s_0$ with probability 0.1; `bwd` swaps these probabilities. Positional-reward variant: $R_{t+1} = e_{S_{t+1}} \in \mathbb{R}^K$ (one-hot of the entered state).

distribution when trained with sliced Cramér. Figure 3 summarizes $\widehat{\mathbf{W}}_2$ across objectives and again highlights that **(U)** is essential for accurate distributional learning. In particular, sliced MMD and sliced Cramér perform on par with MMD.

**Atari** follows Zhang et al. (2021) and considers a subset of six Atari games from the Arcade Learning Environment (Bellemare et al., 2013) whose primitive rewards admit a fixed decomposition into multi-dimensional components. We evaluate *control* performance and report aggregated normalized scores across games and variants in Figure 5. Sliced Cramér performs strongly in this benchmark.

While objectives that satisfy **(U)** yield accurate distributional policy evaluation from pixels, this does not necessarily translate into the strongest control performance. In particular, sliced Wasserstein–2 performs remarkably well despite exhibiting biased gradients and poor distributional matching in earlier experiments. This suggests that control performance may depend primarily on accurate estimation of the return expectation even when the full return distribution is not well captured. This is consistent with the observation in Section J.2.2, where objectives violating **(U)** can still accurately predict the return mean despite collapsing the full distribution. It also aligns with prior work noting that policy improvement depends on the accuracy of the expected return (Rowland et al., 2019a).

The full training procedure and experimental setup for the pixel-based experiments are detailed in Appendix H. Further ablations and diagnostics are reported in Section J, including computational cost, qualitative return distribution visualizations, and an Atari sensitivity study on the number of projection directions (Section J.4).

# 6. Discussion and open question

For anisotropic updates of the form $R + \Gamma Z(S', A')$, Wasserstein is attractive because its contraction criterion depends only on $\|\Gamma\|_{\mathrm{op}}$. However, in our setting it faces three difficulties: (1) high computational cost in large $d$, (2) poor sta-

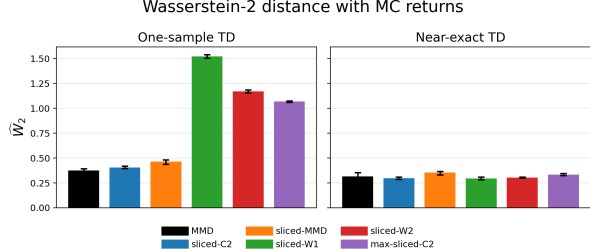

*Figure 2.* Chain policy-evaluation benchmark: empirical Wasserstein–2 distance $\widehat{\mathbf{W}}_2$ between Monte Carlo return samples and the return distribution predicted by the critic at the initial state. Left: standard one-sampled distributional TD. Right: near-exact distributional TD using an explicit mixture Bellman target. Bars show the median across 25 random seeds; error bars are 95% bootstrap confidence intervals (10k resamples).

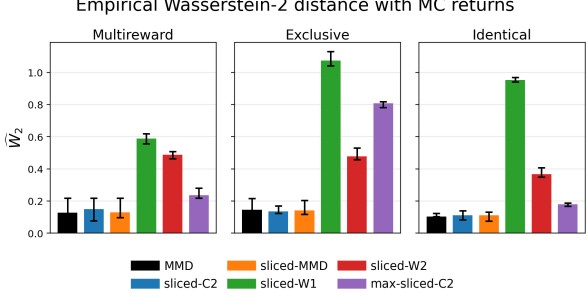

*Figure 3.* Maze policy-evaluation benchmark: empirical Wasserstein–2 distance $\widehat{\mathbf{W}}_2$ between Monte Carlo return samples and the return distribution predicted by the critic at the start state (uniform random evaluation policy). We report results for the three variants MAZE-MULTIREWARD, MAZE-EXCLUSIVE, and MAZE-IDENTICAL. Bars show the median across 5 random seeds; error bars are 95% bootstrap confidence intervals (10k resamples).

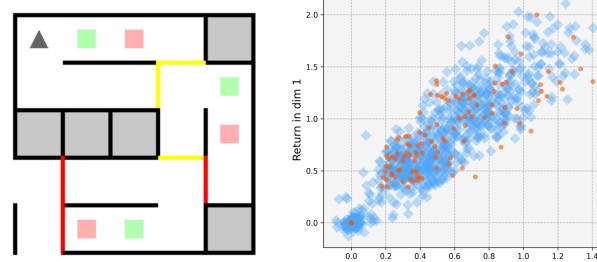

*Figure 4.* Left: initial observation of MAZE-IDENTICAL reproduced from Zhang et al. (2021). Right: Monte Carlo discounted returns (orange) versus critic-predicted return particles (blue) at the start state under the uniform random evaluation policy, for a distributional critic trained with sliced Cramér.

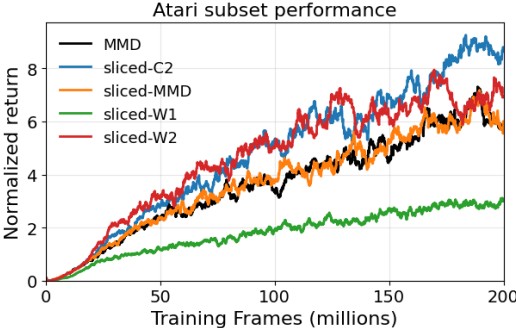

*Figure 5.* Aggregated control performance on an Atari subset with decomposed reward signals. Curves report normalized evaluation returns averaged across ASTEROIDS, GOPHER, MSPACMAN, PONG, and UPNDOWN, excluding AIRRAID, which is omitted from aggregation due to its atypical score scale. All methods are evaluated using the same protocol as in Gallici et al. (2024), following a greedy policy during evaluation and running evaluation environments continuously alongside training. Curves show the median across 5 random seeds.

tistical efficiency, and (3) incompatibility with the usual one-sample Bellman bootstrap because **(U)** fails. Max-slicing is designed to mitigate (1)–(2) by reducing comparisons to one-dimensional projections, but Proposition 6 shows that it can still fail (3) because of the max-selection step. Taken together, our results leave open the question: does there exist a divergence that is provably contractive under general anisotropic discounts in a norm-only sense and that also satisfies **(U)**, while still addressing (1)–(2) through tractable estimators and dimension-friendly statistical rates?

## 7. Conclusion

This work studies how to extend one-dimensional distributional objectives to *multivariate* return distributions in a way that remains compatible with temporal-difference learning. Our main practical takeaway is that **sliced Cramér** provides a strong default for multivariate distributional learning. It is computationally efficient, behaves well under one-sample bootstrapping, and comes with contraction guarantees in the standard isotropic setting.

On the methodological side, we propose a principled framework to lift divergences defined on one-dimensional samples to multivariate returns through slicing. This framework identifies conditions on the one-dimensional base divergence that are sufficient to obtain contraction of the corresponding multivariate distributional Bellman operator under anisotropic updates. We further analyze *max slicing* as an alternative to uniform slicing, and show how it can recover contraction under the same norm-controlled Bellman updates, while also requiring care due to its sample-dependent direction selection.

In particular, our results provide tools to nuance what can go wrong in practice when optimizing sliced objectives with bootstrapped targets. We show that an objective can remain contractive at the population level while still yielding biased sample gradients in the traditional one-sample TD setting,

and we characterize when this bias arises or disappears.

Empirically, our controlled policy-evaluation benchmarks confirm that satisfying the unbiased sample-gradient requirement is essential for accurate distribution matching from bootstrapped targets in the traditional one-sample regime, whereas the near-exact mixture updates largely remove these effects.

Finally, we are far from having explored the full potential of slicing, which has seen many improvements over the years (Kolouri et al., 2019; Rowland et al., 2019b; Nguyen-Tang et al., 2021; Chen et al., 2022). However, it is worth keeping in mind that improvements over uniform slicing can quickly reintroduce gradient bias, potentially making an objective unsuitable for one-sample TD and thus for traditional distributional RL.

## Acknowledgements

This project has received funding from the European Research Council (ERC) under the European Union's Horizon 2020 research and innovation programme (grant agreement n° 101021347).

## Impact Statement

This paper presents work whose goal is to advance the field of machine learning. There are many potential societal consequences of our work, none of which we feel must be specifically highlighted here.

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

## Appendix overview

This appendix collects the technical background, proofs, and experimental details supporting the main paper. **Base probability divergences** (Section A) recalls the definitions, estimators, and key structural properties of the standard discrepancies used as building blocks, namely the Wasserstein distance, the class of cumulative distribution function distances, and the Maximum Mean Discrepancy. **Metric property** (Section B) proves that lifting a base metric through uniform slicing or maximization over projections preserves the metric axioms on the space of multivariate probability measures. **Contraction results** (Section C) establishes sufficient conditions for the distributional Bellman operator to be a contraction, covering univariate and multivariate settings, uniform slicing, explicit counterexamples where norm control fails, and guarantees for max-sliced objectives. **Mixture bias** (Section D) discusses the suitability of different divergences for stochastic gradient-based training, focusing on the unbiased sample gradient property and its interaction with slicing operations. **Sample complexity** (Section E) derives statistical estimation rates, highlighting the dimension-free nature of uniform slicing and providing specific bounds for max-sliced Wasserstein and Cramer distances. **Instantiations** (Section F) summarizes the specific contraction factors, sample complexity bounds, and computational costs for the considered divergence families. **Pseudo-codes** (Section G) provides the algorithmic details for estimating the max-sliced divergence. **Pixel-based control: maze and Atari** (Section H) describes the experimental setup for the control tasks, including environment specifications and neural network architectures. **Chain environment** (Section I) details the tabular environment used for policy evaluation and the specific temporal difference algorithms employed. Finally, **More results and ablations** (Section J) presents additional experimental data and ablation studies.

## A. Base probability divergences

We briefly recall the base discrepancies used in our instantiations, along with the estimators and key structural properties that will be invoked later to obtain contraction factors and sample-complexity bounds. Section A.1 reviews Wasserstein and the contraction properties we use. Section A.2 covers the $\ell_p$ CDF distances on $\mathbb{R}$, including Cramér–2. Section A.3 reviews MMD and its kernel formulations.

### A.1. Wasserstein distance

The Wasserstein distance, arising from optimal transport theory (Villani et al., 2008), provides a principled way of comparing probability measures by quantifying the minimal cost of transporting mass from one distribution to another. Let $(\mathbb{R}^d, d)$ be a metric space and denote by $\mathcal{P}_p(\mathbb{R}^d)$ the set of Borel probability measures with finite $p$-th moment. For $\mu, \nu \in \mathcal{P}_p(\mathbb{R}^d)$, the $p$-Wasserstein distance is defined as

$$\mathbf{W}_p(\mu, \nu) = \left( \inf_{\pi \in \Pi(\mu,\nu)} \int_{\mathbb{R}^d \times \mathbb{R}^d} d(x, y)^p \, d\pi(x, y) \right)^{1/p}, \tag{17}$$

where $\Pi(\mu, \nu)$ denotes the set of couplings (or transport plans) $\pi$ whose marginals are $\mu$ and $\nu$. When the underlying measures admit densities $I_\mu$ and $I_\nu$, we may write $\mathbf{W}_p(I_\mu, I_\nu)$ without ambiguity.

Computing $\mathbf{W}_p$ directly is challenging in high dimensions, but there are settings where closed-form expressions exist. In the special case where $\mu$ and $\nu$ are one-dimensional distributions on a normed linear space, the Wasserstein distance simplifies to

$$\mathbf{W}_p(\mu, \nu) = \left( \int_0^1 \left| F_\mu^{-1}(z) - F_\nu^{-1}(z) \right|^p dz \right)^{1/p}, \tag{18}$$

where $F_\mu^{-1}$ and $F_\nu^{-1}$ are the quantile functions (inverse CDFs) of $\mu$ and $\nu$, respectively.

#### A.1.1. ESTIMATOR

For empirical measures $\tilde{\mu} = \frac{1}{N} \sum_{n=1}^{N} \delta_{x_n}$ and $\tilde{\nu} = \frac{1}{N} \sum_{n=1}^{N} \delta_{y_n}$ in one dimension, $\mathbf{W}_p$ can be computed by sorting the samples and comparing corresponding order statistics (Villani et al., 2008):

$$\mathbf{W}_p(\tilde{\mu}, \tilde{\nu}) = \left( \frac{1}{N} \sum_{n=1}^{N} \left| x_{I_x[n]} - y_{I_y[n]} \right|^p \right)^{1/p}, \tag{19}$$

where $I_x[n]$ and $I_y[n]$ are the indices that sort $\{x_n\}$ and $\{y_n\}$ in ascending order.

A.1.2. PROPERTIES

**Metric.** It is a classical result that the Wasserstein distances are genuine metrics. In particular, Proposition 2 in (Givens & Shortt, 1984) establishes that

$$\mathbf{W}_p \quad \text{is a metric on } \mathcal{P}_p(\mathbb{R}) \qquad \text{for every } p \in [1, \infty],$$

where $\mathcal{P}_p(\mathbb{R}) = \{\mu \in \mathcal{P}(\mathbb{R}) : \int |x|^p \, d\mu(x) < \infty\}$ for $p < \infty$, and $\mathcal{P}_\infty(\mathbb{R}) = \mathcal{P}(\mathbb{R})$.

**Translation invariant.** By definition

**Scale contraction.**

**Proposition A.1** (Linear push-forward contraction for $\mathbf{W}_p$ under matrix discount). *Let $p \in [1, \infty]$ and $d \geq 1$. Equip $\mathbb{R}^d$ with the Euclidean metric $d(x, y) = \|x - y\|_2$. If $p < \infty$, assume $\mu, \nu \in \mathcal{P}_p(\mathbb{R}^d)$; if $p = \infty$, assume $\mathbf{W}_\infty(\mu, \nu) < \infty$. Let $\Gamma \in \mathbb{R}^{d \times d}$ and define the linear map $F_\Gamma : \mathbb{R}^d \to \mathbb{R}^d$ by*

$$F_\Gamma(x) = \Gamma x.$$

*Then*

$$\boxed{\mathbf{W}_p\big((F_\Gamma)_{\#}\mu, \, (F_\Gamma)_{\#}\nu\big) \; \leq \; \|\Gamma\|_{\mathrm{op}} \, \mathbf{W}_p(\mu, \nu).}$$

*In particular, when $\Gamma = \gamma I_d$ this recovers Lemma 3 of (Zhang et al., 2021).*

*Moreover, in the univariate case $d = 1$ and $\Gamma = \gamma \in [0, 1]$, letting $S_\gamma(x) = \gamma x$ yields the scale-contraction form*

$$\mathbf{W}_p\big((S_\gamma)_{\#}\mu, \, (S_\gamma)_{\#}\nu\big) \leq \gamma \, \mathbf{W}_p(\mu, \nu).$$

*Proof.* Fix any $\varepsilon > 0$, and choose a coupling $\pi \in \Pi(\mu, \nu)$ that is $\varepsilon$-optimal:

$$\left(\int_{\mathbb{R}^d \times \mathbb{R}^d} \|x - y\|_2^p \, d\pi(x, y)\right)^{1/p} < \mathbf{W}_p(\mu, \nu) + \varepsilon \qquad (1 \leq p < \infty),$$

and for $p = \infty$ choose $\pi$ with

$$\sup_{(x,y) \in \mathrm{supp}(\pi)} \|x - y\|_2 < \mathbf{W}_\infty(\mu, \nu) + \varepsilon.$$

Push this coupling forward under $F_\Gamma \times F_\Gamma$ to obtain

$$\pi' = (F_\Gamma \times F_\Gamma)_{\#}\pi \; \in \; \Pi\big((F_\Gamma)_{\#}\mu, \, (F_\Gamma)_{\#}\nu\big).$$

**Case** $1 \leq p < \infty$**.** By definition of $\mathbf{W}_p$,

$$\begin{aligned}
\mathbf{W}_p\big((F_\Gamma)_{\#}\mu, \, (F_\Gamma)_{\#}\nu\big)^p &\leq \int \|u - v\|_2^p \, d\pi'(u, v) \\
&= \int \|\Gamma x - \Gamma y\|_2^p \, d\pi(x, y) \\
&\leq \|\Gamma\|_{\mathrm{op}}^p \int \|x - y\|_2^p \, d\pi(x, y) \\
&< \|\Gamma\|_{\mathrm{op}}^p \big(\mathbf{W}_p(\mu, \nu) + \varepsilon\big)^p,
\end{aligned}$$

where we used $\|\Gamma(x - y)\|_2 \leq \|\Gamma\|_{\mathrm{op}} \|x - y\|_2$. Taking $p$th roots and letting $\varepsilon \to 0$ gives

$$\mathbf{W}_p\big((F_\Gamma)_{\#}\mu, \, (F_\Gamma)_{\#}\nu\big) \leq \|\Gamma\|_{\mathrm{op}} \, \mathbf{W}_p(\mu, \nu).$$

**Case $p = \infty$.** Using the same bijection of couplings,

$$
\begin{aligned}
\mathbf{W}_\infty\big((F_\Gamma)_{\#}\mu, (F_\Gamma)_{\#}\nu\big) &\leq \sup_{(u,v)\in\mathrm{supp}(\pi')} \|u - v\|_2 \\
&= \sup_{(x,y)\in\mathrm{supp}(\pi)} \|\Gamma x - \Gamma y\|_2 \\
&\leq \|\Gamma\|_{\mathrm{op}} \sup_{(x,y)\in\mathrm{supp}(\pi)} \|x - y\|_2 \\
&< \|\Gamma\|_{\mathrm{op}}\big(\mathbf{W}_\infty(\mu,\nu) + \varepsilon\big).
\end{aligned}
$$

Letting $\varepsilon \to 0$ yields the desired bound.

Finally, when $d = 1$ and $\Gamma = \gamma \geq 0$, the map is $S_\gamma(x) = \gamma x$ and the above gives $\mathbf{W}_p((S_\gamma)_{\#}\mu, (S_\gamma)_{\#}\nu) \leq \gamma \mathbf{W}_p(\mu,\nu)$. □

**p-convexity.**

**Proposition A.2** (Mixture $p$-convexity for $\mathbf{W}_p$). *Let $(X, d)$ be a metric space, $p \in [1, \infty)$, and let $(\Omega, \mathcal{F}, \rho)$ be a probability space. Let $(\mu_c)_{c\in\Omega}, (\nu_c)_{c\in\Omega} \subset \mathcal{P}_p(X)$ be measurable families. Then*

$$
\mathbf{W}_p\Big(\int_\Omega \mu_c\, \rho(dc),\ \int_\Omega \nu_c\, \rho(dc)\Big) \ \leq\ \left(\int_\Omega \mathbf{W}_p(\mu_c, \nu_c)^p\, \rho(dc)\right)^{1/p}.
$$

*Proof.* **Step 1: $\varepsilon$-optimal couplings for each $c$.**
Fix $\varepsilon > 0$. For each $c \in \Omega$, pick an $\varepsilon$-optimal coupling $\pi_c^\varepsilon \in \Pi(\mu_c, \nu_c)$ such that

$$
\int_{X\times X} d(x, y)^p\, \pi_c^\varepsilon(dx, dy) \ \leq\ \mathbf{W}_p(\mu_c, \nu_c)^p + \varepsilon.
$$

**Step 2: Measurable selection and mixed coupling.**
Assume the family $(\pi_c^\varepsilon)_{c\in\Omega}$ can be chosen measurably, so that $c \mapsto \pi_c^\varepsilon$ is a probability kernel. We then define the mixed coupling

$$
\Pi^\varepsilon(U) \ :=\ \int_\Omega \pi_c^\varepsilon(U)\, \rho(dc), \qquad U \subseteq X \times X \text{ Borel}.
$$

For any measurable $A \subseteq X$,

$$
\Pi^\varepsilon(A \times X) = \int_\Omega \pi_c^\varepsilon(A \times X)\, \rho(dc) = \int_\Omega \mu_c(A)\, \rho(dc) = \Big(\int_\Omega \mu_c\, \rho(dc)\Big)(A),
$$

and similarly

$$
\Pi^\varepsilon(X \times A) = \int_\Omega \pi_c^\varepsilon(X \times A)\, \rho(dc) = \int_\Omega \nu_c(A)\, \rho(dc) = \Big(\int_\Omega \nu_c\, \rho(dc)\Big)(A).
$$

Hence $\Pi^\varepsilon$ has the mixed marginals $\int_\Omega \mu_c\, \rho(dc)$ and $\int_\Omega \nu_c\, \rho(dc)$, i.e.

$$
\Pi^\varepsilon \ \in\ \Pi\Big(\int_\Omega \mu_c\, \rho(dc),\ \int_\Omega \nu_c\, \rho(dc)\Big).
$$

**Step 3: Bound the transport cost of the mixed coupling.**
Since $(c, x, y) \mapsto d(x, y)^p$ is nonnegative and measurable and $c \mapsto \pi_c^\varepsilon$ is a probability kernel, Tonelli's theorem allows us to exchange the order of integration in $(c, x, y)$:

$$
\begin{aligned}
\int_{X\times X} d(x, y)^p\, \Pi^\varepsilon(dx, dy) &= \int_{X\times X} d(x, y)^p \left(\int_\Omega \pi_c^\varepsilon(dx, dy)\, \rho(dc)\right) \\
&= \int_\Omega \left(\int_{X\times X} d(x, y)^p\, \pi_c^\varepsilon(dx, dy)\right) \rho(dc) \\
&\leq \int_\Omega \big(\mathbf{W}_p(\mu_c, \nu_c)^p + \varepsilon\big)\, \rho(dc) \\
&= \int_\Omega \mathbf{W}_p(\mu_c, \nu_c)^p\, \rho(dc)\ +\ \varepsilon.
\end{aligned}
$$

**Step 4: Take the infimum over couplings and pass to the limit.**
By definition of $\mathbf{W}_p$,

$$\mathbf{W}_p\Big(\int \mu_c \, d\rho, \int \nu_c \, d\rho\Big)^p \;\leq\; \int_{X \times X} d(x,y)^p \, \Pi^\varepsilon(dx, dy) \;\leq\; \int_\Omega \mathbf{W}_p(\mu_c, \nu_c)^p \, \rho(dc) + \varepsilon.$$

Taking $p$th roots and letting $\varepsilon \downarrow 0$ yields

$$\mathbf{W}_p\Big(\int_\Omega \mu_c \, \rho(dc), \int_\Omega \nu_c \, \rho(dc)\Big) \;\leq\; \left(\int_\Omega \mathbf{W}_p(\mu_c, \nu_c)^p \, \rho(dc)\right)^{1/p}.$$

$\square$

## A.2. The $\ell_p$–family of CDF distances on $\mathbb{R}$

Let $\mu, \nu \in \mathcal{P}(\mathbb{R})$ be probability measures with cumulative distribution functions (CDFs) $F_\mu, F_\nu$. For $p \in [1, \infty)$, the $\ell_p$ distance between $\mu$ and $\nu$ is defined as

$$\ell_p(\mu, \nu) := \left(\int_{-\infty}^{\infty} \big|F_\mu(t) - F_\nu(t)\big|^p \, dt\right)^{1/p} \;=\; \|F_\mu - F_\nu\|_{L^p(\mathbb{R})},$$

that is, the $\ell_p$–family can be seen as the $L^p$ norm between the two CDFs (Bellemare et al., 2017b).

**Connections to other distances.**

- **Wasserstein distance.** For $p = 1$, one recovers the 1–Wasserstein distance (Bellemare et al., 2017b):

$$\ell_1(\mu, \nu) = \mathbf{W}_1(\mu, \nu) = \int_0^1 \big|F_\mu^{-1}(u) - F_\nu^{-1}(u)\big| \, du.$$

  Thus, $\ell_1$ coincides with the classical earth mover's distance on $\mathbb{R}$.

- **Cramér distance.** For $p = 2$, the squared $\ell_2$ distance coincides with the Cramér distance (also denoted $\mathbf{C}_2$) (Bellemare et al., 2017b):

$$\ell_2^2(\mu, \nu) = \mathbf{C}_2(\mu, \nu) = \int_{-\infty}^{\infty} \big(F_\mu(t) - F_\nu(t)\big)^2 \, dt,$$

  which also admits the energy distance form

$$\ell_2^2(\mu, \nu) = \mathbb{E}|X - Y| - \tfrac{1}{2}\mathbb{E}|X - X'| - \tfrac{1}{2}\mathbb{E}|Y - Y'|,$$

  for $X, X' \sim \mu$ and $Y, Y' \sim \nu$ i.i.d.

- **Cramér distance and the Continuous Ranked Probability Score (CRPS).** The squared $\ell_2$ distance admits a close connection to the *Continuous Ranked Probability Score (CRPS)* (Gneiting & Raftery, 2007). For a predictive distribution $\mu$ with CDF $F_\mu$ and an observation $y \in \mathbb{R}$, the CRPS is defined as

$$\mathrm{CRPS}(\mu, y) := \int_{-\infty}^{\infty} \big(F_\mu(t) - \mathbf{1}\{y \leq t\}\big)^2 \, dt.$$

Taking expectation with respect to $Y \sim \nu$ and exchanging expectation and integration yields

$$\mathbb{E}_{Y \sim \nu} \, \mathrm{CRPS}(\mu, Y) = \int_{-\infty}^{\infty} \Big(F_\mu(t)^2 - 2F_\mu(t)F_\nu(t) + F_\nu(t)\Big) \, dt.$$

Adding and subtracting $F_\nu(t)^2$ inside the integral gives the decomposition

$$\mathbb{E}_{Y \sim \nu} \, \mathrm{CRPS}(\mu, Y) = \underbrace{\int_{-\infty}^{\infty} \big(F_\mu(t) - F_\nu(t)\big)^2 \, dt}_{\ell_2^2(\mu, \nu)} + \underbrace{\int_{-\infty}^{\infty} \big(F_\nu(t) - F_\nu(t)^2\big) \, dt}_{\text{constant in } \mu}.$$

Consequently, minimizing the expected CRPS with respect to $\mu$ is equivalent to minimizing the squared Cramér distance $\ell_2^2(\mu, \nu)$, and both objectives induce identical gradients with respect to the predicted distribution, since the second term is independent of $\mu$.

A.2.1. ESTIMATOR

**Empirical CDF estimator.** For empirical measures $\tilde{\mu} = \frac{1}{n} \sum_{i=1}^{n} \delta_{u_i}$ and $\tilde{\nu} = \frac{1}{m} \sum_{j=1}^{m} \delta_{v_j}$ in one dimension, the $\ell_p$ CDF distance (Cramér when $p = 2$) admits a closed form: after merging and sorting all samples, one tracks the cumulative difference of the two empirical CDFs, which is piecewise constant between successive breakpoints. The distance then reduces to a weighted sum of gap lengths multiplied by the corresponding powers of this difference.

$$\ell_p^p(\tilde{\mu}, \tilde{\nu}) = \sum_{k=1}^{K-1} (t_{k+1} - t_k) \, |\Delta_k|^p.$$

Algorithmically, the estimator amounts to sorting the combined samples once, tracking the cumulative difference of the two empirical CDFs, and summing the piecewise contributions. This requires $\mathcal{O}((n+m)\log(n+m))$ time for sorting and linear time for the scan.

**CRPS–based estimator for the $\ell_2$ (Cramér–2) distance.** Building on the connection established in Section A.2 between the squared $\ell_2$ CDF distance (Cramér–2) and the expected *Continuous Ranked Probability Score (CRPS)*, we construct an empirical estimator using the integral representation of the CRPS. Let $F_\mu$ denote the CDF of $\mu$ and $Q_\mu(\tau) = F_\mu^{-1}(\tau)$ its quantile function. The CRPS satisfies

$$\mathrm{CRPS}(\mu, y) = 2 \int_0^1 \rho_\tau\big(y - Q_\mu(\tau)\big) \, d\tau, \qquad \rho_\tau(r) = r\big(\tau - \mathbf{1}\{r < 0\}\big).$$

In practice, we approximate the integral over $\tau$ by a finite sum over uniformly spaced quantile levels $\tau_i = (2i-1)/(2n)$ and replace $Q_\mu(\tau_i)$ by the order statistics $u_{(i)}$, yielding the empirical objective

$$\widehat{\mathcal{L}}_{\mathrm{CRPS}} = \frac{2}{nm} \sum_{i=1}^{n} \sum_{j=1}^{m} \rho_{\tau_i}\big(v_j - u_{(i)}\big).$$

As shown in Section A.2, the expected CRPS differs from the squared Cramér–2 distance $\ell_2^2(\mu, \nu)$ by an additive constant independent of $\mu$. Accordingly, we use the gradient of the estimator above as a surrogate for the gradient of the true Cramér–2 objective, namely

$$\nabla_\theta \widehat{\mathcal{L}}_{\mathrm{CRPS}} \approx \nabla_\theta \, \ell_2^2(\mu_\theta, \nu),$$

where the approximation reflects the finite-sample and discretization effects of the estimator. Computationally, this estimator requires sorting the $n$ samples from $\mu$ once ($\mathcal{O}(n \log n)$) and evaluating $nm$ terms, resulting in total cost $\mathcal{O}(n \log n + nm)$.

**Energy distance estimator.** A second estimator for the squared $\ell_2$ CDF distance (Cramér–2) follows from its energy distance representation. For $X, X' \sim \mu$ and $Y, Y' \sim \nu$ i.i.d., one has

$$\ell_2^2(\mu, \nu) = \mathbb{E}|X - Y| - \tfrac{1}{2}\mathbb{E}|X - X'| - \tfrac{1}{2}\mathbb{E}|Y - Y'|.$$

Given samples $u_1, \ldots, u_n \sim \mu$ and $v_1, \ldots, v_m \sim \nu$, each expectation is estimated by its corresponding U–statistic, yielding

$$\widehat{\ell}_2^2(\mu, \nu) = \frac{1}{nm} \sum_{i,j} |u_i - v_j| - \frac{1}{2n(n-1)} \sum_{i \neq i'} |u_i - u_{i'}| - \frac{1}{2m(m-1)} \sum_{j \neq j'} |v_j - v_{j'}|.$$

This estimator is unbiased for $\ell_2^2(\mu, \nu)$ and does not rely on sorting or discretization.

Equivalently, the squared Cramér–2 distance coincides with the squared Maximum Mean Discrepancy $\mathrm{MMD}_k^2(\mu, \nu)$ associated with the kernel

$$k(x, y) = -|x - y|.$$

Under this identification, the estimator above coincides (up to a constant factor) with the unbiased MMD U–statistic associated with the kernel $k$, as discussed in Section A.3. Computationally, evaluating the estimator involves $\mathcal{O}(nm)$ cross–sample pairwise terms together with $\mathcal{O}(n^2 + m^2)$ within–sample terms, resulting in an overall cost of $\mathcal{O}((n + m)^2)$.

A.2.2. PROPERTIES

**Metric.**

**Proposition A.3** (Metric property of the $\ell_p$ CDF distance). *Let $\mu, \nu \in \mathcal{P}(\mathbb{R})$ have CDFs $F_\mu, F_\nu$. For $p \in [1, \infty)$ define*

$$\ell_p(\mu, \nu) := \|F_\mu - F_\nu\|_{L^p(\mathbb{R})} = \left( \int_\mathbb{R} |F_\mu(t) - F_\nu(t)|^p \, dt \right)^{1/p}.$$

*Let $\mathcal{P}_1(\mathbb{R}) := \{\xi \in \mathcal{P}(\mathbb{R}) : \int_\mathbb{R} |x| \, d\xi(x) < \infty\}$. Then for every $p \in [1, \infty)$, $\ell_p$ is a metric on $\mathcal{P}_1(\mathbb{R})$.*

*Proof.* **Finiteness on $\mathcal{P}_1(\mathbb{R})$.**
In one dimension, $\ell_1(\mu, \nu) = \int_\mathbb{R} |F_\mu - F_\nu| \, dt = W_1(\mu, \nu)$, hence $\ell_1(\mu, \nu) < \infty$ for $\mu, \nu \in \mathcal{P}_1(\mathbb{R})$. For $p > 1$, since $0 \le |F_\mu - F_\nu| \le 1$,

$$\ell_p(\mu, \nu)^p = \int |F_\mu - F_\nu|^p \, dt \le \int |F_\mu - F_\nu| \, dt = \ell_1(\mu, \nu) < \infty.$$

**Nonnegativity and symmetry.**
By definition, $\ell_p(\mu, \nu) = \|F_\mu - F_\nu\|_{L^p} \ge 0$ and $\ell_p(\mu, \nu) = \|F_\mu - F_\nu\|_{L^p} = \|F_\nu - F_\mu\|_{L^p} = \ell_p(\nu, \mu)$.

**Identity of indiscernibles.**
If $\ell_p(\mu, \nu) = 0$, that means

$$\int |F_\mu(x) - F_\nu(x)|^p \, dx = 0.$$

An $L^p$ norm is zero iff the functions are equal almost everywhere. So $F_\mu = F_\nu$ except maybe on a measure zero set. Now, CDFs are monotone and right–continuous. Such functions cannot differ only on a measure zero set, if they are different at one point, they must differ on an interval of positive length. So "equal almost everywhere" forces them to be equal everywhere. If the CDFs are identical, then the distributions are the same.

**Triangle inequality.**
We use Minkowski's inequality in $L^p(\mathbb{R})$. Writing $F_\mu - F_\lambda = (F_\mu - F_\nu) + (F_\nu - F_\lambda)$, we obtain

$$\begin{aligned} \ell_p(\mu, \lambda) &= \|F_\mu - F_\lambda\|_{L^p} \\ &= \|(F_\mu - F_\nu) + (F_\nu - F_\lambda)\|_{L^p} \\ &\le \|F_\mu - F_\nu\|_{L^p} + \|F_\nu - F_\lambda\|_{L^p}. \quad \text{(Minkowski)} \end{aligned}$$

Therefore $\ell_p(\mu, \lambda) \le \ell_p(\mu, \nu) + \ell_p(\nu, \lambda)$. $\qquad\square$

**Translation invariant.** By non-trivial arguments (see Theorem 2 in (Bellemare et al., 2017b) and Proposition 3.2 in (Odin & Charpentier, 2020)), the $\ell_p$ distance is invariant under translations: for all $\mu, \nu \in \mathcal{P}_1(\mathbb{R})$ and every $t \in \mathbb{R}$,

$$\ell_p\big((T_t)_\# \mu, (T_t)_\# \nu\big) = \ell_p(\mu, \nu), \qquad T_t(x) = x + t.$$

**Scaling.**

**Proposition A.4** (($1/p$)-homogeneity of the $\ell_p$ CDF distance under scaling). *Let $\mu, \nu \in \mathcal{P}(\mathbb{R})$ have CDFs $F_\mu, F_\nu$, and assume $\ell_p(\mu, \nu) < \infty$. For $p \in [1, \infty)$ and $S_s(x) = s\,x$ with $s > 0$, the $\ell_p$ distance satisfies*

$$\ell_p\big((S_s)_\# \mu, (S_s)_\# \nu\big) = s^{1/p} \ell_p(\mu, \nu) \le c(s) \ell_p(\mu, \nu), \qquad c(s) := s^{1/p}.$$

*Proof.* **Scale–sensitivity via change of variables.**

Let $s > 0$. Using $F_{(S_s)_\# \mu}(x) = F_\mu(x/s)$, we compute

$$\ell_p\big((S_s)_\# \mu, (S_s)_\# \nu\big) = \left( \int_\mathbb{R} \big|F_\mu(x/s) - F_\nu(x/s)\big|^p \, dx \right)^{1/p}$$

$$= \left( \int_\mathbb{R} \big|F_\mu(u) - F_\nu(u)\big|^p \, s \, du \right)^{1/p} \quad \text{(C.V. } u = x/s)$$

$$= s^{1/p} \left( \int_\mathbb{R} \big|F_\mu(u) - F_\nu(u)\big|^p \, du \right)^{1/p}$$

$$= s^{1/p} \, \ell_p(\mu, \nu).$$

Thus, for $s > 0$,

$$\ell_p\big((S_s)_\# \mu, (S_s)_\# \nu\big) = s^{1/p} \, \ell_p(\mu, \nu),$$

which shows that $\ell_p$ is positively homogeneous of degree $1/p$ under scaling. $\square$

**p-convexity.**

**Proposition A.5** (Mixture $p$-convexity for $\ell_p$ (integral form))**.** *Let $(\Omega, \mathcal{F}, \rho)$ be a probability space, $p \in [1, \infty)$, and let $(\mu_c)_{c \in \Omega}, (\nu_c)_{c \in \Omega} \subset \mathcal{P}_1(\mathbb{R})$ be measurable families with CDFs $(F_{\mu_c}), (F_{\nu_c})$. Then*

$$\ell_p\Big( \int_\Omega \mu_c \, \rho(dc), \int_\Omega \nu_c \, \rho(dc) \Big) \leq \left( \int_\Omega \ell_p(\mu_c, \nu_c)^p \, \rho(dc) \right)^{1/p}.$$

*Proof.* **CDF linearity under mixtures.**
For every $x \in \mathbb{R}$,

$$F_{\int \mu_c \, d\rho}(x) = \int_\Omega F_{\mu_c}(x) \, \rho(dc), \qquad F_{\int \nu_c \, d\rho}(x) = \int_\Omega F_{\nu_c}(x) \, \rho(dc).$$

Hence

$$F_{\int \mu_c \, d\rho}(x) - F_{\int \nu_c \, d\rho}(x) = \int_\Omega \big( F_{\mu_c}(x) - F_{\nu_c}(x) \big) \, \rho(dc). \tag{20}$$

**Jensen inside the $x$–integral.**
Since $|\cdot|^p$ is convex and $\rho$ is a probability measure,

$$\left| \int_\Omega \big( F_{\mu_c}(x) - F_{\nu_c}(x) \big) \, \rho(dc) \right|^p \leq \int_\Omega \big| F_{\mu_c}(x) - F_{\nu_c}(x) \big|^p \, \rho(dc) \quad \text{(Jensen on } \Omega).$$

**Integrate over $x$ and swap the order.**
Therefore

$$\ell_p\Big( \int \mu_c \, d\rho, \int \nu_c \, d\rho \Big)^p = \int_\mathbb{R} \big| F_{\int \mu_c \, d\rho}(x) - F_{\int \nu_c \, d\rho}(x) \big|^p \, dx \quad \text{(def. of } \ell_p)$$

$$= \int_\mathbb{R} \left| \int_\Omega \big( F_{\mu_c}(x) - F_{\nu_c}(x) \big) \, \rho(dc) \right|^p \, dx \quad \text{(by (20))}$$

$$\leq \int_\mathbb{R} \int_\Omega \big| F_{\mu_c}(x) - F_{\nu_c}(x) \big|^p \, \rho(dc) \, dx \quad \text{(Jensen on } \Omega)$$

$$= \int_\Omega \int_\mathbb{R} \big| F_{\mu_c}(x) - F_{\nu_c}(x) \big|^p \, dx \, \rho(dc) \quad \text{(Fubini–Tonelli)}$$

$$= \int_\Omega \ell_p(\mu_c, \nu_c)^p \, \rho(dc).$$

Taking the $p$-th root yields the claim. $\square$

### A.3. MMD

The Maximum Mean Discrepancy (MMD) is a kernel-based discrepancy that measures how far apart two probability laws are in a reproducing kernel Hilbert space (RKHS) $\mathcal{H}$. Given a *positive definite* kernel $k \colon X \times X \to \mathbb{R}$ with feature map $\phi(x) = k(x, \cdot)$, each distribution admits a *mean embedding* in $\mathcal{H}$:

$$\mu_P = \int_X \phi(x)\, dP(x), \qquad \mu_Q = \int_X \phi(x)\, dQ(x). \tag{21}$$

The distance between these embeddings defines

$$\mathbf{MMD}_k(P, Q) \;=\; \|\mu_P - \mu_Q\|_{\mathcal{H}}. \tag{22}$$

Its square admits the familiar expansion in terms of kernel expectations:

$$\begin{aligned}
\mathbf{MMD}_k^2(P, Q) &= \|\mu_P - \mu_Q\|_{\mathcal{H}}^2 \\
&= \iint k(x, x')\, dP(x)\, dP(x') \;+\; \iint k(y, y')\, dQ(y)\, dQ(y') \\
&\quad -\; 2 \iint k(x, y)\, dP(x)\, dQ(y).
\end{aligned} \tag{23}$$

In this paper we also use a closely related formulation that does not require explicitly starting from an RKHS. Let $X$ be a measurable space and let $k : X \times X \to \mathbb{R}$ be symmetric. We say that $k$ is *conditionally positive definite (CPD)* if

$$\iint_{X \times X} k(x, x')\, d\mu(x)\, d\mu(x') \;\geq\; 0 \qquad \text{for all finite signed measures } \mu \text{ on } X \text{ with } \mu(X) = 0, \tag{24}$$

and *conditionally strictly positive definite (CSPD)* if the inequality is strict for every nonzero such $\mu$. Under this condition, the quadratic form

$$\gamma_k(P, Q)^2 := \iint k(x, y)\, d(P - Q)(x)\, d(P - Q)(y) \tag{25}$$

is nonnegative and behaves like an MMD. Concretely, define

$$\rho_k(x, y) \;:=\; k(x, x) + k(y, y) - 2k(x, y), \tag{26}$$

fix any $z_0 \in X$, and introduce the one-point centered (distance-induced) kernel

$$\begin{aligned}
k^\circ(x, y) &:= \tfrac{1}{2}\big[\rho_k(x, z_0) + \rho_k(y, z_0) - \rho_k(x, y)\big] \\
&= \; k(x, y) - k(x, z_0) - k(z_0, y) + k(z_0, z_0).
\end{aligned} \tag{27}$$

Then $k^\circ$ is positive definite and admits an RKHS $\mathcal{H}_{k^\circ}$, and for any $P, Q$ with finite integrals,

$$\begin{aligned}
\gamma_k(P, Q)^2 &= \iint k^\circ(x, y)\, d(P - Q)(x)\, d(P - Q)(y) \\
&= \big\|\mu_{k^\circ}(P) - \mu_{k^\circ}(Q)\big\|_{\mathcal{H}_{k^\circ}}^2 = \mathbf{MMD}_{k^\circ}(P, Q)^2.
\end{aligned} \tag{28}$$

This is the standard distance-induced kernel construction and equivalence result of Sejdinovic et al. (2013).

**Convention.** To keep notation light, we write $\mathbf{MMD}_k$ to denote either the usual RKHS MMD when $k$ is positive definite, or more generally the square root of the quadratic form $\gamma_k(\mu, \nu)^2 = \iint k\, d(\mu - \nu)\, d(\mu - \nu)$ when $k$ is only conditionally positive definite. In the latter case, one may always replace $k$ by its one-point centered kernel $k^\circ$ in (27), which is positive definite, and then $\gamma_k(\mu, \nu) = \mathbf{MMD}_{k^\circ}(\mu, \nu)$.

### A.3.1. ESTIMATOR

The $\mathbf{MMD}$ can be approximated from samples in two standard ways, both originating from Gretton et al. (2012). Given two sets of $m$ samples $\{x_i\}_{i=1}^m \sim P$ and $\{y_i\}_{i=1}^m \sim Q$, the *biased* estimator is

$$\widehat{\mathbf{MMD}}_b^2 = \frac{1}{m^2} \sum_{i,j=1}^m k(x_i, x_j) + \frac{1}{m^2} \sum_{i,j=1}^m k(y_i, y_j) - \frac{2}{m^2} \sum_{i,j=1}^m k(x_i, y_j). \tag{29}$$

while the *unbiased* estimator excludes diagonal terms:

$$\widehat{\mathbf{MMD}}_u^2 = \frac{1}{m(m-1)} \sum_{\substack{i,j=1 \\ i \neq j}}^m k(x_i, x_j) + \frac{1}{m(m-1)} \sum_{\substack{i,j=1 \\ i \neq j}}^m k(y_i, y_j) - \frac{2}{m^2} \sum_{i,j=1}^m k(x_i, y_j). \tag{30}$$

Although the unbiased form eliminates a small finite-sample bias, the biased estimator is often preferred in practice. In particular, applications of $\mathbf{MMD}$ to distributional RL (Nguyen et al., 2020; Killingberg & Langseth, 2023) consistently rely on the biased version due to its lower variance and greater numerical stability during training.

### A.3.2. PROPERTIES

**Metric.**

**Proposition A.6** (MMD as a Metric on $\mathcal{P}(X)$). *Let $k \colon X \times X \to \mathbb{R}$ be a symmetric kernel, and interpret $\mathbf{MMD}_k$ according to the convention above. We say that $\mathbf{MMD}_k$ defines a metric on $\mathcal{P}(X)$ iff $k$ is* conditionally strictly positive definite (CSPD), *i.e., for every nonzero finite signed Borel measure $\nu$ with $\nu(X) = 0$,*

$$\iint_{X \times X} k(x, y) \, d\nu(x) \, d\nu(y) \; > \; 0.$$

*Then $\mathbf{MMD}_k$ satisfies the metric axioms on $\mathcal{P}(X)$:*

1. Nonnegativity: $\mathbf{MMD}_k(P, Q) \geq 0$.

2. Symmetry: $\mathbf{MMD}_k(P, Q) = \mathbf{MMD}_k(Q, P)$.

3. Identity of indiscernibles: $\mathbf{MMD}_k(P, Q) = 0 \Rightarrow P = Q$.

4. Triangle inequality: *for any $P, Q, R \in \mathcal{P}(X)$, $\mathbf{MMD}_k(P, Q) \leq \mathbf{MMD}_k(P, R) + \mathbf{MMD}_k(R, Q)$.*

Justification. *This is the standard correspondence between negative-type distances, distance-induced kernels, and RKHS MMD metrics as outlined in (Sejdinovic et al., 2013).*

**Scale contraction.** Whether $\mathbf{MMD}_k$ contracts under scaling depends on the choice of kernel. In the contraction-based distributional RL literature, only two kernel families have been instantiated to obtain a Bellman contraction: the negative-type power family introduced by Nguyen-Tang et al. (2021) and the multiquadric kernel studied by Killingberg & Langseth (2023). We record the corresponding scale-contraction bounds in Proposition A.8 (negative-type power kernels) and Proposition A.7 (multiquadric kernel). The multiquadric bound is restated from Debes & Tuytelaars (2026, Lemma 10); the proof is given there.

**Proposition A.7** (Scale contraction of $\mathbf{MMD}^2$ for the multiquadric kernel). *Let $h > 0$ and consider the (negative) multiquadric kernel*

$$k_h(x, y) \; = \; -\sqrt{1 + h^2 \|x - y\|^2}, \qquad x, y \in \mathbb{R}^d.$$

*For probability measures $\mu, \nu$ on $\mathbb{R}^d$ with finite second moments, define*

$$\mathbf{MMD}_{k_h}^2(\mu, \nu) \; = \; \mathbb{E}\, k_h(X, X') + \mathbb{E}\, k_h(Y, Y') - 2\, \mathbb{E}\, k_h(X, Y),$$

*where $X, X' \sim \mu$ i.i.d. and $Y, Y' \sim \nu$ i.i.d. Let $S_s(x) := s\, x$ for $s \in [0, 1]$. Then*

$$\mathbf{MMD}_{k_h}^2\big((S_s)_\# \mu, (S_s)_\# \nu\big) \; \leq \; s\, \mathbf{MMD}_{k_h}^2(\mu, \nu), \qquad 0 \leq s \leq 1. \tag{31}$$

*Equivalently,*

$$\mathbf{MMD}_{k_h}\big((S_s)_\# \mu, (S_s)_\# \nu\big) \; \leq \; \sqrt{s}\, \mathbf{MMD}_{k_h}(\mu, \nu), \qquad 0 \leq s \leq 1. \tag{32}$$

**Proposition A.8** (Scale contraction of $\mathrm{MMD}^2$ for the negative-type power family). *Fix $\beta \in (0, 2]$ and consider the (negative) power kernel*

$$k_\beta(x, y) \; := \; -\|x - y\|^\beta, \qquad x, y \in \mathbb{R}^d.$$

*For probability measures $\mu, \nu$ on $\mathbb{R}^d$ with finite $\beta$-th moments, define*

$$\mathrm{MMD}^2_{k_\beta}(\mu, \nu) \; = \; \mathbb{E}\, k_\beta(X, X') + \mathbb{E}\, k_\beta(Y, Y') - 2\,\mathbb{E}\, k_\beta(X, Y),$$

*where $X, X' \sim \mu$ i.i.d. and $Y, Y' \sim \nu$ i.i.d. Let $S_s(x) := s\,x$ for $s \in [0, 1]$. Then*

$$\mathrm{MMD}^2_{k_\beta}\big((S_s)_{\#}\mu, (S_s)_{\#}\nu\big) \; = \; s^\beta \, \mathrm{MMD}^2_{k_\beta}(\mu, \nu), \qquad 0 \le s \le 1. \tag{33}$$

*Consequently,*

$$\mathrm{MMD}_{k_\beta}\big((S_s)_{\#}\mu, (S_s)_{\#}\nu\big) \; = \; s^{\beta/2} \, \mathrm{MMD}_{k_\beta}(\mu, \nu), \qquad 0 \le s \le 1. \tag{34}$$

*In particular, $\Delta := \mathrm{MMD}^2_{k_\beta}$ satisfies* (S) *with $c(s) = s^\beta$ on $[0, 1]$ (and $\Delta := \mathrm{MMD}_{k_\beta}$ satisfies* (S) *with $c(s) = s^{\beta/2}$).*

*Proof.* Let $X, X' \sim \mu$ i.i.d. and $Y, Y' \sim \nu$ i.i.d. Then $sX, sX' \sim (S_s)_{\#}\mu$ and $sY, sY' \sim (S_s)_{\#}\nu$. By homogeneity of the norm, $\|sx - sy\| = s\|x - y\|$, hence for all $x, y \in \mathbb{R}^d$,

$$k_\beta(sx, sy) = -\|sx - sy\|^\beta = -(s\|x - y\|)^\beta = s^\beta \, k_\beta(x, y).$$

Therefore,

$$\begin{aligned}
\mathrm{MMD}^2_{k_\beta}\big((S_s)_{\#}\mu, (S_s)_{\#}\nu\big) &= \mathbb{E}\, k_\beta(sX, sX') + \mathbb{E}\, k_\beta(sY, sY') - 2\,\mathbb{E}\, k_\beta(sX, sY) \\
&= s^\beta \Big( \mathbb{E}\, k_\beta(X, X') + \mathbb{E}\, k_\beta(Y, Y') - 2\,\mathbb{E}\, k_\beta(X, Y) \Big) \\
&= s^\beta \, \mathrm{MMD}^2_{k_\beta}(\mu, \nu),
\end{aligned}$$

which proves (33). Taking square roots yields (34). Finally, on $[0, 1]$ the function $c(s) = s^\beta$ is nondecreasing, satisfies $c(s) \le 1$, and $c(s) < 1$ for all $s \in [0, 1)$ since $\beta > 0$, so (S) holds. $\qquad\square$

**p-convexity.**

**Proposition A.9** (Mixture $p$–convexity of $\mathrm{MMD}_k$ in an RKHS). *Let $k : X \times X \to \mathbb{R}$ be a symmetric positive–semidefinite reproducing kernel with RKHS $(\mathcal{H}, \langle \cdot, \cdot \rangle)$ and feature map $\phi(x) = k(x, \cdot)$. Let $(\Omega, \mathcal{F}, \rho)$ be a probability space, and let $(\mu_c)_{c \in \Omega}$ and $(\nu_c)_{c \in \Omega}$ be measurable families of probability measures on $X$ for which the mean embeddings $\mu_{\mu_c} := \int_X \phi \, d\mu_c$ and $\mu_{\nu_c} := \int_X \phi \, d\nu_c$ exist in $\mathcal{H}$. Define the mixtures $\bar{\mu} := \int_\Omega \mu_c \, \rho(dc)$ and $\bar{\nu} := \int_\Omega \nu_c \, \rho(dc)$. Assume all mean embeddings and integrals below are well defined. Then for every $p \ge 1$,*

$$\boxed{\; \mathrm{MMD}_k(\bar{\mu}, \bar{\nu}) \; \le \; \left( \int_\Omega \mathrm{MMD}_k(\mu_c, \nu_c)^p \, \rho(dc) \right)^{1/p} \;}$$

*Proof.* By linearity of mean embeddings,

$$\mu_{\bar{\mu}} = \int_\Omega \mu_{\mu_c} \, \rho(dc), \qquad \mu_{\bar{\nu}} = \int_\Omega \mu_{\nu_c} \, \rho(dc),$$

where $\mu_{\mu_c} = \int_X \phi(x) \, d\mu_c(x)$ and $\mu_{\nu_c} = \int_X \phi(x) \, d\nu_c(x)$ are elements of $\mathcal{H}$. Thus,

$$\mu_{\bar{\mu}} - \mu_{\bar{\nu}} = \int_\Omega v(c) \, \rho(dc), \qquad v(c) := \mu_{\mu_c} - \mu_{\nu_c} \in \mathcal{H}.$$

Hence

$$\begin{aligned}
\mathrm{MMD}_k(\bar{\mu}, \bar{\nu}) = \|\mu_{\bar{\mu}} - \mu_{\bar{\nu}}\|_{\mathcal{H}} &= \left\| \int_\Omega v(c) \, \rho(dc) \right\|_{\mathcal{H}} \\
&\le \int_\Omega \|v(c)\|_{\mathcal{H}} \, \rho(dc) && \text{(triangle inequality in } \mathcal{H}) \\
&\le \left( \int_\Omega \|v(c)\|^p_{\mathcal{H}} \, \rho(dc) \right)^{1/p} && (L^1 \le L^p \text{ on a probability space).}
\end{aligned}$$

Finally, $\|v(c)\|_{\mathcal{H}} = \|\mu_{\mu_c} - \mu_{\nu_c}\|_{\mathcal{H}} = \mathrm{MMD}_k(\mu_c, \nu_c)$, which gives the claim. $\qquad\square$

**Proposition A.10** (Mixture $p$–convexity for CPD kernels via the distance–induced RKHS). *Let $k : X \times X \to \mathbb{R}$ be conditionally positive definite (CPD) and let $k^\circ$ be the associated distance–induced (one–point centered) kernel using the equivalence* (28) *with the one-point centered kernel* (27), *so that for all probabilities $P, Q$ with finite integrals,*

$$\gamma_k(P, Q) = \mathbf{MMD}_{k^\circ}(P, Q).$$

*Let $(\Omega, \mathcal{F}, \rho)$ be a probability space, and let $(\mu_c)_{c \in \Omega}$ and $(\nu_c)_{c \in \Omega}$ be measurable families of probability measures on $X$ with finite embeddings for $k^\circ$. Define the mixtures $\bar{\mu} := \int_\Omega \mu_c \, \rho(dc)$ and $\bar{\nu} := \int_\Omega \nu_c \, \rho(dc)$. Then for every $p \geq 1$,*

$$\gamma_k(\bar{\mu}, \bar{\nu}) \leq \left( \int_\Omega \gamma_k(\mu_c, \nu_c)^p \, \rho(dc) \right)^{1/p}.$$

*Proof.* By (28), $\gamma_k = \mathbf{MMD}_{k^\circ}$. Applying Lemma A.9 to the PSD kernel $k^\circ$ and the families $(\mu_c), (\nu_c)$ yields

$$\mathbf{MMD}_{k^\circ}(\bar{\mu}, \bar{\nu}) \leq \left( \int_\Omega \mathbf{MMD}_{k^\circ}(\mu_c, \nu_c)^p \, \rho(dc) \right)^{1/p}.$$

Replacing $\mathbf{MMD}_{k^\circ}$ by $\gamma_k$ using the equivalence (28) gives the claim. $\qquad\square$

# B. Metric property

**Lemma B.1** (Basic metric properties of slicing (from (Nadjahi et al., 2020))). *Let $\Delta : \mathcal{P}(\mathbb{R}) \times \mathcal{P}(\mathbb{R}) \to [0, \infty]$ be a divergence and let $p \in [1, \infty)$. For $\mu, \nu \in \mathcal{P}(\mathbb{R}^d)$ define*

$$\mathbf{S}\Delta_p^p(\mu, \nu) \;=\; \int_{S^{d-1}} \Delta^p\big((P_\theta)_{\#}\mu, (P_\theta)_{\#}\nu\big) \, d\sigma(\theta),$$

*where $(P_\theta)_{\#}\mu$ is the pushforward of $\mu$ by $x \mapsto \langle \theta, x \rangle$ and $\sigma$ is the uniform measure on $S^{d-1}$.* This reproduces Proposition 1 (Nadjahi et al., 2020).

**Statement.** *If $\Delta$ is a metric on $\mathcal{P}(\mathbb{R})$, then $\mathbf{S}\Delta_p$ is a metric on $\mathcal{P}(\mathbb{R}^d)$. In particular:*

- **Nonnegativity and symmetry.** *If $\Delta$ is nonnegative (resp. symmetric) on $\mathcal{P}(\mathbb{R})$, then $\mathbf{S}\Delta_p$ is nonnegative (resp. symmetric) on $\mathcal{P}(\mathbb{R}^d)$.*

- **Identity of indiscernibles.** *If $\Delta(\alpha, \beta) = 0$ iff $\alpha = \beta$ for $\alpha, \beta \in \mathcal{P}(\mathbb{R})$, then $\mathbf{S}\Delta_p(\mu, \nu) = 0$ iff $\mu = \nu$ for $\mu, \nu \in \mathcal{P}(\mathbb{R}^d)$.*

- **Triangle inequality.** *If $\Delta$ is a metric on $\mathcal{P}(\mathbb{R})$, then $\mathbf{S}\Delta_p$ satisfies the triangle inequality on $\mathcal{P}(\mathbb{R}^d)$.*

*Proof (this is a reproduction from (Nadjahi et al., 2020), App. A.1).* We prove that $\mathbf{S}\Delta_p$ satisfies the three defining properties required for a metric on $\mathcal{P}(\mathbb{R}^d)$.

**Nonnegativity and symmetry.**
This is immediate from the definition since the integrand inherits these properties from $\Delta$, and taking a $p$-th root preserves them.

**Identity of indiscernibles.**
We need to show that $\mathbf{S}\Delta_p(\mu, \nu) = 0$ implies $\mu = \nu$. (The converse implication is immediate from the definition, since if $\mu = \nu$ then every slice coincides and the integral vanishes.)

(1) Assume $\mathbf{S}\Delta_p(\mu, \nu) = 0$. Since the integrand is nonnegative, this yields

$$\Delta\big((P_\theta)_{\#}\mu, (P_\theta)_{\#}\nu\big) = 0 \quad \text{for } \sigma\text{-almost every } \theta \in S^{d-1}.$$

By the base property of $\Delta$ in one dimension, we obtain

$$(P_\theta)_{\#}\mu = (P_\theta)_{\#}\nu \quad \text{for } \sigma\text{-almost every } \theta \in S^{d-1}.$$

*Notation.* For a probability measure $\xi$ on $\mathbb{R}^d$, we write $\widehat{\xi}$ for its characteristic function:

$$\widehat{\xi}(z) \;=\; \int_{\mathbb{R}^d} e^{i\langle z, x\rangle} \, d\xi(x), \qquad z \in \mathbb{R}^d.$$

(2) By Lemma C.4, the one–dimensional pushforward $(P_\theta)_{\#}\xi$ satisfies

$$\widehat{(P_\theta)_{\#}\xi}(t) = \int_{\mathbb{R}} e^{itu} \, d\big((P_\theta)_{\#}\xi\big)(u) = \int_{\mathbb{R}^d} e^{it\langle \theta, x\rangle} \, d\xi(x) = \widehat{\xi}(t\theta), \quad t \in \mathbb{R}.$$

Hence $(P_\theta)_{\#}\mu = (P_\theta)_{\#}\nu$ implies

$$\widehat{\mu}(t\theta) = \widehat{\nu}(t\theta) \quad \text{for } \sigma\text{-almost every } \theta \in S^{d-1} \text{ and all } t \in \mathbb{R}.$$

*Interpretation.* Projecting onto $\theta$ in the original space corresponds to restricting $\widehat{\mu}$ to the line $\{t\theta : t \in \mathbb{R}\}$ in frequency space. Thus the two characteristic functions agree along almost all such lines.

(3) Therefore $\widehat{\mu} = \widehat{\nu}$ on $\mathbb{R}^d$, and by the injectivity of characteristic functions (distinct measures cannot share the same characteristic function; see Theorem 26.2 in (Billingsley, 2017)) we conclude $\mu = \nu$.

**Triangle inequality.**

(iii) Assume that $\Delta$ is a metric on $\mathcal{P}(\mathbb{R})$. Let $\mu, \nu, \xi \in \mathcal{P}(\mathbb{R}^d)$. For every $\theta \in S^{d-1}$, the base triangle inequality gives

$$\Delta\big((P_\theta)_{\#}\mu, (P_\theta)_{\#}\nu\big) \;\leq\; \Delta\big((P_\theta)_{\#}\mu, (P_\theta)_{\#}\xi\big) + \Delta\big((P_\theta)_{\#}\xi, (P_\theta)_{\#}\nu\big).$$

Taking the $p$-th power and integrating over the sphere yields

$$\int_{S^{d-1}} \Delta^p\big((P_\theta)_{\#}\mu, (P_\theta)_{\#}\nu\big)\, d\sigma(\theta) \;\leq\; \int_{S^{d-1}} \big[\Delta\big((P_\theta)_{\#}\mu, (P_\theta)_{\#}\xi\big) + \Delta\big((P_\theta)_{\#}\xi, (P_\theta)_{\#}\nu\big)\big]^p \, d\sigma(\theta).$$

*Minkowski's inequality.* For $p \geq 1$ and measurable $f, g$ on a measure space $(X, \mu)$,

$$\left(\int_X |f(x) + g(x)|^p \, d\mu(x)\right)^{1/p} \;\leq\; \left(\int_X |f(x)|^p \, d\mu(x)\right)^{1/p} + \left(\int_X |g(x)|^p \, d\mu(x)\right)^{1/p}.$$

Using this inequality with $X = S^{d-1}$, $\mu = \sigma$, and

$$f(\theta) = \Delta\big((P_\theta)_{\#}\mu, (P_\theta)_{\#}\xi\big), \qquad g(\theta) = \Delta\big((P_\theta)_{\#}\xi, (P_\theta)_{\#}\nu\big),$$

we obtain

$$\mathbf{S}\Delta_p(\mu, \nu) \;\leq\; \mathbf{S}\Delta_p(\mu, \xi) + \mathbf{S}\Delta_p(\xi, \nu).$$

$\square$

**Lemma B.2** (Max–sliced metric properties). *Let $\Delta : \mathcal{P}(\mathbb{R}) \times \mathcal{P}(\mathbb{R}) \to [0, \infty]$ be a metric on $\mathcal{P}(\mathbb{R})$. For $\mu, \nu \in \mathcal{P}(\mathbb{R}^d)$ define*

$$\mathbf{MS}\Delta(\mu, \nu) := \sup_{\theta \in \mathbb{S}^{d-1}} \Delta\big((P_\theta)_{\#}\mu, (P_\theta)_{\#}\nu\big), \qquad P_\theta(x) = \langle \theta, x \rangle.$$

*Then $\mathbf{MS}\Delta$ is a metric on $\mathcal{P}(\mathbb{R}^d)$: it is nonnegative and symmetric, satisfies the identity of indiscernibles, and obeys the triangle inequality.*

*Proof.* We prove that $\mathbf{MS}\Delta$ satisfies the three defining properties required for a metric on $\mathcal{P}(\mathbb{R}^d)$.

**Nonnegativity and symmetry.** Each slice is nonnegative and symmetric because $\Delta$ is; taking a supremum preserves both properties.

**Identity of indiscernibles.** If $\mu = \nu$ then every slice is equal, so $\mathbf{MS}\Delta(\mu, \nu) = 0$. Conversely, if $\mathbf{MS}\Delta(\mu, \nu) = 0$, then

$$(P_\theta)_{\#}\mu = (P_\theta)_{\#}\nu \quad \text{for all } \theta \in \mathbb{S}^{d-1}.$$

The argument given in Proposition B.1 for the sliced case then applies verbatim, showing that $\mu = \nu$.

**Triangle inequality.** For any $\theta \in \mathbb{S}^{d-1}$ and any $\mu, \nu, \xi \in \mathcal{P}(\mathbb{R}^d)$, the base metric property yields

$$\Delta\big((P_\theta)_{\#}\mu, (P_\theta)_{\#}\nu\big) \;\leq\; \Delta\big((P_\theta)_{\#}\mu, (P_\theta)_{\#}\xi\big) + \Delta\big((P_\theta)_{\#}\xi, (P_\theta)_{\#}\nu\big).$$

Taking the supremum over $\theta$ on both sides gives

$$\mathbf{MS}\Delta(\mu, \nu) \;\leq\; \mathbf{MS}\Delta(\mu, \xi) + \mathbf{MS}\Delta(\xi, \nu).$$

All three metric axioms hold; hence $\mathbf{MS}\Delta$ is a metric on $\mathcal{P}(\mathbb{R}^d)$. $\square$

**Lemma B.3** (Supremum lift preserves metricity for SPDs and MaxSPDs—follows closely from Nguyen-Tang et al. (2021), Proposition 1 (Appendix A.1)). *Let $\mathcal{D}$ be a metric on $\mathcal{P}(\mathbb{R}^d)$. (In our use, $\mathcal{D}$ will be either the SPD $\mathbf{S}\Delta^{p,p}$ or the MaxSPD $\mathbf{MS}\Delta$.) Define, for $\mu, \nu : \mathcal{S} \times \mathcal{A} \to \mathcal{P}(\mathbb{R}^d)$,*

$$\overline{\mathcal{D}}(\mu, \nu) := \sup_{(s,a) \in \mathcal{S} \times \mathcal{A}} \mathcal{D}\big(\mu(s, a), \nu(s, a)\big).$$

*Then $\overline{\mathcal{D}}$ is a metric on $\mathcal{P}(\mathbb{R}^d)^{\mathcal{S} \times \mathcal{A}}$.*

*Proof.* **Nonnegativity and symmetry.** Since $\mathcal{D}$ is nonnegative and symmetric pointwise, the supremum of such quantities preserves these properties. Hence $\overline{\mathcal{D}}(\mu, \nu) \geq 0$ and $\overline{\mathcal{D}}(\mu, \nu) = \overline{\mathcal{D}}(\nu, \mu)$.

**Identity of indiscernibles.** If $\mu = \nu$, then every term vanishes and $\overline{\mathcal{D}}(\mu, \nu) = 0$. Conversely, if $\overline{\mathcal{D}}(\mu, \nu) = 0$, then $\mathcal{D}(\mu(s, a), \nu(s, a)) = 0$ for each $(s, a)$, which by metricity of $\mathcal{D}$ implies $\mu(s, a) = \nu(s, a)$ everywhere, hence $\mu = \nu$.

**Triangle inequality.** Let $\mu, \nu, \eta : \mathcal{S} \times \mathcal{A} \to \mathcal{P}(\mathbb{R}^d)$. Then

$$\overline{\mathcal{D}}(\mu, \nu) = \sup_{(s,a)} \mathcal{D}\big(\mu(s, a),\, \nu(s, a)\big) \tag{35}$$

$$\overset{(a)}{\leq} \sup_{(s,a)} \Big\{ \mathcal{D}\big(\mu(s, a), \eta(s, a)\big) + \mathcal{D}\big(\eta(s, a), \nu(s, a)\big) \Big\} \tag{36}$$

$$\overset{(b)}{\leq} \sup_{(s,a)} \mathcal{D}\big(\mu(s, a), \eta(s, a)\big) \;+\; \sup_{(s,a)} \mathcal{D}\big(\eta(s, a), \nu(s, a)\big) \tag{37}$$

$$= \overline{\mathcal{D}}(\mu, \eta) + \overline{\mathcal{D}}(\eta, \nu). \tag{38}$$

Here (a) is the pointwise triangle inequality for $\mathcal{D}$, and (b) uses $\sup(A+B) \leq \sup A + \sup B$.

Thus $\overline{\mathcal{D}}$ satisfies all four metric axioms. Specializing $\mathcal{D}$ to $\mathbf{S}\Delta^{\rho,p}$ or $\mathbf{MS}\Delta$ yields that $\overline{\mathbf{S}\Delta}^{\rho,p}$ and $\overline{\mathbf{MS}\Delta}$ are metrics on $\mathcal{P}(\mathbb{R}^d)^{\mathcal{S} \times \mathcal{A}}$. $\qquad\square$

**Theorem B.1** (Global metricity of (max-)sliced lifts). *Let $\Delta$ be a metric on $\mathcal{P}(\mathbb{R})$ and let $p \in [1, \infty)$. Let $\sigma$ denote the uniform probability measure on the unit sphere $S^{d-1} \subset \mathbb{R}^d$. Define the uniform sliced divergence*

$$\mathbf{S}\Delta_p(\mu, \nu) := \left( \int_{S^{d-1}} \Delta^p\big((P_\theta)_\# \mu, (P_\theta)_\# \nu\big)\, d\sigma(\theta) \right)^{1/p},$$

*and the max–sliced divergence*

$$\mathbf{MS}\Delta(\mu, \nu) := \sup_{\theta \in S^{d-1}} \Delta\big((P_\theta)_\# \mu, (P_\theta)_\# \nu\big), \qquad P_\theta(x) = \langle \theta, x \rangle.$$

*Then:*

1. *$\mathbf{S}\Delta_p$ is a metric on $\mathcal{P}(\mathbb{R}^d)$.*

2. *$\mathbf{MS}\Delta$ is a metric on $\mathcal{P}(\mathbb{R}^d)$.*

3. *For return–distribution functions $\eta_i : \mathcal{S} \times \mathcal{A} \to \mathcal{P}(\mathbb{R}^d)$, the supremum lifts*

$$\overline{\mathbf{S}\Delta}_p(\eta_1, \eta_2) := \sup_{(s,a)} \mathbf{S}\Delta_p\big(\eta_1(s, a), \eta_2(s, a)\big),$$

   *and*

$$\overline{\mathbf{MS}\Delta}(\eta_1, \eta_2) := \sup_{(s,a)} \mathbf{MS}\Delta\big(\eta_1(s, a), \eta_2(s, a)\big),$$

   *are metrics on $\mathcal{P}(\mathbb{R}^d)^{\mathcal{S} \times \mathcal{A}}$.*

*Proof. (i) is Lemma B.1; (ii) is Lemma B.2; (iii) follows from Lemma B.3 by taking $\mathcal{D} = \mathbf{S}\Delta_p$ or $\mathcal{D} = \mathbf{MS}\Delta$.*

# C. Contraction property

In Section C.1, we recall sufficient conditions for the univariate distributional Bellman operator to be a contraction. In Section C.2, we generalize these sufficient conditions to anisotropic updates, allowing random linear maps controlled by an operator-norm bound. In Section C.3, we apply this framework to uniform slicing and infer sufficient conditions on the underlying one-dimensional base divergence. In Section C.4, we nuance the picture with explicit counterexamples showing that norm control alone does not ensure contraction for several multivariate objectives. Finally, in Section C.5, we propose a solution based on max slicing and obtain sufficient conditions for a contraction guarantee under the same norm control.

**Lemma C.4** (Push-forward law identity). *Let $Z$ be a random variable with distribution $\mu$, and let $f$ be any measurable function. Then*

$$\boxed{f_{\#}\mu = \mathrm{Law}\big(f(Z)\big).}$$

*Proof.* For any Borel set $A$,

$$\Pr\big(f(Z) \in A\big) = \Pr\big(Z \in f^{-1}(A)\big) = \mu\big(f^{-1}(A)\big) = f_{\#}\mu(A).$$

Since this holds for all $A$, we conclude $f_{\#}\mu = \mathrm{Law}(f(Z))$. $\qquad\square$

## C.1. Univariate case

**Lemma C.5** (Univariate affine push-forward contraction). *Let $\Delta$ be a metric on $\mathcal{P}(\mathbb{R})$. Assume for all $\mu, \nu \in \mathcal{P}(\mathbb{R})$:*

(T) *Translation non-expansion: for every $t \in \mathbb{R}$,*

$$\Delta\big((T_t)_{\#}\mu, (T_t)_{\#}\nu\big) \le \Delta(\mu, \nu), \quad T_t(x) = x + t.$$

(S) *Scale contraction: there exists a nondecreasing $c : [0, \infty) \to [0, \infty)$ such that for every $s \in [0, 1]$,*

$$\Delta\big((x \mapsto sx)_{\#}\mu, (x \mapsto sx)_{\#}\nu\big) \le c(s)\,\Delta(\mu, \nu),$$

*with $c(s) \le 1$ for all $s \in [0, 1]$ and $c(s) < 1$ for all $s \in [0, 1)$.*

*Let $F(x) = t + \gamma x$ with arbitrary $t \in \mathbb{R}$ and the same $\gamma \in [0, 1)$. Then, for all $\mu, \nu \in \mathcal{P}(\mathbb{R})$,*

$$\boxed{\Delta\big(F_{\#}\mu, F_{\#}\nu\big) \le c(\gamma)\,\Delta(\mu, \nu)}$$

*Consequently, the push-forward $F_{\#}$ is a contraction (with factor $c(\gamma) < 1$) on $(\mathcal{P}(\mathbb{R}), \Delta)$.*

*Proof.* Let $U \sim \mu$ and $V \sim \nu$. By Lemma C.4,

$$\Delta\big(F_{\#}\mu, F_{\#}\nu\big) = \Delta\big(\mathrm{Law}(t + \gamma U), \mathrm{Law}(t + \gamma V)\big).$$

By (T),

$$\Delta\big(\mathrm{Law}(t + \gamma U), \mathrm{Law}(t + \gamma V)\big) \le \Delta\big(\mathrm{Law}(\gamma U), \mathrm{Law}(\gamma V)\big).$$

By (S) with $s = \gamma$,

$$\Delta\big(\mathrm{Law}(\gamma U), \mathrm{Law}(\gamma V)\big) \le c(\gamma)\,\Delta\big(\mathrm{Law}(U), \mathrm{Law}(V)\big) = c(\gamma)\,\Delta(\mu, \nu).$$

$$\square$$

**Lemma C.6** (Mixture $p$-convexity $\Rightarrow$ marginal bound). *Let $\Delta$ be a metric on $\mathcal{P}(\mathbb{R}^d)$ and fix $p \in [1, \infty)$. Assume $\Delta$ satisfies the mixture $p$-convexity property:*

$$\Delta\left(\int_\Omega \mu_c\,\rho(dc), \int_\Omega \nu_c\,\rho(dc)\right) \le \left(\int_\Omega \Delta(\mu_c, \nu_c)^p\,\rho(dc)\right)^{1/p}, \tag{39}$$

*for all probability spaces $(\Omega, \mathcal{F}, \rho)$ and measurable families $(\mu_c)_{c \in \Omega}, (\nu_c)_{c \in \Omega}$.*

*Let $C$ be a random variable with law $\rho$ and let $Z_1, Z_2$ be $\mathbb{R}^d$-valued random variables. If*

$$\sup_{c \in \Omega} \Delta\big(\mathrm{Law}(Z_1 \mid C = c),\, \mathrm{Law}(Z_2 \mid C = c)\big) \leq \delta,$$

*then*

$$\Delta\big(\mathrm{Law}(Z_1),\, \mathrm{Law}(Z_2)\big) \leq \delta.$$

*Proof.* Set $\mu_c := \mathrm{Law}(Z_1 \mid C = c)$ and $\nu_c := \mathrm{Law}(Z_2 \mid C = c)$. By the law of total probability,

$$\mathrm{Law}(Z_1) = \int_\Omega \mu_c\, \rho(dc), \qquad \mathrm{Law}(Z_2) = \int_\Omega \nu_c\, \rho(dc).$$

Define $f(c) := \Delta(\mu_c, \nu_c) \geq 0$. The hypothesis gives the pointwise bound $f(c) \leq \delta$ for all $c \in \Omega$. Applying (39) and then monotonicity of the integral,

$$\Delta\big(\mathrm{Law}(Z_1), \mathrm{Law}(Z_2)\big) \leq \Big( \int_\Omega f(c)^p\, \rho(dc) \Big)^{1/p} \leq \Big( \int_\Omega \delta^p\, \rho(dc) \Big)^{1/p} = \delta.$$

$\square$

**Theorem C.2** (Supremum-$\Delta$ contraction of the univariate distributional Bellman operator)**.** *This proposition slightly generalizes Theorem 4.25 of (Bellemare et al., 2023a).*

*Let $\Delta$ be a metric on $\mathcal{P}(\mathbb{R})$ and define*

$$\bar{\Delta}(\eta_1, \eta_2) := \sup_{(s,a)} \Delta\big(\eta_1(s,a),\, \eta_2(s,a)\big), \qquad \eta_i : \mathcal{S} \times \mathcal{A} \to \mathcal{P}(\mathbb{R}).$$

*Assume $\Delta$ satisfies:*

**(T)** *Translation nonexpansion:* $\Delta\big((T_t)_{\#}\mu, (T_t)_{\#}\nu\big) \leq \Delta(\mu, \nu)$ *for all $t \in \mathbb{R}$.*

**(S)** *Scale contraction: there exists a nondecreasing $c : [0, \infty) \to [0, \infty)$ such that for every $s \in [0, 1]$,*

$$\Delta\big((x \mapsto sx)_{\#}\mu,\, (x \mapsto sx)_{\#}\nu\big) \leq c(s)\, \Delta(\mu, \nu),$$

*with $c(s) \leq 1$ for all $s \in [0, 1]$ and $c(s) < 1$ for all $s \in [0, 1)$.*

**($M_p$)** *Mixture $p$-convexity: for some $p \in [1, \infty)$ and all probability spaces $(\Omega, \mathcal{F}, \rho)$ and measurable families $(\mu_c), (\nu_c) \subset \mathcal{P}(\mathbb{R})$,*

$$\Delta\Big( \int_\Omega \mu_c\, \rho(dc),\, \int_\Omega \nu_c\, \rho(dc) \Big) \leq \Big( \int_\Omega \Delta(\mu_c, \nu_c)^p\, \rho(dc) \Big)^{1/p}.$$

*For each $(s, a)$, let $C$ be a random element, set $(S', A') = g(s, a; C)$, and let $b_{s,a} : \mathrm{supp}(C) \to \mathbb{R}$ be measurable. Define*

$$(T^\pi \eta)(s, a) := \mathrm{Law}\big(b_{s,a}(C) + \gamma X'\big), \qquad X' \sim \eta(S', A') \text{ conditionally on } C.$$

*Then, for all $\eta_1, \eta_2$,*

$$\boxed{\bar{\Delta}\big(T^\pi \eta_1,\, T^\pi \eta_2\big) \leq c(\gamma)\, \bar{\Delta}\big(\eta_1, \eta_2\big).}$$

*Consequently, the operator $T^\pi$ is a contraction (with factor $c(\gamma) < 1$) on $(\mathcal{S} \times \mathcal{A} \to \mathcal{P}(\mathbb{R}), \bar{\Delta})$.*

*Proof.* By definition,

$$\bar{\Delta}\big(T^\pi \eta_1, T^\pi \eta_2\big) = \sup_{(s,a)} \Delta\big((T^\pi \eta_1)(s,a),\, (T^\pi \eta_2)(s,a)\big).$$

Fix $(s, a)$. Let $Z_i := b_{s,a}(C) + \gamma X_i'$ where, conditionally on $C$, $X_i' \sim \eta_i(S', A')$ and $(S', A') = g(s, a; C)$. By the push-forward law identity (Lemma C.4),

$$(T^\pi \eta_i)(s, a) = \mathrm{Law}(Z_i), \qquad \mathrm{Law}(X_i' \mid C) = \eta_i(S', A').$$

Condition on $C$ and define $\Phi_{s,a}(\cdot; C) : x \mapsto b_{s,a}(C) + \gamma x$. By the univariate affine push-forward contraction (Lemma C.5, using **(T)** and **(S)**), we have

$$\Delta\big(\mathrm{Law}(Z_1 \mid C), \mathrm{Law}(Z_2 \mid C)\big) \leq c(\gamma)\, \Delta\big(\mathrm{Law}(X_1' \mid C),\, \mathrm{Law}(X_2' \mid C)\big)$$
$$= c(\gamma)\, \Delta\big(\eta_1(S', A'),\, \eta_2(S', A')\big).$$

Apply mixture $p$-convexity **($M_p$)** to the conditional laws and then Lemma C.6 (with the pointwise bound $\Delta(\mathrm{Law}(Z_1 \mid C), \mathrm{Law}(Z_2 \mid C)) \leq c(\gamma)\, \Delta(\eta_1(S', A'), \eta_2(S', A')))$:

$$\Delta\big(\mathrm{Law}(Z_1),\, \mathrm{Law}(Z_2)\big) \leq \Big(\mathbb{E}\big[\Delta\big(\mathrm{Law}(Z_1 \mid C), \mathrm{Law}(Z_2 \mid C)\big)^p\big]\Big)^{1/p}$$
$$\leq c(\gamma) \Big(\mathbb{E}\big[\Delta\big(\eta_1(S', A'), \eta_2(S', A')\big)^p\big]\Big)^{1/p}$$
$$\leq c(\gamma)\, \bar{\Delta}(\eta_1, \eta_2).$$

Therefore,

$$\Delta\big((T^\pi \eta_1)(s, a),\, (T^\pi \eta_2)(s, a)\big) \leq c(\gamma)\, \bar{\Delta}(\eta_1, \eta_2).$$

Taking the supremum over $(s, a)$ yields the stated bound. $\qquad\square$

## C.2. Multivariate case

In this subsection, we keep the multivariate update as general as possible. We first treat the strong setting where the discount matrix may depend on the state action pair and may also be random, while only requiring a uniform operator norm control, as formalized in Theorem C.3. We then specialize this result to the standard isotropic case where the discount is a scaled identity matrix, as stated in Corollary C.1.

**Theorem C.3** (Supremum-$\Delta$ contraction of the multivariate distributional Bellman operator (anisotropic random linear map))**.** *This theorem generalizes Theorem C.2 to the multivariate setting with a norm-based discount criterion.*

*Let $\Delta$ be a metric on $\mathcal{P}(\mathbb{R}^d)$ and define the supremum metric*

$$\bar{\Delta}(\eta_1, \eta_2) := \sup_{(s,a)} \Delta\big(\eta_1(s, a), \eta_2(s, a)\big), \qquad \eta_i : \mathcal{S} \times \mathcal{A} \to \mathcal{P}(\mathbb{R}^d).$$

*Assume $\Delta$ satisfies:*

**(T)** *Translation nonexpansion: for all $t \in \mathbb{R}^d$,*

$$\Delta\big((T_t)_{\#}\mu, (T_t)_{\#}\nu\big) \leq \Delta(\mu, \nu), \qquad \forall \mu, \nu \in \mathcal{P}(\mathbb{R}^d),$$

*where $T_t(x) := x + t$.*

**($S_d$)** *Anisotropic scale contraction (norm-based): there exists a nondecreasing $c : [0, 1] \to [0, 1]$ such that for every linear map $A : \mathbb{R}^d \to \mathbb{R}^d$ with $\|A\|_{\mathrm{op}} \leq 1$,*

$$\Delta\big(A_{\#}\mu,\, A_{\#}\nu\big) \leq c(\|A\|_{\mathrm{op}})\, \Delta(\mu, \nu), \qquad \forall \mu, \nu \in \mathcal{P}(\mathbb{R}^d),$$

*with $c(s) < 1$ for all $s \in [0, 1)$.*

**($M_p$)** *Mixture $p$-convexity: for some $p \in [1, \infty)$ and all probability spaces $(\Omega, \mathcal{F}, \rho)$ and measurable families $(\mu_\omega), (\nu_\omega) \subset \mathcal{P}(\mathbb{R}^d)$,*

$$\Delta\Big(\int_\Omega \mu_\omega\, \rho(d\omega), \int_\Omega \nu_\omega\, \rho(d\omega)\Big) \leq \Big(\int_\Omega \Delta(\mu_\omega, \nu_\omega)^p\, \rho(d\omega)\Big)^{1/p}.$$

***Bellman update (anisotropic random linear map).*** *Fix* $(s, a)$*. Gather all environment/policy randomness into a single random element* $C$*, which determines the successor index through a measurable mapping*

$$(S', A') := g(s, a; C).$$

*At* $(s, a)$*, apply an affine transformation with* $C$*-dependent translation and* $C$*-dependent* linear map:

$$b_{s,a}(C) \in \mathbb{R}^d, \qquad A_{s,a}(C) \in \mathbb{R}^{d \times d},$$

*and assume* $\|A_{s,a}(C)\|_{\mathrm{op}} \le 1$ *for all* $C$*. Conditioned on* $C$*, draw the next sample from the law at the successor index:*

$$X' \mid C \sim \eta(S', A').$$

*Define the Bellman update as the push-forward of* $X'$ *by this affine map:*

$$(T^\pi \eta)(s, a) := \mathrm{Law}\big(b_{s,a}(C) + A_{s,a}(C)\, X'\big).$$

*Define, for each* $C$*,*

$$L(C) := \|A_{s,a}(C)\|_{\mathrm{op}},$$

*and the global envelope*

$$\bar{L} := \sup_{(s,a)} \sup_{C} L(C).$$

*Then, for all* $\eta_1, \eta_2$*,*

$$\boxed{\bar{\Delta}\big(T^\pi \eta_1,\ T^\pi \eta_2\big) \ \le\ c(\bar{L})\, \bar{\Delta}\big(\eta_1, \eta_2\big).}$$

*Consequently, if* $\bar{L} < 1$*, then* $T^\pi$ *is a contraction on* $(\mathcal{S} \times \mathcal{A} \to \mathcal{P}(\mathbb{R}^d), \bar{\Delta})$ *with factor* $c(\bar{L}) < 1$*.*

*Proof.* By definition,

$$\bar{\Delta}\big(T^\pi \eta_1, T^\pi \eta_2\big) = \sup_{(s,a)} \Delta\big((T^\pi \eta_1)(s, a),\ (T^\pi \eta_2)(s, a)\big).$$

Fix $(s, a)$ and condition on $C$. Define, for $i \in \{1, 2\}$,

$$X_i' \mid C \sim \eta_i(S', A'), \qquad Z_i := b_{s,a}(C) + A_{s,a}(C)\, X_i',$$

where $(S', A') = g(s, a; C)$. Let the $C$-dependent affine map be

$$\Phi_{s,a}(\cdot; C) : x \mapsto b_{s,a}(C) + A_{s,a}(C)\, x.$$

By the push-forward law identity,

$$\mathrm{Law}(Z_i \mid C) = \big(\Phi_{s,a}(\cdot; C)\big)_{\#}\, \eta_i(S', A').$$

**Conditional contraction at fixed** $C$**.** Using **(T)** and **(S**$_d$**)** (translation by $b_{s,a}(C)$ and linear map $A_{s,a}(C)$),

$$\Delta\big(\mathrm{Law}(Z_1 \mid C), \mathrm{Law}(Z_2 \mid C)\big) = \Delta\Big(\big(\Phi_{s,a}(\cdot; C)\big)_{\#}\eta_1(S', A'),\ \big(\Phi_{s,a}(\cdot; C)\big)_{\#}\eta_2(S', A')\Big)$$
$$\le c(\|A_{s,a}(C)\|_{\mathrm{op}})\, \Delta\big(\eta_1(S', A'), \eta_2(S', A')\big)$$
$$= c\big(L(C)\big)\, \Delta\big(\eta_1(S', A'), \eta_2(S', A')\big).$$

**Averaging over** $C$ **via mixture** $p$**-convexity.** Apply **(M**$_p$**)** to the family of conditional laws $\mathrm{Law}(Z_i \mid C)$:

$$\Delta\big(\mathrm{Law}(Z_1), \mathrm{Law}(Z_2)\big) \le \Big(\mathbb{E}\big[\,\Delta\big(\mathrm{Law}(Z_1 \mid C), \mathrm{Law}(Z_2 \mid C)\big)^p\big]\Big)^{1/p}$$
$$\le \Big(\mathbb{E}\big[\,\big(c(L(C))\, \Delta(\eta_1(S', A'), \eta_2(S', A'))\big)^p\big]\Big)^{1/p}.$$

Since $c$ is nondecreasing and $L(C) \leq \sup_C L(C) \leq \bar{L}$, we have $c(L(C)) \leq c(\bar{L})$ for all $C$, hence

$$\Delta\big(\mathrm{Law}(Z_1),\, \mathrm{Law}(Z_2)\big) \leq c(\bar{L}) \left(\mathbb{E}\big[\,\Delta(\eta_1(S', A'), \eta_2(S', A'))^p\big]\right)^{1/p}.$$

**Supremum bound.** For all realizations of $(S', A')$,

$$\Delta\big(\eta_1(S', A'),\, \eta_2(S', A')\big) \leq \sup_{(u,v)} \Delta\big(\eta_1(u, v),\, \eta_2(u, v)\big) = \bar{\Delta}(\eta_1, \eta_2),$$

so

$$\left(\mathbb{E}\big[\,\Delta(\eta_1(S', A'), \eta_2(S', A'))^p\big]\right)^{1/p} \leq \bar{\Delta}(\eta_1, \eta_2).$$

Combining the above,

$$\Delta\big((T^\pi \eta_1)(s, a),\, (T^\pi \eta_2)(s, a)\big) = \Delta\big(\mathrm{Law}(Z_1),\, \mathrm{Law}(Z_2)\big) \leq c(\bar{L})\, \bar{\Delta}(\eta_1, \eta_2).$$

Taking the supremum over $(s, a)$ yields

$$\bar{\Delta}\big(T^\pi \eta_1,\, T^\pi \eta_2\big) \leq c(\bar{L})\, \bar{\Delta}\big(\eta_1, \eta_2\big).$$

$\square$

**Corollary C.1** (Supremum-$\Delta$ contraction for a fixed isotropic discount $\Gamma = \gamma I$)**.** *Assume the setting of Theorem C.3, except that the linear map in the Bellman update is a fixed isotropic discount*

$$\Gamma = \gamma I_d, \qquad \gamma \in [0, 1].$$

*Equivalently, for every $(s, a)$ we consider the update*

$$(T^\pi \eta)(s, a) := \mathrm{Law}\big(b_{s,a}(C) + \Gamma X'\big), \qquad X' \mid C \sim \eta(S', A'),\ (S', A') = g(s, a; C).$$

*In this isotropic case, the anisotropic norm-based condition $(\mathbf{S}_d)$ is only used through the special choice $A = \Gamma$ and therefore reduces to the following scalar scaling assumption on the multivariate metric:*

$$(\mathbf{S}_d^{\mathrm{iso}}) \qquad \Delta\big((x \mapsto sx)_{\#}\mu,\, (x \mapsto sx)_{\#}\nu\big) \leq c(s)\, \Delta(\mu, \nu), \qquad \forall\, \mu, \nu \in \mathcal{P}(\mathbb{R}^d),\ \forall\, s \in [0, 1], \tag{40}$$

*for some nondecreasing $c : [0, 1] \to [0, 1]$ with $c(s) < 1$ for all $s \in [0, 1)$.*

*Assuming $(\mathbf{T})$, $(\mathbf{M}_p)$, and (40), we have for all $\eta_1, \eta_2$,*

$$\boxed{\bar{\Delta}\big(T^\pi \eta_1,\, T^\pi \eta_2\big) \ \leq\ c(\gamma)\, \bar{\Delta}\big(\eta_1, \eta_2\big).}$$

*Consequently, if $\gamma < 1$, the operator $T^\pi$ is a contraction with factor $c(\gamma) < 1$ on $(\mathcal{S} \times \mathcal{A} \to \mathcal{P}(\mathbb{R}^d), \bar{\Delta})$.*

*Proof.* This is a direct specialization of Theorem C.3. Set $A_{s,a}(C) \equiv \Gamma$ for all $(s, a)$ and all $C$. Then

$$L(C) = \|A_{s,a}(C)\|_{\mathrm{op}} = \|\Gamma\|_{\mathrm{op}}, \qquad \bar{L} = \sup_{(s,a)} \sup_C L(C) = \|\Gamma\|_{\mathrm{op}}.$$

For $\Gamma = \gamma I_d$ we have $\|\Gamma\|_{\mathrm{op}} = \gamma$. In the proof of Theorem C.3, the only place where $(\mathbf{S}_d)$ is invoked is to control the linear push-forward by $\Gamma$; in the isotropic case $\Gamma = \gamma I_d$ this coincides with (40) (with $s = \gamma$), and substituting $\bar{L} = \gamma$ yields

$$\bar{\Delta}\big(T^\pi \eta_1,\, T^\pi \eta_2\big) \leq c(\gamma)\, \bar{\Delta}\big(\eta_1, \eta_2\big).$$

$\square$

**Corollary C.2** (Worst-case tightness with explicit norm bound). *Fix $\bar{L} \in (0,1)$. Let $\Delta$ be a metric on $\mathcal{P}(\mathbb{R}^d)$ and $\bar{\Delta}(\eta_1, \eta_2) := \sup_{(s,a)} \Delta\big(\eta_1(s,a),\, \eta_2(s,a)\big)$. Assume $\Delta$ satisfies* (T). *If there exist $A$ and $\mu, \nu$ with $\|A\|_{\mathrm{op}} \leq \bar{L}$ such that*

$$\Delta\big(A_\#\mu,\, A_\#\nu\big) \; > \; \Delta(\mu,\nu), \tag{$\star$}$$

*then there exists an MDP/policy pair of the form in Theorem* C.3 *for which the linear maps satisfy the uniform bound*

$$\sup_{(s,a)} \sup_C \|A_{s,a}(C)\|_{\mathrm{op}} \;=\; \|A\|_{\mathrm{op}} \;=:\; \bar{L}' \;\leq\; \bar{L},$$

*and whose distributional Bellman operator $T^\pi$ is not nonexpansive under $\bar{\Delta}$. In particular, a contraction guarantee intended to hold uniformly over all such Bellman updates (with $\|A_{s,a}(C)\|_{\mathrm{op}} \leq \bar{L}$) requires $\Delta$ to be nonexpansive under every push-forward $A_\#$ with $\|A\|_{\mathrm{op}} \leq \bar{L}$.*

*Proof.* Consider the one-state one-action MDP $(\mathcal{S}, \mathcal{A}) = \{(s,a)\}$. Let $C$ be deterministic and set $g(s, a; C) = (s, a)$. Define $b_{s,a}(C) \equiv 0$ and $A_{s,a}(C) \equiv A$. Then $\sup_{(s,a)} \sup_C \|A_{s,a}(C)\|_{\mathrm{op}} = \|A\|_{\mathrm{op}} =: \bar{L}' \leq \bar{L}$. For any $\eta$ we have

$$(T^\pi \eta)(s, a) = A_\# \eta(s, a).$$

Take $\eta_1(s,a) = \mu$ and $\eta_2(s,a) = \nu$. Since there is only one index,

$$\bar{\Delta}(\eta_1, \eta_2) = \Delta(\mu, \nu), \qquad \bar{\Delta}(T^\pi \eta_1, T^\pi \eta_2) = \Delta(A_\# \mu, A_\# \nu).$$

Using $(\star)$ yields $\bar{\Delta}(T^\pi \eta_1, T^\pi \eta_2) > \bar{\Delta}(\eta_1, \eta_2)$, so $T^\pi$ is not nonexpansive. $\square$

### C.3. Uniform slicing

**Lemma C.7** (Uniform sliced scale contraction)**.** *Let $\Delta$ be a divergence on $\mathcal{P}(\mathbb{R})$. Assume that for all $\alpha, \beta \in \mathcal{P}(\mathbb{R})$ the following holds:*

(S) ***Scale contraction:*** *there exists a nondecreasing $c : [0,1] \to [0,\infty)$ such that for every $s \in [0,1]$,*

$$\Delta\big((x \mapsto sx)_\#\alpha, \, (x \mapsto sx)_\#\beta\big) \leq c(s)\,\Delta(\alpha, \beta),$$

*with $c(s) \leq 1$ for all $s \in [0,1]$ and $c(s) < 1$ for all $s \in [0,1)$.*

*For $\sigma$ a rotation-invariant probability measure on $\mathbb{S}^{d-1}$ and $q \in [1,\infty)$, define the sliced lift*

$$\mathbf{S}\Delta_q(\mu, \nu) := \Big( \int_{\mathbb{S}^{d-1}} \Delta\big((P_\theta)_\#\mu, (P_\theta)_\#\nu\big)^q \, d\sigma(\theta) \Big)^{1/q}, \quad P_\theta(x) = \langle \theta, x \rangle.$$

*Then $\mathbf{S}\Delta_q$ satisfies $(\mathbf{S}_d^{\mathrm{iso}})$: for every $s \in [0,1]$ and all $\mu, \nu \in \mathcal{P}(\mathbb{R}^d)$,*

$$\boxed{\mathbf{S}\Delta_q\big((x \mapsto sx)_\#\mu, \, (x \mapsto sx)_\#\nu\big) \;\leq\; c(s)\,\mathbf{S}\Delta_q(\mu, \nu).}$$

*In particular, if $s \in [0,1)$ then $(x \mapsto sx)_\#$ is a contraction on $(\mathcal{P}(\mathbb{R}^d), \mathbf{S}\Delta_q)$ with factor $c(s) < 1$.*

*Proof.* Fix $\theta \in \mathbb{S}^{d-1}$ and let $S_s(x) := sx$ on $\mathbb{R}^d$. For any $X \sim \mu$,

$$P_\theta(S_s X) = \langle \theta, sX \rangle = s\,\langle \theta, X \rangle,$$

hence

$$(P_\theta)_\#(S_s)_\#\mu \;=\; (x \mapsto sx)_\#(P_\theta)_\#\mu, \qquad (P_\theta)_\#(S_s)_\#\nu \;=\; (x \mapsto sx)_\#(P_\theta)_\#\nu.$$

Applying (S) to the one-dimensional laws $(P_\theta)_\#\mu$ and $(P_\theta)_\#\nu$ gives

$$\Delta\big((P_\theta)_\#(S_s)_\#\mu, \, (P_\theta)_\#(S_s)_\#\nu\big) \leq c(s)\,\Delta\big((P_\theta)_\#\mu, \, (P_\theta)_\#\nu\big).$$

Raise to the $q$-th power and integrate over $\theta \sim \sigma$:

$$\int_{\mathbb{S}^{d-1}} \Delta\big((P_\theta)_\#(S_s)_\#\mu, \, (P_\theta)_\#(S_s)_\#\nu\big)^q \, d\sigma(\theta) \;\leq\; c(s)^q \int_{\mathbb{S}^{d-1}} \Delta\big((P_\theta)_\#\mu, \, (P_\theta)_\#\nu\big)^q \, d\sigma(\theta).$$

Taking the $q$-th root yields $\mathbf{S}\Delta_q\big((S_s)_\#\mu, (S_s)_\#\nu\big) \leq c(s)\,\mathbf{S}\Delta_q(\mu, \nu)$, which is exactly $(\mathbf{S}_d^{\mathrm{iso}})$. $\qquad\square$

**Lemma C.8** (Directional slice scaling under a linear map)**.** *Let $\Delta$ be a divergence on $\mathcal{P}(\mathbb{R})$. Assume that there exists $\alpha > 0$ such that for all $\alpha_1, \alpha_2 \in \mathcal{P}(\mathbb{R})$ and all $s \in [0,1]$,*

$$\Delta\big((x \mapsto sx)_\#\alpha_1, \, (x \mapsto sx)_\#\alpha_2\big) \;\leq\; s^\alpha\,\Delta(\alpha_1, \alpha_2). \tag{41}$$

*Let $d \geq 1$ and let $A \in \mathbb{R}^{d \times d}$ satisfy $\|A\|_{\mathrm{op}} \leq 1$. For $\theta \in \mathbb{S}^{d-1}$ define*

$$r_A(\theta) := \|A^\top \theta\|_2 \in [0,1], \qquad \theta_A := \begin{cases} A^\top \theta / \|A^\top \theta\|_2, & r_A(\theta) > 0, \\ \text{any element of } \mathbb{S}^{d-1}, & r_A(\theta) = 0. \end{cases}$$

*Then for all $\mu, \nu \in \mathcal{P}(\mathbb{R}^d)$ and all $\theta \in \mathbb{S}^{d-1}$,*

$$\boxed{\Delta\big((P_\theta)_\# A_\#\mu, \, (P_\theta)_\# A_\#\nu\big) \;\leq\; r_A(\theta)^\alpha\,\Delta\big((P_{\theta_A})_\#\mu, \, (P_{\theta_A})_\#\nu\big),} \qquad P_\theta(x) = \langle \theta, x \rangle.$$

*In particular, for any probability measure $\sigma$ on $\mathbb{S}^{d-1}$ and any $q \in [1,\infty)$,*

$$\mathbf{S}\Delta_q(A_\#\mu, A_\#\nu)^q \;\leq\; \int_{\mathbb{S}^{d-1}} r_A(\theta)^{\alpha q}\,\Delta\big((P_{\theta_A})_\#\mu, \, (P_{\theta_A})_\#\nu\big)^q \, d\sigma(\theta).$$

*Proof.* Fix $\theta \in \mathbb{S}^{d-1}$. For any $x \in \mathbb{R}^d$ we have

$$P_\theta(Ax) = \langle \theta, Ax \rangle = \langle A^\top \theta, x \rangle.$$

If $r_A(\theta) = \|A^\top \theta\|_2 > 0$, then $A^\top \theta = r_A(\theta)\, \theta_A$ and therefore

$$P_\theta(Ax) = r_A(\theta)\, \langle \theta_A, x \rangle = \big(x \mapsto r_A(\theta)\, x\big)\big(P_{\theta_A}(x)\big).$$

Taking push-forwards gives the commutation identity

$$(P_\theta)_\# A_\# \mu = \big(x \mapsto r_A(\theta)\, x\big)_\# (P_{\theta_A})_\# \mu, \qquad (P_\theta)_\# A_\# \nu = \big(x \mapsto r_A(\theta)\, x\big)_\# (P_{\theta_A})_\# \nu.$$

Applying the one-dimensional scale bound (41) with $s = r_A(\theta) \in (0,1]$ yields

$$\Delta\big((P_\theta)_\# A_\# \mu,\ (P_\theta)_\# A_\# \nu\big) \leq r_A(\theta)^\alpha\, \Delta\big((P_{\theta_A})_\# \mu,\ (P_{\theta_A})_\# \nu\big).$$

If instead $r_A(\theta) = 0$, then $A^\top \theta = 0$ and hence $P_\theta(Ax) = 0$ for all $x$. Thus $(P_\theta)_\# A_\# \mu = (P_\theta)_\# A_\# \nu = \delta_0$ and the left-hand side is 0, so the displayed inequality holds as well.

Finally, raise the pointwise bound to the $q$-th power and integrate with respect to $\sigma$:

$$\int_{\mathbb{S}^{d-1}} \Delta\big((P_\theta)_\# A_\# \mu,\ (P_\theta)_\# A_\# \nu\big)^q d\sigma(\theta) \ \leq\ \int_{\mathbb{S}^{d-1}} r_A(\theta)^{\alpha q}\, \Delta\big((P_{\theta_A})_\# \mu,\ (P_{\theta_A})_\# \nu\big)^q d\sigma(\theta).$$

By definition of the sliced lift,

$$\mathbf{S}\Delta_q(A_\# \mu, A_\# \nu)^q = \int_{\mathbb{S}^{d-1}} \Delta\big((P_\theta)_\# A_\# \mu,\ (P_\theta)_\# A_\# \nu\big)^q d\sigma(\theta),$$

which yields the stated integral inequality.

$\square$

**Lemma C.9** (Mixture $p$-convexity lifts to the sliced divergence)**.** *Let $\Delta$ be a divergence on $\mathcal{P}(\mathbb{R})$ satisfying mixture $p$-convexity: for every probability space $(\Omega, \mathcal{F}, \rho)$ and measurable families $(\mu_c)_{c \in \Omega}, (\nu_c)_{c \in \Omega} \subset \mathcal{P}(\mathbb{R})$,*

$$\Delta\left(\int_\Omega \mu_c\, \rho(dc),\ \int_\Omega \nu_c\, \rho(dc)\right) \leq \left(\int_\Omega \Delta(\mu_c, \nu_c)^p\, \rho(dc)\right)^{1/p}, \quad p \in [1, \infty).$$

*Fix any probability measure $\sigma$ on $\mathbb{S}^{d-1}$. Define the sliced lift for $\mu, \nu \in \mathcal{P}(\mathbb{R}^d)$ by*

$$\mathbf{S}\Delta_p(\mu, \nu) := \left(\int_{\mathbb{S}^{d-1}} \Delta\big((P_\theta)_\# \mu, (P_\theta)_\# \nu\big)^p\, \sigma(d\theta)\right)^{1/p}, \qquad P_\theta(x) = \langle \theta, x \rangle.$$

*Then $\mathbf{S}\Delta_p$ is mixture $p$-convex on $\mathcal{P}(\mathbb{R}^d)$, i.e., for any measurable families $(\mu_c)_{c \in \Omega}, (\nu_c)_{c \in \Omega} \subset \mathcal{P}(\mathbb{R}^d)$,*

$$\boxed{\ \mathbf{S}\Delta_p\left(\int_\Omega \mu_c\, \rho(dc),\ \int_\Omega \nu_c\, \rho(dc)\right) \ \leq\ \left(\int_\Omega \mathbf{S}\Delta_p(\mu_c, \nu_c)^p\, \rho(dc)\right)^{1/p}.\ }$$

*Proof.* Fix $\theta \in \mathbb{S}^{d-1}$ and set $\mu_c^\theta := (P_\theta)_\# \mu_c,\ \nu_c^\theta := (P_\theta)_\# \nu_c \in \mathcal{P}(\mathbb{R})$. By linearity of pushforward w.r.t. mixtures,

$$(P_\theta)_\# \left(\int_\Omega \mu_c\, \rho(dc)\right) = \int_\Omega \mu_c^\theta\, \rho(dc), \qquad (P_\theta)_\# \left(\int_\Omega \nu_c\, \rho(dc)\right) = \int_\Omega \nu_c^\theta\, \rho(dc).$$

Applying mixture $p$-convexity of $\Delta$ in 1-D at this fixed $\theta$,

$$\Delta\left(\int_\Omega \mu_c^\theta\, \rho(dc),\ \int_\Omega \nu_c^\theta\, \rho(dc)\right) \ \leq\ \left(\int_\Omega \Delta(\mu_c^\theta, \nu_c^\theta)^p\, \rho(dc)\right)^{1/p}.$$

Raise to the $p$th power and integrate over $\theta \sim \sigma$; Tonelli/Fubini yields

$$\int_{\mathbb{S}^{d-1}} \Delta\big((P_\theta)_\# \textstyle\int \mu_c \, d\rho, \ (P_\theta)_\# \textstyle\int \nu_c \, d\rho\big)^p \, \sigma(d\theta)$$

$$\leq \int_\Omega \bigg( \int_{\mathbb{S}^{d-1}} \Delta\big((P_\theta)_\# \mu_c, \ (P_\theta)_\# \nu_c\big)^p \, \sigma(d\theta) \bigg) \rho(dc).$$

By the very definition of the sliced divergence,

$$\int_{\mathbb{S}^{d-1}} \Delta\big((P_\theta)_\# \textstyle\int \mu_c \, d\rho, \ (P_\theta)_\# \textstyle\int \nu_c \, d\rho\big)^p \, \sigma(d\theta) = \mathbf{S}\Delta_p\big( \textstyle\int \mu_c \, d\rho, \ \int \nu_c \, d\rho\big)^p$$

$$\leq \int_\Omega \bigg( \int_{\mathbb{S}^{d-1}} \Delta\big((P_\theta)_\# \mu_c, \ (P_\theta)_\# \nu_c\big)^p \, \sigma(d\theta) \bigg) \rho(dc)$$

$$= \int_\Omega \mathbf{S}\Delta_p(\mu_c, \nu_c)^p \, \rho(dc).$$

Taking the $p$th root gives

$$\mathbf{S}\Delta_p\bigg( \int_\Omega \mu_c \, \rho(dc), \ \int_\Omega \nu_c \, \rho(dc) \bigg) \ \leq \ \bigg( \int_\Omega \mathbf{S}\Delta_p(\mu_c, \nu_c)^p \, \rho(dc) \bigg)^{1/p}.$$

$\square$

**Theorem C.4** (Supremum–uniform–sliced contraction of the multivariate distributional Bellman operator (fixed isotropic discount)). *Let $\Delta$ be a divergence on $\mathcal{P}(\mathbb{R})$ and fix $p \in [1, \infty)$. For $\sigma$ a rotation-invariant probability measure on $\mathbb{S}^{d-1}$, define the uniform sliced lift*

$$\mathbf{S}\Delta_p(\mu, \nu) := \Big( \int_{\mathbb{S}^{d-1}} \Delta\big((P_\theta)_\# \mu, \ (P_\theta)_\# \nu\big)^p \, d\sigma(\theta) \Big)^{1/p}, \quad P_\theta(x) = \langle \theta, x \rangle.$$

*Assume that for all $\alpha, \beta \in \mathcal{P}(\mathbb{R})$ the following hold:*

(T) ***Translation nonexpansion:*** *for every $t \in \mathbb{R}$,*

$$\Delta\big((x \mapsto x + t)_\# \alpha, \ (x \mapsto x + t)_\# \beta\big) \leq \Delta(\alpha, \beta).$$

(S) ***Scale contraction:*** *there exists a nondecreasing $c : [0, 1] \to [0, 1]$ such that for every $s \in [0, 1]$,*

$$\Delta\big((x \mapsto sx)_\# \alpha, \ (x \mapsto sx)_\# \beta\big) \leq c(s) \, \Delta(\alpha, \beta),$$

*with $c(s) < 1$ for all $s \in [0, 1)$.*

($\mathbf{M}_p$) ***Mixture $p$-convexity:*** *for every probability space $(\Omega, \mathcal{F}, \rho)$ and measurable families $(\mu_c), (\nu_c) \subset \mathcal{P}(\mathbb{R})$,*

$$\Delta\Big( \int_\Omega \mu_c \, \rho(dc), \ \int_\Omega \nu_c \, \rho(dc) \Big) \leq \Big( \int_\Omega \Delta(\mu_c, \nu_c)^p \, \rho(dc) \Big)^{1/p}.$$

*Define the supremum metric over state–action indices by*

$$\overline{\mathbf{S}\Delta_p}(\eta_1, \eta_2) := \sup_{(s, a)} \mathbf{S}\Delta_p\big(\eta_1(s, a), \eta_2(s, a)\big), \quad \eta_i : \mathcal{S} \times \mathcal{A} \to \mathcal{P}(\mathbb{R}^d).$$

***Bellman update (fixed isotropic discount).*** *Fix $\gamma \in [0, 1]$ and set $\Gamma := \gamma I_d$. For each $(s, a)$, gather all environment/policy randomness into a single random element $C$ determining $(S', A') := g(s, a; C)$ and a measurable translation $b_{s,a}(C) \in \mathbb{R}^d$. Conditioned on $C$, draw*

$$X' \mid C \sim \eta(S', A'),$$

*and define*

$$(T^\pi \eta)(s, a) := \mathrm{Law}\big(b_{s,a}(C) + \Gamma X'\big).$$

*Then, for all $\eta_1, \eta_2$,*

$$\boxed{\overline{\mathbf{S}\Delta_p}\big(T^\pi \eta_1, \ T^\pi \eta_2\big) \ \leq \ c(\gamma)\, \overline{\mathbf{S}\Delta_p}\big(\eta_1, \eta_2\big).}$$

*Consequently, if $\gamma < 1$, the operator $T^\pi$ is a contraction on $(\mathcal{S} \times \mathcal{A} \to \mathcal{P}(\mathbb{R}^d), \overline{\mathbf{S}\Delta_p})$ with factor $c(\gamma) < 1$.*

*Proof.* We verify that the multivariate divergence $\mathbf{S}\Delta_p$ satisfies the assumptions of Corollary C.1.

**Translation nonexpansion lifts.** Fix $t \in \mathbb{R}^d$ and define $T_t(x) = x + t$ on $\mathbb{R}^d$. For every $\theta \in \mathbb{S}^{d-1}$ and any $\mu, \nu \in \mathcal{P}(\mathbb{R}^d)$,

$$(P_\theta)_\# (T_t)_\# \mu = (x \mapsto x + \langle \theta, t \rangle)_\# (P_\theta)_\# \mu, \qquad (P_\theta)_\# (T_t)_\# \nu = (x \mapsto x + \langle \theta, t \rangle)_\# (P_\theta)_\# \nu.$$

Applying **(T)** to the one-dimensional laws $(P_\theta)_\# \mu$ and $(P_\theta)_\# \nu$ yields

$$\Delta\big((P_\theta)_\# (T_t)_\# \mu, \ (P_\theta)_\# (T_t)_\# \nu\big) \leq \Delta\big((P_\theta)_\# \mu, \ (P_\theta)_\# \nu\big).$$

Raising to the $p$-th power, integrating over $\theta \sim \sigma$, and taking the $p$-th root gives

$$\mathbf{S}\Delta_p\big((T_t)_\# \mu, (T_t)_\# \nu\big) \leq \mathbf{S}\Delta_p(\mu, \nu),$$

which is **(T)** for $\mathbf{S}\Delta_p$ on $\mathcal{P}(\mathbb{R}^d)$.

**Scale contraction lifts to $(\mathbf{S}_d^{\mathrm{iso}})$.** By Lemma C.7, **(S)** for $\Delta$ implies that $\mathbf{S}\Delta_p$ satisfies $(\mathbf{S}_d^{\mathrm{iso}})$ on $\mathcal{P}(\mathbb{R}^d)$ with the same function $c$: for all $s \in [0, 1]$,

$$\mathbf{S}\Delta_p\big((x \mapsto sx)_\# \mu, \ (x \mapsto sx)_\# \nu\big) \ \leq \ c(s)\, \mathbf{S}\Delta_p(\mu, \nu).$$

**Mixture $p$-convexity lifts.** By Lemma C.9, mixture $p$-convexity $(\mathbf{M}_p)$ of $\Delta$ lifts to mixture $p$-convexity of $\mathbf{S}\Delta_p$ on $\mathcal{P}(\mathbb{R}^d)$.

Having established **(T)**, $(\mathbf{S}_d^{\mathrm{iso}})$, and $(\mathbf{M}_p)$ for the multivariate divergence $\mathbf{S}\Delta_p$, we may apply Corollary C.1 with $\Delta$ replaced by $\mathbf{S}\Delta_p$ and $\Gamma = \gamma I_d$, yielding

$$\overline{\mathbf{S}\Delta_p}\big(T^\pi \eta_1, \ T^\pi \eta_2\big) \leq c(\gamma)\, \overline{\mathbf{S}\Delta_p}\big(\eta_1, \eta_2\big).$$

$\square$

**C.4. Negative results.**

Our goal in this section is to test whether multivariate contraction can be controlled solely through the operator norm of the linear part of the Bellman update. Corollary C.2 gives a simple reduction: it suffices to find one matrix $A$ and one pair of laws $\mu, \nu$ such that

$$\|A\|_{\mathrm{op}} \leq \bar{L} \qquad \text{and} \qquad \Delta\big(A_{\#}\mu, \, A_{\#}\nu\big) > \Delta(\mu, \nu). \tag{$\star$}$$

If such a triple exists, then there is an MDP/policy pair whose Bellman update respects the same norm bound $\|A_{s,a}(C)\|_{\mathrm{op}} \leq \bar{L}$ but for which the associated distributional Bellman operator is not nonexpansive under the supremum metric $\bar{\Delta}$. In other words, $(\star)$ rules out any contraction guarantee that is meant to hold uniformly over all such updates and that depends only on $\|A\|_{\mathrm{op}}$.

Accordingly, we now construct explicit instances of $(\star)$ for the two kernel families that have been instantiated to obtain contraction-based MMD objectives in distributional RL: the negative-type power family (Nguyen-Tang et al., 2021) and the multiquadric family (Killingberg & Langseth, 2023), as detailed in Section C.4.1; we then apply the same obstruction strategy to *uniform sliced* objectives, obtained by averaging the corresponding one-dimensional discrepancies over directions $\theta \in \mathbb{S}^1$, as detailed in Section C.4.2.

C.4.1. MMD.

Throughout, we use the MMD convention of Sec. A.3. In particular, for the (possibly CPD) kernels considered below, $\mathbf{MMD}_k^2(P,Q)$ admits the expansion

$$\mathbf{MMD}_k^2(P,Q) = \underbrace{\mathbb{E}\, k(X,X')}_{\text{within } P} + \underbrace{\mathbb{E}\, k(Y,Y')}_{\text{within } Q} - 2 \underbrace{\mathbb{E}\, k(X,Y)}_{\text{cross}}, \qquad X, X' \sim P, \ Y, Y' \sim Q \text{ i.i.d.} \tag{42}$$

The examples below exploit a simple mechanism visible in (42): since the objective is a cross term minus two within terms, one can arrange for these expectations to be individually large and close, so that $\mathbf{MMD}_k^2(P,Q)$ results from a delicate cancellation. An anisotropic contraction can then shrink the within terms much more than the cross term, breaking that cancellation and yielding an *increase* of $\mathbf{MMD}_k^2$ despite the map having operator norm $< 1$.

Fix $s \in (0,1)$ and $\varepsilon \in (0,1)$ and consider the full-rank diagonal map on $\mathbb{R}^2$,

$$A_{s,\varepsilon} := \mathrm{diag}(s, \ s\varepsilon), \qquad \|A_{s,\varepsilon}\|_{\mathrm{op}} = s < 1.$$

Fix $D > 0$ and $M > 0$ and define

$$P_{D,M} := \tfrac{1}{2}\delta_{(0,M)} + \tfrac{1}{2}\delta_{(0,-M)}, \qquad Q_{D,M} := \tfrac{1}{2}\delta_{(D,M)} + \tfrac{1}{2}\delta_{(D,-M)}.$$

Under $A_{s,\varepsilon}$, the separation $D$ becomes $sD$ while the spread $M$ becomes $\varepsilon M$, which can be made tiny when $\varepsilon \ll 1$.

**Negative-type power family ($\beta = 1$, energy distance).** Consider the negative-type power kernel $k_\beta(x,y) := -\|x-y\|^\beta$ on $\mathbb{R}^2$. For $\beta = 1$, (42) becomes

$$\mathbf{MMD}_{k_1}^2(P,Q) = 2\,\mathbb{E}\|X - Y\| - \mathbb{E}\|X - X'\| - \mathbb{E}\|Y - Y'\|.$$

For $(P,Q) = (P_{D,M}, Q_{D,M})$, the two-value enumeration yields

$$\mathbf{MMD}_{k_1}^2(P_{D,M}, Q_{D,M}) = D + \sqrt{D^2 + 4M^2} - 2M, \tag{43}$$

and after applying $A_{s,\varepsilon}$,

$$\mathbf{MMD}_{k_1}^2\big((A_{s,\varepsilon})_{\#}P_{D,M}, \ (A_{s,\varepsilon})_{\#}Q_{D,M}\big) = sD + s\sqrt{D^2 + 4\varepsilon^2 M^2} - 2s\varepsilon M. \tag{44}$$

*Numerical instance.* Take $D = 1$, $M = 1000$, $s = 0.75$, $\varepsilon = 10^{-6}$ and write $A := A_{s,\varepsilon}$. Then $\|A\|_{\mathrm{op}} = 0.75 < 1$, while a direct evaluation of (43)–(44) gives

$$\mathbf{MMD}_{k_1}^2(P_{1,1000}, Q_{1,1000}) \approx 1.000249999984, \qquad \mathbf{MMD}_{k_1}^2(A_{\#}P_{1,1000}, A_{\#}Q_{1,1000}) \approx 1.498501499999.$$

**Negative multiquadric kernel.** Fix $h = 1$ and consider the negative multiquadric kernel

$$k_{\mathrm{MQ}}(x, y) := -\sqrt{1 + \|x - y\|^2}.$$

With $f(r) := \sqrt{1 + r^2}$, (42) becomes

$$\mathbf{MMD}^2_{k_{\mathrm{MQ}}}(P, Q) = 2\,\mathbb{E}\,f(\|X - Y\|) - \mathbb{E}\,f(\|X - X'\|) - \mathbb{E}\,f(\|Y - Y'\|).$$

For $(P, Q) = (P_{D,M}, Q_{D,M})$,

$$\mathbf{MMD}^2_{k_{\mathrm{MQ}}}(P_{D,M}, Q_{D,M}) = f(D) + f(\sqrt{D^2 + 4M^2}) - 1 - f(2M), \tag{45}$$

and after applying $A_{s,\varepsilon}$,

$$\mathbf{MMD}^2_{k_{\mathrm{MQ}}}\big((A_{s,\varepsilon})_\# P_{D,M},\ (A_{s,\varepsilon})_\# Q_{D,M}\big) = f(sD) + f\Big(s\sqrt{D^2 + 4\varepsilon^2 M^2}\Big) - 1 - f(2s\varepsilon M). \tag{46}$$

*Numerical instance.* With the same choice $D = 1$, $M = 1000$, $s = 0.75$, $\varepsilon = 10^{-6}$ (so $\|A\|_{\mathrm{op}} = 0.75 < 1$), a direct evaluation of (45)–(46) gives

$$\mathbf{MMD}^2_{k_{\mathrm{MQ}}}(P_{1,1000}, Q_{1,1000}) \approx 0.414463562326, \qquad \mathbf{MMD}^2_{k_{\mathrm{MQ}}}(A_\# P_{1,1000}, A_\# Q_{1,1000}) \approx 0.499999775000.$$

### C.4.2. UNIFORM SLICING.

Fix $d = 2$ and let $\sigma$ be the uniform probability measure on $\mathbb{S}^1$. Given a divergence $\Delta$ on $\mathcal{P}(\mathbb{R})$ and $p \in [1, \infty)$, define its uniform sliced lift by

$$\mathbf{S}\Delta_p(\mu, \nu) := \left( \int_{\mathbb{S}^1} \Delta\big((P_\theta)_\# \mu, (P_\theta)_\# \nu\big)^p \, d\sigma(\theta) \right)^{1/p}, \qquad P_\theta(x) := \langle \theta, x \rangle. \tag{47}$$

In this subsection we take $\Delta = \mathbf{W}_1$ and $p = 1$, so that

$$\mathbf{SW}_1(\mu, \nu) = \int_{\mathbb{S}^1} \mathbf{W}_1\big((P_\theta)_\# \mu, (P_\theta)_\# \nu\big) \, d\sigma(\theta).$$

**Wasserstein ($\mathbf{W}_1$).** Let

$$\mu := \tfrac{1}{2}\delta_{(1,0)} + \tfrac{1}{2}\delta_{(-1,0)}, \qquad \nu := \tfrac{1}{2}\delta_{(0,1)} + \tfrac{1}{2}\delta_{(0,-1)}.$$

Fix $s \in (0, 1)$ and $\varepsilon \in (0, 1)$ and consider the full–rank diagonal map

$$A := \mathrm{diag}(s,\ s\varepsilon), \qquad \|A\|_{\mathrm{op}} = s < 1.$$

Parametrize $\theta \in \mathbb{S}^1$ as $\theta = (\cos \varphi, \sin \varphi)$ with $\varphi \in [0, 2\pi)$. Then the one-dimensional projections are

$$(P_\theta)_\# \mu = \tfrac{1}{2}\delta_{-\cos\varphi} + \tfrac{1}{2}\delta_{\cos\varphi}, \qquad (P_\theta)_\# \nu = \tfrac{1}{2}\delta_{-\sin\varphi} + \tfrac{1}{2}\delta_{\sin\varphi},$$

and similarly,

$$(P_\theta)_\#(A_\# \mu) = \tfrac{1}{2}\delta_{-s\cos\varphi} + \tfrac{1}{2}\delta_{s\cos\varphi}, \qquad (P_\theta)_\#(A_\# \nu) = \tfrac{1}{2}\delta_{-s\varepsilon\sin\varphi} + \tfrac{1}{2}\delta_{s\varepsilon\sin\varphi}.$$

In one dimension, for $a, b \geq 0$,

$$\mathbf{W}_1\big(\tfrac{1}{2}\delta_{-a} + \tfrac{1}{2}\delta_a,\ \tfrac{1}{2}\delta_{-b} + \tfrac{1}{2}\delta_b\big) = |a - b|.$$

Therefore, using (47) with $p = 1$ and $d\sigma(\theta) = \frac{1}{2\pi}\, d\varphi$, we obtain

$$\mathbf{SW}_1(\mu, \nu) = \frac{1}{2\pi} \int_0^{2\pi} \big| |\cos\varphi| - |\sin\varphi| \big| \, d\varphi, \tag{48}$$

$$\mathbf{SW}_1(A_\# \mu, A_\# \nu) = \frac{s}{2\pi} \int_0^{2\pi} \big| |\cos\varphi| - \varepsilon |\sin\varphi| \big| \, d\varphi. \tag{49}$$

We use the following identity: for any integrable $H : [0,1]^2 \to \mathbb{R}$,

$$\int_0^{2\pi} H\big(|\cos\varphi|, |\sin\varphi|\big)\, d\varphi = 4 \int_0^{\pi/2} H\big(\cos\varphi, \sin\varphi\big)\, d\varphi. \tag{50}$$

Applying (50) to (48)–(49) (and using that $\cos\varphi, \sin\varphi \geq 0$ on $[0, \pi/2]$) yields

$$\mathbf{SW}_1(\mu, \nu) = \frac{2}{\pi} \int_0^{\pi/2} |\cos\varphi - \sin\varphi|\, d\varphi, \tag{51}$$

$$\mathbf{SW}_1(A_\#\mu, A_\#\nu) = \frac{2s}{\pi} \int_0^{\pi/2} |\cos\varphi - \varepsilon \sin\varphi|\, d\varphi. \tag{52}$$

We evaluate the one-dimensional integrals in (51)–(52) using `scipy.integrate.quad` (Virtanen et al., 2020). For $s = 0.9$ and $\varepsilon = 0.01$ this yields $\|A\|_{\mathrm{op}} = 0.9 < 1$ but

$$\mathbf{SW}_1(\mu, \nu) \approx 0.527393, \qquad \mathbf{SW}_1(A_\#\mu, A_\#\nu) \approx 0.567286,$$

so $\mathbf{SW}_1(A_\#\mu, A_\#\nu) > \mathbf{SW}_1(\mu, \nu)$, which instantiates ($\star$) with $\Delta = \mathbf{SW}_1$.

**Cramér–2 ($\ell_2^2$).** The same construction yields a counterexample for the uniform sliced Cramér–2 divergence. For $a, b \geq 0$, a direct computation gives

$$\ell_2^2\big(\tfrac{1}{2}\delta_{-a} + \tfrac{1}{2}\delta_a,\ \tfrac{1}{2}\delta_{-b} + \tfrac{1}{2}\delta_b\big) = \frac{1}{2}|a - b|.$$

In one dimension we also have

$$\mathbf{W}_1\big(\tfrac{1}{2}\delta_{-a} + \tfrac{1}{2}\delta_a,\ \tfrac{1}{2}\delta_{-b} + \tfrac{1}{2}\delta_b\big) = |a - b|,$$

hence, for every $\theta$,

$$\ell_2^2\big((P_\theta)_\#\mu, (P_\theta)_\#\nu\big) = \tfrac{1}{2}\mathbf{W}_1\big((P_\theta)_\#\mu, (P_\theta)_\#\nu\big), \qquad \ell_2^2\big((P_\theta)_\# A_\#\mu, (P_\theta)_\# A_\#\nu\big) = \tfrac{1}{2}\mathbf{W}_1\big((P_\theta)_\# A_\#\mu, (P_\theta)_\# A_\#\nu\big).$$

Using the definition (47) with $p = 1$ and linearity of the integral, we obtain the exact identities

$$\mathbf{S}\ell_2^2(\mu, \nu) = \tfrac{1}{2}\mathbf{SW}_1(\mu, \nu), \qquad \mathbf{S}\ell_2^2(A_\#\mu, A_\#\nu) = \tfrac{1}{2}\mathbf{SW}_1(A_\#\mu, A_\#\nu). \tag{53}$$

In particular, for $s = 0.9$ and $\varepsilon = 0.01$ (so $\|A\|_{\mathrm{op}} = 0.9 < 1$), using the values reported in the Wasserstein case we get

$$\mathbf{S}\ell_2^2(\mu, \nu) \approx \tfrac{1}{2} \cdot 0.527393 = 0.263697, \qquad \mathbf{S}\ell_2^2(A_\#\mu, A_\#\nu) \approx \tfrac{1}{2} \cdot 0.567286 = 0.283643,$$

so $\mathbf{S}\ell_2^2(A_\#\mu, A_\#\nu) > \mathbf{S}\ell_2^2(\mu, \nu)$, which instantiates ($\star$) with $\Delta = \mathbf{S}\ell_2^2$.

**Uniform sliced MMD with the negative-type power family.** Fix $\beta \in (0, 2]$ and consider the negative-type power kernel on $\mathbb{R}$,

$$k_\beta(x, y) := -|x - y|^\beta.$$

For one-dimensional laws $\mu, \nu \in \mathcal{P}(\mathbb{R})$,

$$\mathbf{MMD}_{k_\beta}^2(\mu, \nu) = \underbrace{2\,\mathbb{E}|X - Y|^\beta}_{\text{cross}} - \underbrace{\mathbb{E}|X - X'|^\beta}_{\text{within } \mu} - \underbrace{\mathbb{E}|Y - Y'|^\beta}_{\text{within } \nu}, \qquad X, X' \sim \mu,\ Y, Y' \sim \nu \text{ i.i.d.}$$

For the two-point symmetric laws $\mu = \tfrac{1}{2}\delta_{+a} + \tfrac{1}{2}\delta_{-a}$ and $\nu = \tfrac{1}{2}\delta_{+b} + \tfrac{1}{2}\delta_{-b}$ with $a, b \geq 0$, we have

$$\mathbb{E}|X - Y|^\beta = \tfrac{1}{2}|a - b|^\beta + \tfrac{1}{2}(a + b)^\beta, \qquad \mathbb{E}|X - X'|^\beta = 2^{\beta-1}a^\beta, \qquad \mathbb{E}|Y - Y'|^\beta = 2^{\beta-1}b^\beta,$$

and substituting into the definition yields

$$\mathbf{MMD}_{k_\beta}^2(\mu, \nu) = |a - b|^\beta + (a + b)^\beta - 2^{\beta-1}\big(a^\beta + b^\beta\big). \tag{54}$$

Now consider the same pair of laws $\mu, \nu \in \mathcal{P}(\mathbb{R}^2)$ and the full-rank anisotropic map $A_{s,\varepsilon} = \operatorname{diag}(s, s\varepsilon)$ as above. For $\theta = (\cos\varphi, \sin\varphi) \in \mathbb{S}^1$ with $\varphi \in [0, 2\pi)$, the projected laws are

$$(P_\theta)_{\#}\mu = \tfrac{1}{2}\delta_{+|\cos\varphi|} + \tfrac{1}{2}\delta_{-|\cos\varphi|}, \qquad (P_\theta)_{\#}\nu = \tfrac{1}{2}\delta_{+|\sin\varphi|} + \tfrac{1}{2}\delta_{-|\sin\varphi|},$$

and after applying $A_{s,\varepsilon}$,

$$(P_\theta)_{\#}(A_{s,\varepsilon})_{\#}\mu = \tfrac{1}{2}\delta_{+s|\cos\varphi|} + \tfrac{1}{2}\delta_{-s|\cos\varphi|}, \qquad (P_\theta)_{\#}(A_{s,\varepsilon})_{\#}\nu = \tfrac{1}{2}\delta_{+s\varepsilon|\sin\varphi|} + \tfrac{1}{2}\delta_{-s\varepsilon|\sin\varphi|}.$$

Define the uniform sliced lift (with $p = 1$ and $\sigma$ uniform on $\mathbb{S}^1$) by

$$\mathbf{SMMD}^2_{k_\beta}(\mu, \nu) := \int_{\mathbb{S}^1} \mathbf{MMD}^2_{k_\beta}\big((P_\theta)_{\#}\mu, (P_\theta)_{\#}\nu\big)\, d\sigma(\theta).$$

Using the identity (50) (as in the Wasserstein paragraph) to reduce the $\varphi$-integral to $[0, \pi/2]$, and then applying (54) with $a = \cos\varphi$ and $b = \sin\varphi$, this becomes

$$\mathbf{SMMD}^2_{k_\beta}(\mu, \nu) = \frac{2}{\pi}\int_0^{\pi/2} \Big( |\cos\varphi - \sin\varphi|^\beta + (\cos\varphi + \sin\varphi)^\beta$$
$$- 2^{\beta-1}\big(\cos^\beta\varphi + \sin^\beta\varphi\big) \Big)\, d\varphi, \tag{55}$$

$$\mathbf{SMMD}^2_{k_\beta}\big((A_{s,\varepsilon})_{\#}\mu, (A_{s,\varepsilon})_{\#}\nu\big) = \frac{2}{\pi}\int_0^{\pi/2} \Big( |s\cos\varphi - s\varepsilon\sin\varphi|^\beta + (s\cos\varphi + s\varepsilon\sin\varphi)^\beta$$
$$- 2^{\beta-1}\big((s\cos\varphi)^\beta + (s\varepsilon\sin\varphi)^\beta\big) \Big)\, d\varphi. \tag{56}$$

*Concrete numbers (computed by the same numerical integration method as earlier).* We use $\varepsilon = 0.01$ throughout, with $s = 0.9$ for $\beta \in \{0.5, 1\}$ and $s = 0.98$ for $\beta = 1.5$. When $\beta = 1$, (54) simplifies to $\mathbf{MMD}^2_{k_1}(\tfrac{1}{2}\delta_{\pm a}, \tfrac{1}{2}\delta_{\pm b}) = |a - b|$, so the sliced objective coincides with the uniform sliced Wasserstein-1 objective from the previous paragraph; numerically this gives

$$\mathbf{SMMD}^2_{k_1}(\mu, \nu) \approx 0.52739, \qquad \mathbf{SMMD}^2_{k_1}\big((A_{0.9,0.01})_{\#}\mu, (A_{0.9,0.01})_{\#}\nu\big) \approx 0.56729, \qquad \text{ratio} \approx 1.07564.$$

For $\beta = 0.5$,

$$\mathbf{SMMD}^2_{k_{0.5}}(\mu, \nu) \approx 0.73545, \qquad \mathbf{SMMD}^2_{k_{0.5}}\big((A_{0.9,0.01})_{\#}\mu, (A_{0.9,0.01})_{\#}\nu\big) \approx 0.88443, \qquad \text{ratio} \approx 1.20256.$$

For $\beta = 1.5$,

$$\mathbf{SMMD}^2_{k_{1.5}}(\mu, \nu) \approx 0.29777, \qquad \mathbf{SMMD}^2_{k_{1.5}}\big((A_{0.98,0.01})_{\#}\mu, (A_{0.98,0.01})_{\#}\nu\big) \approx 0.31553, \qquad \text{ratio} \approx 1.05964.$$

Thus, for several $\beta \in (0, 2]$, the uniform sliced objective $\mathbf{SMMD}^2_{k_\beta}$ can *increase* under a full-rank anisotropic $A_{s,\varepsilon}$ with $\|A_{s,\varepsilon}\|_{\mathrm{op}} = s < 1$.

**Uniform sliced MMD with the multiquadric kernel.** Fix $h > 0$ and consider the (negative) multiquadric kernel on $\mathbb{R}$,

$$k_h(x, y) := -\sqrt{1 + h^2(x - y)^2}.$$

For one-dimensional laws $\mu, \nu \in \mathcal{P}(\mathbb{R})$, with $X, X' \sim \mu$ i.i.d. and $Y, Y' \sim \nu$ i.i.d.,

$$\mathbf{MMD}^2_{k_h}(\mu, \nu) = \underbrace{2\, \mathbb{E}\sqrt{1 + h^2(X - Y)^2}}_{\text{cross}} - \underbrace{\mathbb{E}\sqrt{1 + h^2(X - X')^2}}_{\text{within } \mu} - \underbrace{\mathbb{E}\sqrt{1 + h^2(Y - Y')^2}}_{\text{within } \nu}. \tag{57}$$

For the two-point symmetric laws $\mu = \tfrac{1}{2}\delta_{+a} + \tfrac{1}{2}\delta_{-a}$ and $\nu = \tfrac{1}{2}\delta_{+b} + \tfrac{1}{2}\delta_{-b}$ with $a, b \geq 0$, we have

$$\mathbb{E}\sqrt{1 + h^2(X - Y)^2} = \tfrac{1}{2}\sqrt{1 + h^2(a - b)^2} + \tfrac{1}{2}\sqrt{1 + h^2(a + b)^2},$$
$$\mathbb{E}\sqrt{1 + h^2(X - X')^2} = \tfrac{1}{2} + \tfrac{1}{2}\sqrt{1 + 4h^2a^2},$$
$$\mathbb{E}\sqrt{1 + h^2(Y - Y')^2} = \tfrac{1}{2} + \tfrac{1}{2}\sqrt{1 + 4h^2b^2},$$

and substituting into (57) yields

$$\mathbf{MMD}^2_{k_h}(\mu, \nu) = \sqrt{1 + h^2(a-b)^2} + \sqrt{1 + h^2(a+b)^2} - 1 - \tfrac{1}{2}\left(\sqrt{1 + 4h^2a^2} + \sqrt{1 + 4h^2b^2}\right). \tag{58}$$

Now consider the same $\mu, \nu \in \mathcal{P}(\mathbb{R}^2)$ and the full-rank anisotropic map $A_{s,\varepsilon} = \mathrm{diag}(s, s\varepsilon)$ as above. Define for $a, b \geq 0$

$$F_{\mathrm{MQ},h}(a, b) := \sqrt{1 + h^2(a-b)^2} + \sqrt{1 + h^2(a+b)^2} - 1 - \tfrac{1}{2}\left(\sqrt{1 + 4h^2a^2} + \sqrt{1 + 4h^2b^2}\right).$$

Using the identity (50) to reduce the $\varphi$-integral to $[0, \pi/2]$, and applying (58) slice-wise with $a = \cos\varphi$ and $b = \sin\varphi$, we obtain

$$\mathbf{SMMD}^2_{k_h}(\mu, \nu) = \frac{2}{\pi} \int_0^{\pi/2} F_{\mathrm{MQ},h}\left(\cos\varphi, \sin\varphi\right) d\varphi, \tag{59}$$

$$\mathbf{SMMD}^2_{k_h}\left((A_{s,\varepsilon})_{\#}\mu, (A_{s,\varepsilon})_{\#}\nu\right) = \frac{2}{\pi} \int_0^{\pi/2} F_{\mathrm{MQ},h}\left(s\cos\varphi, s\varepsilon\sin\varphi\right) d\varphi. \tag{60}$$

*Concrete numbers.* We evaluate the one-dimensional integrals in (59)–(60) using the same numerical integration method as earlier. Instantiating $h = 100$ (the value used in (Killingberg & Langseth, 2023)), and choosing $s = 0.9$ and $\varepsilon = 0.01$ (so $\|A_{s,\varepsilon}\|_{\mathrm{op}} = 0.9 < 1$), we obtain

$$\mathbf{SMMD}^2_{k_{100}}(\mu, \nu) \approx 51.758656, \qquad \mathbf{SMMD}^2_{k_{100}}\left((A_{0.9,0.01})_{\#}\mu, (A_{0.9,0.01})_{\#}\nu\right) \approx 55.552543, \qquad \text{ratio} \approx 1.07330,$$

so $\mathbf{SMMD}^2_{k_{100}}\left((A_{0.9,0.01})_{\#}\mu, (A_{0.9,0.01})_{\#}\nu\right) > \mathbf{SMMD}^2_{k_{100}}(\mu, \nu)$, which instantiates ($\star$) with $\Delta = \mathbf{SMMD}^2_{k_{100}}$.

## C.5. Max slicing

**Lemma C.10** (Max–sliced anisotropic scale contraction). *Let $\Delta$ be a divergence on $\mathcal{P}(\mathbb{R})$. Assume that for all $\alpha, \beta \in \mathcal{P}(\mathbb{R})$ the following holds:*

(S) *Scale contraction: there exists a nondecreasing $c : [0, \infty) \to [0, \infty)$ such that for every $s \in [0, 1]$,*

$$\Delta\big((x \mapsto sx)_{\#}\alpha, \, (x \mapsto sx)_{\#}\beta\big) \leq c(s)\,\Delta(\alpha, \beta),$$

*with $c(s) \leq 1$ for all $s \in [0, 1]$ and $c(s) < 1$ for all $s \in [0, 1)$.*

*Define the max–sliced lift of $\Delta$ by*

$$\mathbf{MS}\Delta(\mu, \nu) := \sup_{\theta \in \mathbb{S}^{d-1}} \Delta\big((P_\theta)_{\#}\mu, \, (P_\theta)_{\#}\nu\big), \qquad P_\theta(x) = \langle \theta, x \rangle.$$

*Then $\mathbf{MS}\Delta$ satisfies (S$_d$): for every linear map $A : \mathbb{R}^d \to \mathbb{R}^d$ with $\|A\|_{\mathrm{op}} \leq 1$,*

$$\boxed{\ \mathbf{MS}\Delta\big(A_{\#}\mu, \, A_{\#}\nu\big) \ \leq \ c(\|A\|_{\mathrm{op}})\,\mathbf{MS}\Delta(\mu, \nu), \qquad \forall\, \mu, \nu \in \mathcal{P}(\mathbb{R}^d).\ }$$

*In particular, if $\|A\|_{\mathrm{op}} < 1$ then $A_{\#}$ is a contraction on $(\mathcal{P}(\mathbb{R}^d), \mathbf{MS}\Delta)$ with factor $c(\|A\|_{\mathrm{op}}) < 1$.*

*Proof.* Fix $\theta \in \mathbb{S}^{d-1}$ and set $w_\theta := A^\top \theta$.

**Case 1:** $w_\theta = 0$. Then $P_\theta(Ax) = \langle A^\top \theta, x \rangle \equiv 0$, hence $(P_\theta)_{\#}A_{\#}\mu = (P_\theta)_{\#}A_{\#}\nu$ and

$$\Delta\big((P_\theta)_{\#}A_{\#}\mu, \, (P_\theta)_{\#}A_{\#}\nu\big) = 0.$$

**Case 2:** $\|w_\theta\| > 0$. Write $r_\theta := \|w_\theta\| \in (0, \infty)$ and $\phi_\theta := w_\theta / r_\theta \in \mathbb{S}^{d-1}$. For any $X \sim \mu$,

$$P_\theta(AX) = \langle \theta, AX \rangle = \langle A^\top \theta, X \rangle = r_\theta \langle \phi_\theta, X \rangle,$$

and similarly for $Y \sim \nu$. Therefore,

$$(P_\theta)_{\#}A_{\#}\mu \ = \ (x \mapsto r_\theta x)_{\#}(P_{\phi_\theta})_{\#}\mu, \qquad (P_\theta)_{\#}A_{\#}\nu \ = \ (x \mapsto r_\theta x)_{\#}(P_{\phi_\theta})_{\#}\nu.$$

Since $r_\theta = \|A^\top \theta\| \leq \|A^\top\|_{\mathrm{op}} = \|A\|_{\mathrm{op}} \leq 1$, we may apply (S) with $s = r_\theta$ and obtain

$$\Delta\big((P_\theta)_{\#}A_{\#}\mu, \, (P_\theta)_{\#}A_{\#}\nu\big) \leq c(r_\theta)\,\Delta\big((P_{\phi_\theta})_{\#}\mu, \, (P_{\phi_\theta})_{\#}\nu\big).$$

Taking the supremum over $\theta \in \mathbb{S}^{d-1}$ gives

$$\mathbf{MS}\Delta(A_{\#}\mu, A_{\#}\nu) = \sup_{\theta} \Delta\big((P_\theta)_{\#}A_{\#}\mu, \, (P_\theta)_{\#}A_{\#}\nu\big) \leq \sup_{\theta} c(r_\theta) \, \sup_{\phi} \Delta\big((P_\phi)_{\#}\mu, \, (P_\phi)_{\#}\nu\big).$$

Since $c$ is nondecreasing and $r_\theta \leq \|A\|_{\mathrm{op}}$ for all $\theta$, we have $\sup_\theta c(r_\theta) \leq c(\|A\|_{\mathrm{op}})$, hence

$$\mathbf{MS}\Delta(A_{\#}\mu, A_{\#}\nu) \leq c(\|A\|_{\mathrm{op}})\,\mathbf{MS}\Delta(\mu, \nu),$$

which is exactly (S$_d$) for $\mathbf{MS}\Delta$. $\square$

**Lemma C.11** (Max–sliced mixture $p$-convexity). *This result is the max–sliced analogue of Lemma C.9.*

*Let $\Delta$ be a divergence on $\mathcal{P}(\mathbb{R})$ that is mixture $p$-convex for some $p \in [1, \infty)$: for every probability space $(\Omega, \mathcal{F}, \rho)$ and measurable families $(\mu_c), (\nu_c) \subset \mathcal{P}(\mathbb{R})$,*

$$\Delta\left(\int_\Omega \mu_c\,\rho(dc), \int_\Omega \nu_c\,\rho(dc)\right) \leq \left(\int_\Omega \Delta(\mu_c, \nu_c)^p\,\rho(dc)\right)^{1/p}.$$

*Define the max–sliced lift on $\mathcal{P}(\mathbb{R}^d)$ by*

$$\mathbf{MS}\Delta(\mu,\nu) := \sup_{\theta \in \mathbb{S}^{d-1}} \Delta\big((P_\theta)_\# \mu, (P_\theta)_\# \nu\big), \qquad P_\theta(x) = \langle \theta, x \rangle.$$

*Then $\mathbf{MS}\Delta$ is also mixture $p$-convex:*

$$\mathbf{MS}\Delta\left(\int_\Omega \mu_c\, \rho(dc), \int_\Omega \nu_c\, \rho(dc)\right) \leq \left(\int_\Omega \mathbf{MS}\Delta(\mu_c,\nu_c)^p\, \rho(dc)\right)^{1/p}.$$

*Proof.* Fix $\theta \in \mathbb{S}^{d-1}$ and set

$$\mu_c^\theta := (P_\theta)_\# \mu_c, \qquad \nu_c^\theta := (P_\theta)_\# \nu_c \ \in \mathcal{P}(\mathbb{R}).$$

Pushforward commutes with mixtures:

$$(P_\theta)_\#\left(\int \mu_c\, d\rho\right) = \int \mu_c^\theta\, d\rho, \qquad (P_\theta)_\#\left(\int \nu_c\, d\rho\right) = \int \nu_c^\theta\, d\rho.$$

By mixture $p$-convexity of $\Delta$ in one dimension,

$$\Delta\big((P_\theta)_\# \int \mu_c\, d\rho,\ (P_\theta)_\# \int \nu_c\, d\rho\big) \leq \left(\int \Delta(\mu_c^\theta,\nu_c^\theta)^p\, d\rho\right)^{1/p}. \tag{61}$$

Taking the supremum over $\theta$ on the left-hand side of (61) gives

$$\sup_\theta\ \Delta\big((P_\theta)_\# \int \mu_c\, d\rho,\ (P_\theta)_\# \int \nu_c\, d\rho\big) \leq \sup_\theta\ \left(\int \Delta(\mu_c^\theta,\nu_c^\theta)^p\, d\rho\right)^{1/p}. \tag{62}$$

Define $f(\theta,c) := \Delta(\mu_c^\theta,\nu_c^\theta)$ and $h(c) := \sup_\phi f(\phi,c) = \mathbf{MS}\Delta(\mu_c,\nu_c)$. Since $f(\theta,c) \leq h(c)$ pointwise in $c$, we obtain for every $\theta$,

$$\left(\int f(\theta,c)^p\, d\rho(c)\right)^{1/p} \leq \left(\int h(c)^p\, d\rho(c)\right)^{1/p}.$$

Taking $\sup_\theta$ yields

$$\sup_\theta\ \left(\int \Delta(\mu_c^\theta,\nu_c^\theta)^p\, d\rho\right)^{1/p} \leq \left(\int \mathbf{MS}\Delta(\mu_c,\nu_c)^p\, d\rho\right)^{1/p}. \tag{63}$$

Combining (62) and (63) shows

$$\mathbf{MS}\Delta\left(\int \mu_c\, d\rho,\ \int \nu_c\, d\rho\right) \leq \left(\int \mathbf{MS}\Delta(\mu_c,\nu_c)^p\, \rho(dc)\right)^{1/p},$$

as claimed. $\qquad\square$

**Theorem C.5** (Supremum–max–sliced contraction of the multivariate distributional Bellman operator (anisotropic random linear map))**.** *Let $\Delta$ be a divergence on $\mathcal{P}(\mathbb{R})$ and fix $p \in [1,\infty)$. Define the max–sliced lift on $\mathcal{P}(\mathbb{R}^d)$ by*

$$\mathbf{MS}\Delta(\mu,\nu) := \sup_{\theta \in \mathbb{S}^{d-1}} \Delta\big((P_\theta)_\# \mu, (P_\theta)_\# \nu\big), \qquad P_\theta(x) = \langle \theta, x \rangle.$$

*Define the supremum metric over state–action indices by*

$$\overline{\mathbf{MS}\Delta}(\eta_1,\eta_2) := \sup_{(s,a)} \mathbf{MS}\Delta\big(\eta_1(s,a), \eta_2(s,a)\big), \qquad \eta_i : \mathcal{S} \times \mathcal{A} \to \mathcal{P}(\mathbb{R}^d).$$

*Assume that for all $\alpha, \beta \in \mathcal{P}(\mathbb{R})$ the following hold:*

**(T)** *Translation nonexpansion: for every $t \in \mathbb{R}$,*

$$\Delta\big((x \mapsto x + t)_{\#}\alpha,\ (x \mapsto x + t)_{\#}\beta\big) \le \Delta(\alpha, \beta).$$

**(S)** *Scale contraction: there exists a nondecreasing $c : [0, 1] \to [0, 1]$ such that for every $s \in [0, 1]$,*

$$\Delta\big((x \mapsto sx)_{\#}\alpha,\ (x \mapsto sx)_{\#}\beta\big) \le c(s)\, \Delta(\alpha, \beta),$$

*with $c(s) < 1$ for all $s \in [0, 1)$.*

**($M_p$)** *Mixture $p$-convexity: for every probability space $(\Omega, \mathcal{F}, \rho)$ and measurable families $(\mu_c), (\nu_c) \subset \mathcal{P}(\mathbb{R})$,*

$$\Delta\Big( \int_{\Omega} \mu_c\, \rho(dc),\ \int_{\Omega} \nu_c\, \rho(dc) \Big) \le \Big( \int_{\Omega} \Delta(\mu_c, \nu_c)^p\, \rho(dc) \Big)^{1/p}.$$

*Bellman update (anisotropic random linear map). Fix $(s, a)$. Gather all environment/policy randomness into a single random element $C$, which determines the successor index through a measurable mapping*

$$(S', A') := g(s, a; C).$$

*At $(s, a)$, apply an affine transformation with $C$-dependent translation and $C$-dependent linear map:*

$$b_{s,a}(C) \in \mathbb{R}^d, \qquad A_{s,a}(C) \in \mathbb{R}^{d \times d},$$

*and assume $\|A_{s,a}(C)\|_{\mathrm{op}} \le 1$ for all $C$. Conditioned on $C$, draw the next sample from the law at the successor index:*

$$X' \mid C \sim \eta(S', A').$$

*Define the Bellman update as the push-forward of $X'$ by this affine map:*

$$(T^{\pi}\eta)(s, a) := \mathrm{Law}\big(b_{s,a}(C) + A_{s,a}(C)\, X'\big).$$

*Define, for each $(s, a)$,*

$$L_{s,a}(C) := \|A_{s,a}(C)\|_{\mathrm{op}}, \qquad \bar{L} := \sup_{(s,a)} \sup_C L_{s,a}(C).$$

*Then, for all $\eta_1, \eta_2$,*

$$\boxed{\ \overline{\mathrm{MS}\Delta}\big(T^{\pi}\eta_1,\ T^{\pi}\eta_2\big)\ \le\ c(\bar{L})\, \overline{\mathrm{MS}\Delta}\big(\eta_1, \eta_2\big). \ }$$

*Consequently, if $\bar{L} < 1$, then $T^{\pi}$ is a contraction on $(\mathcal{S} \times \mathcal{A} \to \mathcal{P}(\mathbb{R}^d), \overline{\mathrm{MS}\Delta})$ with factor $c(\bar{L}) < 1$.*

*Proof.* We apply Theorem C.3 with the multivariate divergence

$$\Delta_{\mathrm{mv}} := \mathbf{MS}\Delta.$$

It suffices to verify that $\Delta_{\mathrm{mv}}$ satisfies **(T)**, **($S_d$)**, and **($M_p$)** with the same function $c$.

**Translation nonexpansion lifts.** Fix $t \in \mathbb{R}^d$ and let $T_t(x) = x + t$ on $\mathbb{R}^d$. For any $\theta \in \mathbb{S}^{d-1}$ and $\mu, \nu \in \mathcal{P}(\mathbb{R}^d)$,

$$(P_{\theta})_{\#}(T_t)_{\#}\mu = (x \mapsto x + \langle\theta, t\rangle)_{\#}(P_{\theta})_{\#}\mu, \qquad (P_{\theta})_{\#}(T_t)_{\#}\nu = (x \mapsto x + \langle\theta, t\rangle)_{\#}(P_{\theta})_{\#}\nu.$$

Applying **(T)** for $\Delta$ in one dimension gives

$$\Delta\big((P_{\theta})_{\#}(T_t)_{\#}\mu,\ (P_{\theta})_{\#}(T_t)_{\#}\nu\big) \le \Delta\big((P_{\theta})_{\#}\mu,\ (P_{\theta})_{\#}\nu\big).$$

Taking $\sup_{\theta}$ yields

$$\mathbf{MS}\Delta\big((T_t)_{\#}\mu,\ (T_t)_{\#}\nu\big) \le \mathbf{MS}\Delta(\mu, \nu),$$

which is **(T)** for $\Delta_{\mathrm{mv}}$.

**Anisotropic scale contraction $(\mathbf{S}_d)$ holds.** By Lemma C.10, $(\mathbf{S})$ for the base divergence $\Delta$ implies that $\mathbf{MS}\Delta$ satisfies $(\mathbf{S}_d)$: for every linear map $A$ with $\|A\|_{\mathrm{op}} \leq 1$,

$$\mathbf{MS}\Delta\big(A_{\#}\mu,\ A_{\#}\nu\big)\ \leq\ c(\|A\|_{\mathrm{op}})\, \mathbf{MS}\Delta(\mu, \nu).$$

**Mixture $p$-convexity $(\mathbf{M}_p)$ holds.** By Lemma C.11, mixture $p$-convexity of $\Delta$ in one dimension lifts to mixture $p$-convexity of $\mathbf{MS}\Delta$ on $\mathcal{P}(\mathbb{R}^d)$: for all probability spaces $(\Omega, \mathcal{F}, \rho)$ and measurable families $(\mu_c), (\nu_c) \subset \mathcal{P}(\mathbb{R}^d)$,

$$\mathbf{MS}\Delta\left(\int_\Omega \mu_c\, \rho(dc),\ \int_\Omega \nu_c\, \rho(dc)\right) \leq \left(\int_\Omega \mathbf{MS}\Delta(\mu_c, \nu_c)^p\, \rho(dc)\right)^{1/p}.$$

Having established $(\mathbf{T})$, $(\mathbf{S}_d)$, and $(\mathbf{M}_p)$ for $\Delta_{\mathrm{mv}} = \mathbf{MS}\Delta$, Theorem C.3 yields

$$\overline{\mathbf{MS}\Delta}\big(T^\pi \eta_1,\ T^\pi \eta_2\big)\ \leq\ c(\bar{L})\, \overline{\mathbf{MS}\Delta}\big(\eta_1, \eta_2\big),$$

with $\bar{L} = \sup_{(s,a)} \sup_C \|A_{s,a}(C)\|_{\mathrm{op}}$. $\hfill\square$

# D. Mixture bias

We first recall the *unbiased sample gradient* property (U) from Section 4.4. Let $\mu$ denote the data law and let $\nu_\theta$ be a parametric model law. Writing $\widehat{\mu}_m$ for the empirical measure of $m$ i.i.d. samples from $\mu$, we say that a divergence $\mathcal{D}$ satisfies (U) if

$$\mathbb{E}_{X_{1:m}\sim\mu}[\nabla_\theta\,\mathcal{D}(\widehat{\mu}_m,\,\nu_\theta)] \;=\; \nabla_\theta\,\mathcal{D}(\mu,\,\nu_\theta)\,. \qquad \text{(U)}$$

A small but important terminology point is that *divergence* here only refers to a population-level separation property (nonnegativity and $\mathcal{D}(\mu,\nu)=0$ iff $\mu=\nu$), so that $\nu=\mu$ minimizes the population objective $\nu \mapsto \mathcal{D}(\mu,\nu)$. In practice, however, learning minimizes the *sample loss* $\theta \mapsto \mathcal{D}(\widehat{\mu}_m,\nu_\theta)$ using plug-in gradients; (U) is precisely what ensures that, in expectation over the sampling of $\widehat{\mu}_m$, these gradients point in the direction of the population objective. Equivalently, when (U) holds, the expected sample loss is minimized at $\nu=\mu$, which is why such losses are called *proper* in this context (Bellemare et al., 2017b).

**Why this matters in distributional RL.** In distributional RL, the Bellman target is the *mixture law* induced by the transition and policy randomness, as introduced by the distributional Bellman operator in Eq. 3 (Section 2). However, a standard TD update draws a single sampled $(S', A')$ (and reward) and therefore forms a *one-sample based* target distribution $\text{Law}\big(R(s,a) + \Gamma(s,a)\,Z_{\phi^-}(S', A')\big)$ rather than the full mixture $(\mathcal{T}^\pi Z_{\phi^-})(s,a)$. As a result, learning implicitly targets the expected *sample loss*, which without (U) need not coincide with the intended *population* objective:

$$\arg\min_\phi\;\mathbb{E}_{S',A'}\Big[\mathcal{D}\big(\text{Law}(R+\Gamma Z_{\phi^-}(S', A')),\,Z_\phi(s,a)\big)\Big] \;\neq\; \arg\min_\phi\,\mathcal{D}\big((\mathcal{T}^\pi Z_{\phi^-})(s,a),\,Z_\phi(s,a)\big), \qquad (64)$$

as highlighted for Wasserstein objectives in Bellemare et al. (2017a).

**Wasserstein.** Wasserstein distances provide a canonical illustration where (U) fails in distributional RL: the Bellman target is a mixture over transition and policy randomness, yet training typically instantiates it from a *single* sampled next state and next action. In that regime, one generally *cannot* permute expectation and gradient as above, and the resulting plug-in stochastic gradients can be biased relative to the population Wasserstein objective (Bellemare et al., 2017a;b; Fatras et al., 2019). It is one reason why Wasserstein-based discrepancies are delicate in stochastic training for distributional RL (Bellemare et al., 2017a).

**Energy distance and MMD.** A broad family of kernel discrepancies used in practice can be written in the quadratic "energy" form

$$\mathcal{E}_k(\mu,\nu) := \mathbb{E}_{x,x'\sim\mu}[k(x,x')] + \mathbb{E}_{y,y'\sim\nu}[k(y,y')] - 2\mathbb{E}_{x\sim\mu,\,y\sim\nu}[k(x,y)], \qquad (65)$$

for a symmetric function $k$. When $k$ is positive definite this is exactly the usual squared MMD objective, and under our convention (Section A.3) the same formula can also be viewed as the squared "kernel energy" $\gamma_k(\mu,\nu)^2$ associated with a conditionally positive definite kernel (Sejdinovic et al., 2013).

For stochastic training, the key point is that the *squared* objective (65) satisfies (U). Indeed,

$$\mathcal{E}_k(\widehat{\mu}_m,\nu_\theta) = \mathbb{E}_{x,x'\sim\widehat{\mu}_m}[k(x,x')] + \mathbb{E}_{y,y'\sim\nu_\theta}[k(y,y')] - 2\mathbb{E}_{x\sim\widehat{\mu}_m,\,y\sim\nu_\theta}[k(x,y)],$$

where the first term does not depend on $\theta$ and thus vanishes under $\nabla_\theta$. Linearity of expectation then yields

$$\mathbb{E}_{X_{1:m}\sim\mu}[\nabla_\theta\,\mathcal{E}_k(\widehat{\mu}_m,\nu_\theta)] = \nabla_\theta\,\mathcal{E}_k(\mu,\nu_\theta),$$

so (U) holds (Bellemare et al., 2017b; Bińkowski et al., 2018).

**Consequences for energy distance and Cramér.** Because the (squared) energy distance admits the representation (65), it satisfies (U). Finally, in one dimension, the Cramér–2 divergence coincides with the (squared) energy distance, and the corresponding identity was already presented in Sec. A.2:

$$\ell_2^2(\mu,\nu) \;=\; \int_\mathbb{R}\big(F_\mu(t) - F_\nu(t)\big)^2 dt \;=\; \mathbb{E}\,|X-Y| \;-\; \tfrac{1}{2}\,\mathbb{E}\,|X-X'| \;-\; \tfrac{1}{2}\,\mathbb{E}\,|Y-Y'|, \qquad X, X' \sim \mu,\ Y, Y' \sim \nu \text{ i.i.d.}$$

## D.1. Interaction with slicing

### D.1.1. UNIFORM SLICING AND THE UNBIASED SAMPLE GRADIENT PROPERTY

Let $\Delta : \mathcal{P}(\mathbb{R}) \times \mathcal{P}(\mathbb{R}) \to [0, \infty)$ be a base divergence and let $p \in [1, \infty)$. For $\mu, \nu \in \mathcal{P}(\mathbb{R}^d)$, recall the uniform sliced powered objective

$$\mathbf{S}\Delta_p^p(\mu, \nu) := \int_{S^{d-1}} \Delta^p\big((P_\theta)_{\#}\mu, (P_\theta)_{\#}\nu\big) \, d\sigma(\theta), \qquad P_\theta(x) = \langle \theta, x \rangle.$$

**Lemma D.12** (Uniform slicing preserves (U) for the powered objective)**.** *Assume that $\Delta^p$ satisfies* (U) *in one dimension, i.e., for any data law $\mu \in \mathcal{P}(\mathbb{R})$, any model law $\nu_\phi \in \mathcal{P}(\mathbb{R})$, and the empirical measure $\widehat{\mu}_m$ of $m$ i.i.d. samples from $\mu$,*

$$\mathbb{E}\big[\nabla_\phi \, \Delta^p(\widehat{\mu}_m, \nu_\phi)\big] = \nabla_\phi \, \Delta^p(\mu, \nu_\phi).$$

*Then the sliced powered objective also satisfies* (U) *in $\mathbb{R}^d$: for any $\mu \in \mathcal{P}(\mathbb{R}^d)$, any model law $\nu_\phi \in \mathcal{P}(\mathbb{R}^d)$, and $\widehat{\mu}_m$ the empirical measure of $m$ i.i.d. samples from $\mu$,*

$$\boxed{\mathbb{E}\big[\nabla_\phi \, \mathbf{S}\Delta_p^p(\widehat{\mu}_m, \nu_\phi)\big] = \nabla_\phi \, \mathbf{S}\Delta_p^p(\mu, \nu_\phi).}$$

*Proof.* **Fix a direction and identify the projected empirical measure.**
Fix $\theta \in S^{d-1}$. Let $X_{1:m} \sim \mu$ in $\mathbb{R}^d$ and define $U_i := P_\theta(X_i) = \langle \theta, X_i \rangle$. Then $U_{1:m}$ are i.i.d. with law $(P_\theta)_{\#}\mu$, and

$$(P_\theta)_{\#}\widehat{\mu}_m = \frac{1}{m} \sum_{i=1}^m \delta_{P_\theta(X_i)} = \frac{1}{m} \sum_{i=1}^m \delta_{U_i},$$

so $(P_\theta)_{\#}\widehat{\mu}_m$ is exactly the empirical measure of the projected samples.

**Apply the one-dimensional (U) slice-wise.**
By the assumed one-dimensional (U) property for $\Delta^p$, applied to the pair $\big((P_\theta)_{\#}\mu, (P_\theta)_{\#}\nu_\phi\big)$, we obtain

$$\mathbb{E}_{X_{1:m}\sim\mu}\big[\nabla_\phi \, \Delta^p\big((P_\theta)_{\#}\widehat{\mu}_m, (P_\theta)_{\#}\nu_\phi\big)\big] = \nabla_\phi \, \Delta^p\big((P_\theta)_{\#}\mu, (P_\theta)_{\#}\nu_\phi\big).$$

**Integrate over directions.**
Averaging the slice-wise identity over directions $\theta \sim \sigma$ gives the result; *assuming*

$$\int_{S^{d-1}} \mathbb{E}_{X_{1:m}\sim\mu}\Big[\big\|\nabla_\phi \, \Delta^p\big((P_\theta)_{\#}\widehat{\mu}_m, (P_\theta)_{\#}\nu_\phi\big)\big\|\Big] \, d\sigma(\theta) < \infty,$$

we may exchange the expectation and the $\theta$-integral by Fubini. Concretely:

$$
\begin{aligned}
\mathbb{E}_{X_{1:m}\sim\mu}\big[\nabla_\phi \, \mathbf{S}\Delta_p^p(\widehat{\mu}_m, \nu_\phi)\big] &= \mathbb{E}_{X_{1:m}\sim\mu}\bigg[\nabla_\phi \int_{S^{d-1}} \Delta^p\big((P_\theta)_{\#}\widehat{\mu}_m, (P_\theta)_{\#}\nu_\phi\big) \, d\sigma(\theta)\bigg] \\
&= \mathbb{E}_{X_{1:m}\sim\mu}\bigg[\int_{S^{d-1}} \nabla_\phi \, \Delta^p\big((P_\theta)_{\#}\widehat{\mu}_m, (P_\theta)_{\#}\nu_\phi\big) \, d\sigma(\theta)\bigg] \\
&= \int_{S^{d-1}} \mathbb{E}_{X_{1:m}\sim\mu}\big[\nabla_\phi \, \Delta^p\big((P_\theta)_{\#}\widehat{\mu}_m, (P_\theta)_{\#}\nu_\phi\big)\big] \, d\sigma(\theta) \quad \text{(Fubini)} \\
&= \int_{S^{d-1}} \nabla_\phi \, \Delta^p\big((P_\theta)_{\#}\mu, (P_\theta)_{\#}\nu_\phi\big) \, d\sigma(\theta) \quad \text{(apply (U) on each fixed } \theta) \\
&= \nabla_\phi \int_{S^{d-1}} \Delta^p\big((P_\theta)_{\#}\mu, (P_\theta)_{\#}\nu_\phi\big) \, d\sigma(\theta) \\
&= \nabla_\phi \, \mathbf{S}\Delta_p^p(\mu, \nu_\phi).
\end{aligned}
$$

$\square$

**Lemma D.13** (Max-slicing generally breaks (U) via selection bias)**.** *Let $\Delta : \mathcal{P}(\mathbb{R}) \times \mathcal{P}(\mathbb{R}) \to [0, \infty]$ be a divergence and fix $p \geq 1$. For $\mu \in \mathcal{P}(\mathbb{R}^d)$ and a model $\nu_\phi \in \mathcal{P}(\mathbb{R}^d)$, define the max-sliced objective by*

$$\mathbf{MS}\Delta(\mu, \nu_\phi) := \sup_{\theta \in S^{d-1}} \Delta\big((P_\theta)_{\#}\mu, \, (P_\theta)_{\#}\nu_\phi\big), \qquad P_\theta(x) = \langle \theta, x \rangle.$$

*Let $\widehat{\mu}_m$ be the empirical measure of $m$ i.i.d. samples from $\mu$, and define the slice-wise empirical loss*

$$\widehat{\mathcal{L}}_\theta(\nu_\phi) := \Delta^p\big((P_\theta)_{\#}\widehat{\mu}_m, \, (P_\theta)_{\#}\nu_\phi\big), \qquad \theta \in S^{d-1}.$$

*Then for every fixed $\phi$,*

$$\mathbb{E}\Big[ \sup_{\theta \in S^{d-1}} \widehat{\mathcal{L}}_\theta(\nu_\phi) \Big] \geq \sup_{\theta \in S^{d-1}} \mathbb{E}\big[ \widehat{\mathcal{L}}_\theta(\nu_\phi) \big], \tag{66}$$

*and the inequality is typically strict whenever the maximizing direction depends on the same sample used to evaluate $\widehat{\mathcal{L}}_\theta$ (a standard* selection bias *effect).*

*Consequently, even if each fixed-direction objective $\widehat{\mathcal{L}}_\theta(\nu_\phi)$ enjoys (U) as a function of $\phi$ (i.e., unbiased plug-in gradients direction-wise), the max-sliced objective $\widehat{\mathcal{L}}_{\max}(\nu_\phi) := \sup_{\theta \in S^{d-1}} \widehat{\mathcal{L}}_\theta(\nu_\phi)$ need not satisfy (U) in general:*

$$\boxed{\mathbb{E}\Big[ \nabla_\phi \widehat{\mathcal{L}}_{\max}(\nu_\phi) \Big] \neq \nabla_\phi \Big( \sup_{\theta \in S^{d-1}} \Delta^p\big((P_\theta)_{\#}\mu, \, (P_\theta)_{\#}\nu_\phi\big) \Big) \qquad \textit{in general.}}$$

*Proof.* Fix $\phi$ and write $\widehat{\mathcal{L}}_\theta := \widehat{\mathcal{L}}_\theta(\nu_\phi)$. For each realization of $X_{1:m} \sim \mu$, the collection $\{\widehat{\mathcal{L}}_\theta\}_{\theta \in \Theta}$ is a family of real numbers indexed by $\Theta$ (for max-slicing, $\Theta = S^{d-1}$). Moreover, for any $f, g : \Theta \to \mathbb{R}$ and any $\lambda \in [0, 1]$,

$$\sup_{\theta \in \Theta} \big( \lambda f(\theta) + (1 - \lambda) g(\theta) \big) \leq \sup_{\theta \in \Theta} \big( \lambda f(\theta) \big) + \sup_{\theta \in \Theta} \big( (1 - \lambda) g(\theta) \big)$$
$$= \lambda \sup_{\theta \in \Theta} f(\theta) + (1 - \lambda) \sup_{\theta \in \Theta} g(\theta),$$

so $f \mapsto \sup_{\theta \in \Theta} f(\theta)$ is convex. Jensen's inequality then gives

$$\mathbb{E}\Big[ \sup_{\theta \in \Theta} \widehat{\mathcal{L}}_\theta \Big] \geq \sup_{\theta \in \Theta} \mathbb{E}\big[ \widehat{\mathcal{L}}_\theta \big],$$

which yields (66) (with $\Theta = S^{d-1}$). When the maximizer $\hat{\theta}(X_{1:m}) \in \arg\max_{\theta \in \Theta} \widehat{\mathcal{L}}_\theta$ depends *nontrivially* on the same sample $X_{1:m}$, the inequality is typically strict: the max step induces a selection bias by favoring indices $\theta$ for which the realized empirical loss is unusually large.

The same selection bias can propagate to gradients. Define $\widehat{\mathcal{L}}_{\max}(\nu_\phi) := \sup_{\theta \in \Theta} \widehat{\mathcal{L}}_\theta(\nu_\phi)$ and assume (for simplicity) that the maximizer is unique for the realized sample, so that

$$\widehat{\mathcal{L}}_{\max}(\nu_\phi) = \widehat{\mathcal{L}}_{\hat{\theta}(X_{1:m})}(\nu_\phi) \qquad \text{and} \qquad \nabla_\phi \widehat{\mathcal{L}}_{\max}(\nu_\phi) = \nabla_\phi \widehat{\mathcal{L}}_{\hat{\theta}(X_{1:m})}(\nu_\phi).$$

Even if for every fixed $\theta$ we have direction-wise (U),

$$\mathbb{E}\Big[ \nabla_\phi \widehat{\mathcal{L}}_\theta(\nu_\phi) \Big] = \nabla_\phi \Delta^p\big((P_\theta)_{\#}\mu, \, (P_\theta)_{\#}\nu_\phi\big),$$

the random choice $\hat{\theta}(X_{1:m})$ couples the selected index to the same sample used to evaluate the loss, so taking expectation of the selected gradient generally does not recover the gradient of the population max objective.

**A concrete strictness example (two indices).**
Fix two indices $\theta_1, \theta_2 \in \Theta$. Let $Z_1, Z_2$ be independent real random variables with $\mathbb{P}(Z_i = 1) = \mathbb{P}(Z_i = -1) = \frac{1}{2}$, and define for $\phi \in \mathbb{R}$

$$\widehat{\mathcal{L}}_{\theta_i}(\nu_\phi) := \phi Z_i, \qquad i = 1, 2.$$

Then $\nabla_\phi \widehat{\mathcal{L}}_{\theta_i}(\nu_\phi) = Z_i$ and $\mathbb{E}[Z_i] = 0$, so each fixed-index expected gradient vanishes. For any $\phi > 0$,

$$\widehat{\mathcal{L}}_{\max}(\nu_\phi) = \max\{\phi Z_1, \phi Z_2\} = \phi \max\{Z_1, Z_2\},$$

hence $\nabla_\phi \widehat{\mathcal{L}}_{\max}(\nu_\phi) = \max\{Z_1, Z_2\}$ and

$$\mathbb{E}\left[\nabla_\phi \widehat{\mathcal{L}}_{\max}(\nu_\phi)\right] = \mathbb{E}[\max\{Z_1, Z_2\}] = \frac{1}{2} \neq 0.$$

On the other hand, $\sup_{i \in \{1,2\}} \mathbb{E}[\widehat{\mathcal{L}}_{\theta_i}(\nu_\phi)] = 0$, so the gradient of the population restricted max objective is $0$. This shows explicitly that direction-wise unbiasedness does not lift through the max over $\theta$. $\square$

# E. Sample complexity

## E.1. Uniform slicing

**Theorem E.6** (Sample complexity of sliced divergences)**.** *This is a rewrite of Theorem 5 from Nadjahi et al. (2020).*

*Fix $p \in [1, \infty)$. Let $\Delta$ be a divergence on $\mathcal{P}(\mathbb{R})$ and assume there exists a function $\alpha(p, n) \geq 0$ such that for every $\mu \in \mathcal{P}(\mathbb{R})$ with empirical $\hat{\mu}_n$,*

$$\mathbb{E}\big[\Delta(\hat{\mu}_n, \mu)^p\big] \leq \alpha(p, n).$$

*For $\mu, \nu \in \mathcal{P}(\mathbb{R}^d)$, define*

$$\mathbf{S}\Delta_p(\mu, \nu) := \left( \int_{S^{d-1}} \Delta^p\big((P_\theta)_{\#}\mu, (P_\theta)_{\#}\nu\big) \, d\sigma(\theta) \right)^{1/p},$$

*where $P_\theta(x) = \langle \theta, x \rangle$ and $\sigma$ is the uniform probability measure on $S^{d-1}$. Then:*

(i) *For any $\mu \in \mathcal{P}(\mathbb{R}^d)$ with empirical $\hat{\mu}_n$,*

$$\mathbb{E}\,\big|\mathbf{S}\Delta_p^p(\hat{\mu}_n, \mu)\big| \leq \alpha(p, n).$$

(ii) *If $\Delta$ verifies nonnegativity, symmetry, and the triangle inequality on $\mathcal{P}(\mathbb{R})$ (hence $\mathbf{S}\Delta_p$ verifies them on $\mathcal{P}(\mathbb{R}^d)$ by Proposition 1), then for any $\mu, \nu \in \mathcal{P}(\mathbb{R}^d)$ with empirical measures $\hat{\mu}_n, \hat{\nu}_n$,*

$$\mathbb{E}\,\big|\mathbf{S}\Delta_p(\mu, \nu) - \mathbf{S}\Delta_p(\hat{\mu}_n, \hat{\nu}_n)\big| \leq 2\,\alpha(p, n)^{1/p}.$$

*Proof.* **(i) One-sample bound for $\mathbf{S}\Delta_p^p$.**

$$
\begin{aligned}
\mathbb{E}\,\big|\mathbf{S}\Delta_p^p(\hat{\mu}_n, \mu)\big| &= \mathbb{E}\left| \int_{S^{d-1}} \Delta^p\big((P_\theta)_{\#}\hat{\mu}_n, (P_\theta)_{\#}\mu\big) \, d\sigma(\theta) \right| \\
&\leq \mathbb{E} \int_{S^{d-1}} \big|\Delta^p\big((P_\theta)_{\#}\hat{\mu}_n, (P_\theta)_{\#}\mu\big)\big| \, d\sigma(\theta) \quad \text{(triangle inequality for the integral)} \\
&= \int_{S^{d-1}} \mathbb{E}\,\big|\Delta^p\big((P_\theta)_{\#}\hat{\mu}_n, (P_\theta)_{\#}\mu\big)\big| \, d\sigma(\theta) \quad \text{(Tonelli)} \\
&= \int_{S^{d-1}} \mathbb{E}\,\Delta^p\big((P_\theta)_{\#}\hat{\mu}_n, (P_\theta)_{\#}\mu\big) \, d\sigma(\theta) \quad \text{(non-negativity)} \\
&\leq \int_{S^{d-1}} \alpha(p, n) \, d\sigma(\theta) = \alpha(p, n).
\end{aligned}
$$

**(ii) Two-sample bound for $\mathbf{S}\Delta_p$.** By Proposition B.1 (triangle–inequality item), the triangle inequality for $\Delta$ on $\mathcal{P}(\mathbb{R})$ implies that $\mathbf{S}\Delta_p$ satisfies the triangle inequality on $\mathcal{P}(\mathbb{R}^d)$. Hence

$$
\begin{aligned}
\big|\mathbf{S}\Delta_p(\mu, \nu) - \mathbf{S}\Delta_p(\hat{\mu}_n, \hat{\nu}_n)\big| &\leq \big|\mathbf{S}\Delta_p(\hat{\mu}_n, \mu)\big| + \big|\mathbf{S}\Delta_p(\hat{\nu}_n, \nu)\big| \quad \text{(triangle inequality)} \\
&= \mathbf{S}\Delta_p(\hat{\mu}_n, \mu) + \mathbf{S}\Delta_p(\hat{\nu}_n, \nu) \quad \text{(non-negativity).}
\end{aligned}
$$

Taking expectations with respect to the empirical draws $(\hat{\mu}_n, \hat{\nu}_n)$,

$$\mathbb{E}\,\big|\mathbf{S}\Delta_p(\mu, \nu) - \mathbf{S}\Delta_p(\hat{\mu}_n, \hat{\nu}_n)\big| \leq \mathbb{E}\,\big|\mathbf{S}\Delta_p(\hat{\mu}_n, \mu)\big| + \mathbb{E}\,\big|\mathbf{S}\Delta_p(\hat{\nu}_n, \nu)\big|.$$

Since $x \mapsto x^{1/p}$ is concave for $p \geq 1$, Jensen's inequality gives

$$
\begin{aligned}
\mathbb{E}\,\big|\mathbf{S}\Delta_p(\hat{\mu}_n, \mu)\big| &\leq \big\{\mathbb{E}\,\big|\mathbf{S}\Delta_p(\hat{\mu}_n, \mu)\big|^p\big\}^{1/p} = \big\{\mathbb{E}\,\mathbf{S}\Delta_p^p(\hat{\mu}_n, \mu)\big\}^{1/p}, \\
\mathbb{E}\,\big|\mathbf{S}\Delta_p(\hat{\nu}_n, \nu)\big| &\leq \big\{\mathbb{E}\,\big|\mathbf{S}\Delta_p(\hat{\nu}_n, \nu)\big|^p\big\}^{1/p} = \big\{\mathbb{E}\,\mathbf{S}\Delta_p^p(\hat{\nu}_n, \nu)\big\}^{1/p}.
\end{aligned}
$$

Applying the bound from part (i) to both terms,

$$\mathbb{E}\,\big|\mathbf{S}\Delta_p(\mu, \nu) - \mathbf{S}\Delta_p(\hat{\mu}_n, \hat{\nu}_n)\big| \leq \alpha(p, n)^{1/p} + \alpha(p, n)^{1/p} = 2\,\alpha(p, n)^{1/p}.$$

$\square$

**Lemma E.14** (Mean $L^2(F)$ CDF discrepancy for the empirical CDF). *Let $X_1, \ldots, X_n$ be i.i.d. with cumulative distribution function $F$, and let $F_n(t) := \frac{1}{n} \sum_{i=1}^{n} \mathbf{1}\{X_i \le t\}$ be the empirical CDF. Then*

$$\boxed{\mathbb{E}\,\|F_n - F\|_{L^2(F)} \;\le\; \frac{1}{2\sqrt{n}} \;=\; \mathcal{O}(n^{-1/2}).}$$

*Proof.* By definition,

$$\|F_n - F\|_{L^2(F)}^2 \;=\; \int_{\mathbb{R}} (F_n(t) - F(t))^2 \, dF(t),$$

and therefore

$$\mathbb{E}\|F_n - F\|_{L^2(F)}^2 = \mathbb{E}\left[\int_{\mathbb{R}} (F_n(t) - F(t))^2 \, dF(t)\right].$$

Since the integrand is nonnegative, Tonelli's theorem yields

$$\mathbb{E}\left[\int_{\mathbb{R}} (F_n(t) - F(t))^2 \, dF(t)\right] = \int_{\mathbb{R}} \mathbb{E}\big[(F_n(t) - F(t))^2\big] \, dF(t),$$

so that

$$\mathbb{E}\|F_n - F\|_{L^2(F)}^2 = \int_{\mathbb{R}} \mathbb{E}\big[(F_n(t) - F(t))^2\big] \, dF(t).$$

Fix $t \in \mathbb{R}$ and set $Y_i(t) := \mathbf{1}\{X_i \le t\}$, so that $F_n(t) = \frac{1}{n} \sum_{i=1}^{n} Y_i(t)$. Moreover,

$$\mathbb{P}\big(Y_i(t) = 1\big) = \mathbb{P}(X_i \le t) = F(t),$$

so $Y_i(t) \sim \mathrm{Bernoulli}(F(t))$. Hence

$$\mathbb{E}[Y_i(t)] = F(t), \qquad \mathrm{Var}(Y_i(t)) = F(t)\big(1 - F(t)\big).$$

By independence,

$$\mathrm{Var}(F_n(t)) = \mathrm{Var}\left(\frac{1}{n} \sum_{i=1}^{n} Y_i(t)\right) = \frac{1}{n^2} \sum_{i=1}^{n} \mathrm{Var}(Y_i(t)) = \frac{F(t)(1 - F(t))}{n}.$$

Moreover $\mathbb{E}[F_n(t)] = F(t)$, so $F_n(t)$ is unbiased and therefore

$$\mathbb{E}\big[(F_n(t) - F(t))^2\big] = \mathrm{Var}(F_n(t)) = \frac{F(t)(1 - F(t))}{n}.$$

Substituting back gives the identity

$$\mathbb{E}\|F_n - F\|_{L^2(F)}^2 = \frac{1}{n} \int_{\mathbb{R}} F(t)(1 - F(t)) \, dF(t).$$

Since $u(1 - u) \le \frac{1}{4}$ for all $u \in [0, 1]$ and $F(t) \in [0, 1]$, we obtain

$$\mathbb{E}\|F_n - F\|_{L^2(F)}^2 \le \frac{1}{n} \int_{\mathbb{R}} \frac{1}{4} \, dF(t) = \frac{1}{4n}.$$

Finally, by Jensen's inequality for the concave map $x \mapsto \sqrt{x}$,

$$\mathbb{E}\|F_n - F\|_{L^2(F)} = \mathbb{E}\sqrt{\|F_n - F\|_{L^2(F)}^2} \le \sqrt{\mathbb{E}\|F_n - F\|_{L^2(F)}^2} \le \sqrt{\frac{1}{4n}} = \frac{1}{2\sqrt{n}}.$$

$\square$

### E.2. Max slicing

**Lemma E.15** (Half–spaces and CDFs of projections). *As noted in the proof of Theorem 4 of (Nguyen-Tang et al., 2021), the CDF of a projection can be written as the probability of a half–space.*

*Let $P \in \mathcal{P}(\mathbb{R}^d)$ and $X_1, \ldots, X_n \overset{iid}{\sim} P$, with empirical measure $P_n = \frac{1}{n} \sum_{i=1}^n \delta_{X_i}$. For $\theta \in \mathbb{S}^{d-1}$ and $t \in \mathbb{R}$, define the half–space*

$$H_{\theta,t} := \{x \in \mathbb{R}^d : \langle \theta, x \rangle \le t\}.$$

*We also write $P_\theta(x) = \langle \theta, x \rangle$ for the one–dimensional projection map. Then, for all $t \in \mathbb{R}$, the CDF of the projection $(P_\theta)_\# P$ is*

$$F_\theta(t) = (P_\theta)_\# P((-\infty, t]) = P(H_{\theta,t}),$$

*while the empirical CDF of the projection $(P_\theta)_\# P_n$ is*

$$F_{n,\theta}(t) = (P_\theta)_\# P_n((-\infty, t]) = P_n(H_{\theta,t}) = \frac{1}{n} \sum_{i=1}^n \mathbf{1}\{\langle \theta, X_i \rangle \le t\}.$$

*Proof.* By definition of the pushforward, for any Borel $A \subseteq \mathbb{R}$,

$$(P_\theta)_\# P(A) = P\big(\{x \in \mathbb{R}^d : P_\theta(x) \in A\}\big).$$

Taking $A = (-\infty, t]$ yields

$$F_\theta(t) = (P_\theta)_\# P((-\infty, t]) = P\big(\{x : \langle \theta, x \rangle \le t\}\big) = P(H_{\theta,t}).$$

The same argument with $P$ replaced by $P_n$ gives

$$F_{n,\theta}(t) = (P_\theta)_\# P_n((-\infty, t]) = P_n(H_{\theta,t}).$$

Finally, since $P_n$ is the empirical measure,

$$P_n(H_{\theta,t}) = \frac{1}{n} \sum_{i=1}^n \mathbf{1}\{\langle \theta, X_i \rangle \le t\}.$$

$\square$

**Lemma E.16** (VC inequality for half–spaces in $\mathbb{R}^d$). *Let $P \in \mathcal{P}(\mathbb{R}^d)$, let $X_1, \ldots, X_n \overset{iid}{\sim} P$ with empirical measure $P_n = \frac{1}{n} \sum_{i=1}^n \delta_{X_i}$, and let*

$$\mathcal{H} = \big\{H_{\theta,t} = \{x \in \mathbb{R}^d : \langle \theta, x \rangle \le t\} : \theta \in \mathbb{S}^{d-1}, t \in \mathbb{R}\big\}.$$

*Define*

$$Z := \sup_{H \in \mathcal{H}} \big|P_n(H) - P(H)\big| = \sup_{\theta \in \mathbb{S}^{d-1}, t \in \mathbb{R}} \big|P_n(H_{\theta,t}) - P(H_{\theta,t})\big|.$$

*Then, for any $\delta \in (0, 1)$,*

$$\Pr\Big(Z \le c_{n,\delta}\Big) \ge 1 - \delta, \qquad c_{n,\delta} := \sqrt{\frac{32}{n}\Big((d+1)\log(n+1) + \log \frac{8}{\delta}\Big)}.$$

*This is the explicit VC bound used in the proof of Theorem 4 of (Nguyen-Tang et al., 2021).*

**Theorem E.7** (Max–sliced bound from a 1D CDF control, in expectation). *Let $P \in \mathcal{P}(\mathbb{R}^d)$ and $X_1, \ldots, X_n \overset{iid}{\sim} P$ with empirical measure $P_n = \frac{1}{n} \sum_{i=1}^n \delta_{X_i}$. Assume $\mathrm{diam}(\mathrm{supp}\, P) \le D$ (so for every $\theta$, the range of $x \mapsto \langle \theta, x \rangle$ over $\mathrm{supp}\, P$ has length $\le D$). Let $\Delta$ be a divergence on $\mathcal{P}(\mathbb{R})$ such that for any one–dimensional laws $\mu, \nu$ supported on an interval of length $\le D$ there exist*

$$\alpha \in (0, 1], \qquad \beta \ge 0, \qquad L > 0$$

*with the* CDF–dominance *inequality*

$$\Delta(\mu, \nu) \ \leq \ L\, D^\beta \, \|F_\mu - F_\nu\|_\infty^\alpha. \tag{A}$$

*Define*

$$\mathbf{MS}\Delta(\mu, \nu) := \sup_{\theta \in \mathbb{S}^{d-1}} \Delta\big((P_\theta)_\# \mu, \, (P_\theta)_\# \nu\big), \qquad P_\theta(x) = \langle \theta, x \rangle.$$

*Then*

$$\boxed{\mathbb{E}\,\mathbf{MS}\Delta(P_n, P) = \mathcal{O}\Big( D^\beta \left( \tfrac{d \log n}{n} \right)^{\alpha/2} \Big).}$$

*More precisely, there exists a constant $C_\Delta$ depending only on $L$ and $\alpha$ such that*

$$\mathbb{E}\,\mathbf{MS}\Delta(P_n, P) \ \leq \ L\, D^\beta \left( \sqrt{\tfrac{32(d+1)\log(n+1)}{n}} + 4\sqrt{\tfrac{32\pi}{n}} \right)^\alpha \ \leq \ C_\Delta \, D^\beta \left( \sqrt{\tfrac{d \log(n+1)}{n}} \right)^\alpha.$$

*Proof.* Let

$$Z \ := \ \sup_{\theta \in \mathbb{S}^{d-1},\, t \in \mathbb{R}} \big| F_{n,\theta}(t) - F_\theta(t) \big|,$$

where Lemma E.15 identifies $F_{n,\theta}(t) = P_n(H_{\theta,t})$ and $F_\theta(t) = P(H_{\theta,t})$. By (A), for each $\theta$,

$$\Delta\big((P_\theta)_\# P_n, (P_\theta)_\# P\big) \ \leq \ L\, D^\beta \, \|F_{n,\theta} - F_\theta\|_\infty^\alpha,$$

hence, after taking $\sup_\theta$,

$$\mathbf{MS}\Delta(P_n, P) \ \leq \ L\, D^\beta \, Z^\alpha.$$

Taking expectations and using Jensen (concavity of $x \mapsto x^\alpha$ for $\alpha \in (0,1]$),

$$\mathbb{E}\,\mathbf{MS}\Delta(P_n, P) \ \leq \ L\, D^\beta \, \mathbb{E}[Z^\alpha] \ \leq \ L\, D^\beta \, (\mathbb{E}Z)^\alpha.$$

By Lemma E.16, for any $\delta \in (0,1)$, $\Pr(Z \leq c_{n,\delta}) \geq 1 - \delta$ with $c_{n,\delta}$ as stated. Put $b_n := \sqrt{32(d+1)\log(n+1)/n}$ and take $\delta = 8e^{-ns^2/32}$ so that $c_{n,\delta} \leq b_n + s$ and $\Pr(Z > b_n + s) \leq 8e^{-ns^2/32}$ for all $s \geq 0$. Integrating the tail,

$$\mathbb{E}Z \ = \ \int_0^\infty \Pr(Z > t)\, dt \ \leq \ b_n + \int_0^\infty 8e^{-ns^2/32}\, ds \ = \ b_n + 4\sqrt{\tfrac{32\pi}{n}}.$$

Insert this into the previous display and absorb numerical constants into $C_\Delta$ to obtain the claim. $\qquad\square$

**Corollary E.3** (Max–sliced $\mathbf{W}_1$). *If $\Delta = \mathbf{W}_1$ (one-dimensional Wasserstein–1), then*

$$\boxed{\mathbb{E}\,\mathbf{MSW}_1(P_n, P) = \mathcal{O}\Big( D\, \sqrt{\tfrac{d \log n}{n}} \Big).}$$

*Proof.* By Vallender's identity (Vallender, 1974), for probability laws $\alpha, \beta$ on $\mathbb{R}$ with CDFs $F_\alpha, F_\beta$,

$$\mathbf{W}_1(\alpha, \beta) \ = \ \int_{\mathbb{R}} \big| F_\alpha(x) - F_\beta(x) \big|\, dx.$$

If the support of $\alpha$ and $\beta$ lies within an interval of length $D$, then

$$\int_{\mathbb{R}} \big| F_\alpha(x) - F_\beta(x) \big|\, dx \ \leq \ D\, \|F_\alpha - F_\beta\|_\infty.$$

Hence

$$\mathbf{W}_1(\alpha, \beta) \ \leq \ D\, \|F_\alpha - F_\beta\|_\infty,$$

which verifies condition (A) with $(\alpha, \beta, L) = (1, 1, 1)$. Applying Theorem E.7 concludes the proof. $\qquad\square$

**Corollary E.4** (Max–sliced $\mathbf{W}_p$ for $p > 1$). *Fix $p > 1$ and $\Delta = \mathbf{W}_p$. Then*

$$\boxed{\mathbb{E}\,\mathbf{MSW}_p(P_n, P) = \mathcal{O}\Big( D \left( \tfrac{d \log n}{n} \right)^{1/(2p)} \Big).}$$

*Proof.* By the 1D quantile representation,

$$\mathbf{W}_p^p(\alpha, \beta) = \int_0^1 \left| F_\alpha^{-1}(u) - F_\beta^{-1}(u) \right|^p du.$$

If $\alpha, \beta$ are supported on an interval of length $D$, then every quantile difference $F_\alpha^{-1}(u) - F_\beta^{-1}(u)$ lies in $[-D, D]$. Hence, for $x = F_\alpha^{-1}(u) - F_\beta^{-1}(u)$,

$$|x|^p = |x|^{p-1} |x| \ \leq \ D^{p-1} |x|.$$

Applying this bound inside the integral gives

$$\mathbf{W}_p^p(\alpha, \beta) \ \leq \ D^{p-1} \int_0^1 \left| F_\alpha^{-1}(u) - F_\beta^{-1}(u) \right| du.$$

The integral on the right is exactly the 1D Wasserstein–1 distance,

$$\int_0^1 \left| F_\alpha^{-1}(u) - F_\beta^{-1}(u) \right| du \ = \ \mathbf{W}_1(\alpha, \beta).$$

Hence

$$\mathbf{W}_p^p(\alpha, \beta) \ \leq \ D^{p-1} \, \mathbf{W}_1(\alpha, \beta).$$

By Vallender's identity (Vallender, 1974) and the support bound of length $D$, we already established in Corollary E.3 that

$$\mathbf{W}_1(\alpha, \beta) \ \leq \ D \, \|F_\alpha - F_\beta\|_\infty.$$

Combining the two inequalities yields

$$\mathbf{W}_p^p(\alpha, \beta) \ \leq \ D^p \, \|F_\alpha - F_\beta\|_\infty.$$

Taking the $p$-th root finally gives

$$\mathbf{W}_p(\alpha, \beta) \ \leq \ D \, \|F_\alpha - F_\beta\|_\infty^{1/p}.$$

Thus condition (A) holds with $(\alpha, \beta, L) = (1/p, 1, 1)$, and Theorem E.7 applies. $\qquad\square$

**Corollary E.5** (Max–sliced Cramér). *Let* $\Delta(\alpha, \beta) = \|F_\alpha - F_\beta\|_{L^2(\mathbb{R})}$. *Then*

$$\boxed{\mathbb{E} \, \mathbf{MSC}_2(\hat{\mu}_n, \mu) = \mathcal{O}\!\left( \sqrt{D} \, \sqrt{\tfrac{d \log n}{n}} \right).}$$

*Proof.* On an interval of length $D$, one has $\| \cdot \|_{L^2} \leq D^{1/2} \| \cdot \|_\infty$, so (A) holds with $(\alpha, \beta, L) = (1, 1/2, 1)$. Applying Theorem E.7 yields the result. $\qquad\square$

## F. Instantiations

This section collects concrete instantiations of our contraction results for Wasserstein, Cramér, and MMD divergences. For each choice, we state the relevant structural properties, contraction factors under shared scalar and general matrix discounting, and we record suitability for distributional RL training and sample complexity. We cover Wasserstein in Section F.1, Cramér in Section F.2, and MMD in Section F.3. We summarize all results in Table 1.

| Divergence | 3 prop. | Contr. fac. $\gamma$ | Contr. fac. $\bar{L}$ | Sample complexity | Comp. cost | Grad. bias (U) |
|---|---|---|---|---|---|---|
| $\mathbf{SW}_p$ | ✓ | $\gamma$ | / | $\mathcal{O}\big(n^{-1/(2p)}\big)$ | $\mathcal{O}(L\,n\log n)$ | unavoidable |
| $\mathbf{MSW}_p$ | ✓ | $\gamma$ | $\bar{L}$ | $\mathcal{O}\Big(D\big(\frac{d\log n}{n}\big)^{1/(2p)}\Big)$ | $\mathcal{O}(D\,n\log n)$ | unavoidable |
| $\mathbf{SC}_2$ | ✓ | $\gamma^{1/2}$ | / | $\mathcal{O}\big(n^{-1/2}\big)$ | $\mathcal{O}(L\,n\log n)$ | avoidable |
| $\mathbf{MSC}_2$ | ✓ | $\gamma^{1/2}$ | $\bar{L}^{1/2}$ | $\mathcal{O}\Big(\sqrt{D}\,\sqrt{\frac{d\log n}{n}}\Big)$ | $\mathcal{O}(D\,n\log n)$ | unavoidable |
| $\mathbf{SMMD}_{k_h}$ | ✓ | $\gamma^{1/2}$ | / | $\mathcal{O}\big(n^{-1/2}\big)$ | $\mathcal{O}(L\,n^2)$ | avoidable |
| $\mathbf{MSMMD}_{k_h}$ | ✓ | $\gamma^{1/2}$ | $\bar{L}^{1/2}$ | $\times$ | $\mathcal{O}(D\,n^2)$ | unavoidable |
| $\mathbf{W}_p$ | ✓ | $\gamma$ | $\bar{L}$ | $\times$ | $\mathcal{O}(n^3\log n)$ | unavoidable |

*Table 1.* **Summary of contraction factors, sample complexity, computational cost, and gradient bias (U).** Summary of contraction factors and sample complexity results for sliced and max–sliced divergences under matrix discounting $R + \Gamma Z(S', A')$. Uniform sliced contractions with $\Gamma = \gamma I$ follow from Theorem C.4. Max–sliced contractions with a general linear map $\Gamma$ follow from Theorem C.5, where $\bar{L}$ denotes the corresponding operator–norm envelope. In the computational costs, $L$ denotes the number of Monte Carlo projection directions used for uniform slicing, and $D$ denotes the number of candidate directions evaluated for max slicing. Uniform slicing yields dimension–free rates by Theorem E.6 together with the stated one–dimensional base bounds; max–sliced Wasserstein and Cramér rates follow from Theorem E.7 and Corollaries E.3–E.4, E.5. Exact OT in $\mathbb{R}^d$ can scale as $\mathcal{O}(n^3\log n)$ for standard solvers (Genevay et al., 2019). For completeness, we defer sample complexity bounds for $p$–Wasserstein distances to (Fournier & Guillin, 2015). For $\mathbf{MSMMD}_{k_h}$, we do not provide a sharp sample complexity bound.

### F.1. Wasserstein

Wasserstein is a metric on $\mathcal{P}_p(\mathbb{R})$ for every $p \in [1, \infty]$ (Proposition 2 in (Givens & Shortt, 1984)). It satisfies (T) since it is translation invariant. It satisfies (S) with $c(s) = s$ by linear push-forward contraction (Proposition A.1). It satisfies ($\mathbf{M}_p$) by mixture $p$-convexity (Proposition A.2).

*Gradient bias.* As discussed in Section D, Wasserstein discrepancies generally fail (U) in the one-sample TD regime when a mixture target is instantiated from a single sampled $(S', A')$ and reward as in Eq. 64. Therefore, gradient bias is unavoidable for one-sample TD training with Wasserstein objectives.

*Contraction and sample complexity.* Combining (T), Proposition A.1, and Proposition A.2 yields

$$\mathbf{W}_p\big(T^\pi\eta_1, T^\pi\eta_2\big) \;\leq\; \|\Gamma\|_{\mathrm{op}}\,\mathbf{W}_p(\eta_1, \eta_2),$$

hence strict contraction whenever $\|\Gamma\|_{\mathrm{op}} < 1$. In particular, for $\Gamma = \gamma I_d$ this reduces to the factor $\gamma$. For empirical Wasserstein in dimension $d$, the expected estimation error deteriorates with dimension (Fournier & Guillin, 2015), so we do not record a dimension-robust rate for $\mathbf{W}_p$ itself.

*Sliced variants.* By Theorem C.4 with $\Delta = \mathbf{W}_p$, the uniform sliced Wasserstein update with $\Gamma = \gamma I_d$ contracts with factor $c(\gamma) < 1$,

$$\mathbf{SW}_p\big(T^\pi\eta_1, T^\pi\eta_2\big) \;\leq\; c(\gamma)\,\mathbf{SW}_p(\eta_1, \eta_2).$$

By Theorem C.5 with $\Delta = \mathbf{W}_p$, the max-sliced Wasserstein update under a general random linear map (with envelope $\bar{L} = \sup_{(s,a),C} \|A_{s,a}(C)\|_{\mathrm{op}}$) contracts with factor $c(\bar{L})$, strictly so whenever $\bar{L} < 1$,

$$\mathbf{MSW}_p\big(T^\pi\eta_1, T^\pi\eta_2\big) \;\leq\; c(\bar{L})\,\mathbf{MSW}_p(\eta_1, \eta_2).$$

*Sample complexity (uniform slicing).* Let $p \in [1, \infty)$ and assume $\mu \in \mathcal{P}_q(\mathbb{R}^d)$ with $q > 2p$ (finite $q$-th moment). Let $\hat{\mu}_n$ be the empirical measure from $n$ samples. Carrying the same steps as in Corollary 2 of Nadjahi et al. (2020) but in

the one-sample setting, and plugging the 1D base bound from Theorem 1 of Fournier & Guillin (2015), we obtain the dimension–free rate

$$\mathbb{E}[\mathbf{SW}_p(\hat{\mu}_n, \mu)] = \mathcal{O}\Big(n^{-1/(2p)}\Big).$$

Thus, uniform slicing avoids the curse of dimensionality.

*Sample complexity (max-sliced).* By Theorem E.7 and Corollaries E.3–E.4, if $\operatorname{diam}(\operatorname{supp}\mu) \leq D$ then

$$\mathbb{E}\,\mathbf{MSW}_1(\hat{\mu}_n, \mu) = \mathcal{O}\Big(D\,\sqrt{\tfrac{d \log n}{n}}\Big), \qquad \mathbb{E}\,\mathbf{MSW}_p(\hat{\mu}_n, \mu) = \mathcal{O}\Big(D\,\big(\tfrac{d \log n}{n}\big)^{1/(2p)}\Big) \qquad (p > 1).$$

## F.2. Cramér

The Cramér distance enjoys all the structural assumptions we require. By Proposition A.3, it is a metric. It satisfies **(T)** by Proposition 2 in (Bellemare et al., 2017b) and Proposition 3.2 in (Odin & Charpentier, 2020), and **(S)** with $c(s) = s^{1/2}$ via Proposition A.4. It also satisfies **(M$_p$)** (Proposition A.5).

*Gradient bias.* As discussed in Section D, the *squared* energy-distance form satisfies **(U)**, and in one dimension Cramér–2 coincides with squared energy distance. By Lemma D.12, this unbiasedness lifts to uniform slicing for the powered objective, so the guarantee applies to the *squared* sliced Cramér objective $\mathbf{SC}_2^2$.

*Contraction factors.* By Theorem C.4 with $\Delta = \mathbf{C}_2$ (so $c(s) = s^{1/2}$), the *sliced Cramér* update with $\Gamma = \gamma I$ contracts with factor $\gamma^{1/2}$:

$$\mathbf{SC}_2\big(T^\pi \eta_1, T^\pi \eta_2\big) \leq \gamma^{1/2}\,\mathbf{SC}_2\big(\eta_1, \eta_2\big).$$

By Theorem C.5, the *max–sliced Cramér* update with a general linear map $\Gamma$ contracts with factor $c(\bar{L}) = \bar{L}^{1/2}$, strictly so whenever $\bar{L} < 1$:

$$\mathbf{MSC}_2\big(T^\pi \eta_1, T^\pi \eta_2\big) \leq \bar{L}^{1/2}\,\mathbf{MSC}_2\big(\eta_1, \eta_2\big).$$

*Sample complexity (uniform slicing).* For the one–dimensional Cramér distance (the $L^2$–CDF discrepancy), Lemma E.14 yields

$$\mathbb{E}\,\|F_n - F\|_{L^2(F)} = \mathcal{O}\Big(n^{-1/2}\Big).$$

Plugging this base rate into Theorem E.6 yields the dimension–free bound

$$\mathbb{E}[\mathbf{SC}_2(\hat{\mu}_n, \mu)] = \mathcal{O}\Big(n^{-1/2}\Big).$$

Thus, uniform slicing avoids the curse of dimensionality.

*Sample complexity (max–sliced).* By Theorem E.7 and Corollary E.5, for $\operatorname{diam}(\operatorname{supp}\mu) \leq D$,

$$\mathbb{E}[\mathbf{MSC}_2(\hat{\mu}_n, \mu)] = \mathcal{O}\Big(\sqrt{D}\,\sqrt{\tfrac{d \log n}{n}}\Big).$$

## F.3. MMD

The Maximum Mean Discrepancy (MMD) with a conditionally strictly positive definite kernel (Gretton et al., 2012; Sejdinovic et al., 2013) is a valid metric on probability laws. With the multiquadric (MQ) kernel $k_h(x,y) = -\sqrt{1 + h^2\|x - y\|^2}$, it enjoys all the structural assumptions we require. By Proposition A.6, it is a metric. It satisfies **(T)** since MMD is translation invariant for all shift–invariant kernels. It satisfies **(S)** with $c(s) = s^{1/2}$ for the MQ kernel (Proposition A.7), reflecting its scale–sensitivity. Finally, it satisfies **(M$_p$)** by mixture convexity of RKHS embeddings (Proposition A.9).

*Gradient bias.* As discussed in Section D, squared kernel energy objectives of the form (65) satisfy **(U)**. By Lemma D.12, this unbiasedness lifts to uniform slicing for the powered objective $\mathbf{SMMD}_{k_h}^2$, so the guarantee applies to the squared sliced MMD.

*Contraction factors.* By Theorem C.4 with $\Delta = \mathbf{MMD}_{k_h}$ and the scale bound

$$c(s) = s^{1/2},$$

the *sliced MMD* update with $\Gamma = \gamma I$ satisfies

$$\mathbf{SMMD}_{k_h}\big(T^\pi \eta_1, \, T^\pi \eta_2\big) \; \leq \; c(\gamma) \, \mathbf{SMMD}_{k_h}\big(\eta_1, \, \eta_2\big).$$

In particular, for scalar discounts $\gamma \in (0, 1)$ we have $c(\gamma) = \gamma^{1/2}$.

By Theorem C.5, the *max–sliced MMD* update with a general linear map $\Gamma$ satisfies

$$\mathbf{MSMMD}_{k_h}\big(T^\pi \eta_1, \, T^\pi \eta_2\big) \; \leq \; c(\bar{L}) \, \mathbf{MSMMD}_{k_h}\big(\eta_1, \, \eta_2\big), \qquad c(\bar{L}) = \bar{L}^{1/2}.$$

In particular, under $\bar{L} < 1$ this is a strict contraction.

*Sample complexity (uniform slicing).* In one dimension, the unbiased empirical MMD (equivalently, the energy distance) is a U–statistic (Gretton et al., 2012; Sejdinovic et al., 2013), so classical U–statistic theory yields the standard rate

$$\mathbb{E}[\mathbf{MMD}_{k_h}(\hat{\mu}_n, \mu)] \; = \; \mathcal{O}\Big(n^{-1/2}\Big).$$

Plugging this into Theorem E.6(i) yields the dimension–free bound

$$\mathbb{E}[\mathbf{SMMD}_{k_h}(\hat{\mu}_n, \mu)] \; = \; \mathcal{O}\Big(n^{-1/2}\Big).$$

Thus, uniform slicing avoids the curse of dimensionality.

*Sample complexity (max–sliced).* We were not able to establish a sharp sample complexity bound for the max–sliced MMD. Deriving such a result remains an open problem for future work.

# G. Pseudo-codes

---

**Algorithm 2** Estimation of MS$\Delta$ from empirical samples

---

**Input:** empirical samples $X = \{x_i\}_{i=1}^N \subset \mathbb{R}^d$, $Y = \{y_i\}_{i=1}^N \subset \mathbb{R}^d$
**Input:** base 1D divergence $\Delta$; number of gradient steps $T$; step size $\eta$
**Initialize random unit direction**
$w \sim \mathcal{N}(0, I_d)$
$\theta \leftarrow w/\|w\|$
**for** $t = 1$ **to** $T$ **do**
    **Project samples**
    **for** $i = 1$ **to** $N$ **do**
        $u_i \leftarrow \langle \theta, x_i \rangle$
        $v_i \leftarrow \langle \theta, y_i \rangle$
    **end for**
    **Objective along direction**
    $J(\theta) \leftarrow \Delta\big(\{u_i\}_{i=1}^N, \{v_i\}_{i=1}^N\big)$
    **Gradient ascent on direction**
    $g \leftarrow \nabla_\theta J(\theta)$
    $w \leftarrow w + \eta\, g$
    **Re-normalize onto the unit sphere**
    $\theta \leftarrow w/\|w\|$
**end for**
**Freeze optimized direction**
$\bar{\theta} \leftarrow \mathrm{stop\_grad}(\theta)$
**Output**
$\widehat{\mathrm{MS}\Delta}(X, Y) \leftarrow \Delta\big(\{\langle \bar{\theta}, x_i \rangle\}_{i=1}^N, \{\langle \bar{\theta}, y_i \rangle\}_{i=1}^N\big)$

---

# H. Pixel-based control: maze and Atari

We follow the maze environment and Atari subset of Zhang et al. (2021). The maze environment was reimplemented purely in JAX (Bradbury et al., 2018) to improve parallelism. This section collects the design choices shared by both pixel-based setups. Details specific to each are provided in Section H.4 and H.5.

## H.1. From multivariate rewards to a control signal

Our critic models the full multivariate return distribution $Z^\pi(s, a) \in \mathbb{R}^d$. To interface with standard control algorithms, we fix a preference (weight) vector $\alpha \in \mathbb{R}^d$ and derive a scalar action-value by taking the expected return and projecting it:

$$Q_\alpha^\pi(s, a) := \alpha^\top \mathbb{E}[Z^\pi(s, a)].$$

This produces a standard scalar control objective while retaining a multivariate distributional critic (used for the TD updates and richer modeling). We then plug this signal into a standard control algorithm, using PQN in our experiments (Gallici et al., 2024). Other common choices include DQN (Mnih et al., 2013) and actor-critic methods for continuous control (Lillicrap et al., 2015).

## H.2. Baseline algorithm: PQN

**Adapting TD-$\lambda$.** PQN uses TD-$\lambda$ (Sutton, 1988) to build training targets that interpolate between one-step bootstrapping and Monte Carlo returns. Given an episode (or truncated rollout) $(S_t, A_t, R_{t+1})_{t=0}^{T-1}$, a discount $\gamma \in [0, 1)$, and a bootstrap value $V_t := V(S_t)$, define the $n$-step return

$$G_t^{(n)} := \sum_{k=0}^{n-1} \gamma^k R_{t+k+1} + \gamma^n V(S_{t+n}), \qquad n \geq 1. \tag{67}$$

The TD-$\lambda$ target is the geometric mixture of these $n$-step returns,

$$G_t^\lambda := (1 - \lambda) \sum_{n=1}^{T-t-1} \lambda^{n-1} G_t^{(n)} + \lambda^{T-t-1} G_t^{(T-t)}, \qquad \lambda \in [0, 1]. \tag{68}$$

This canonical definition is equivalent to the backward recursion used for implementation:

$$G_t^\lambda = R_{t+1} + \gamma(1 - D_{t+1})\Big((1 - \lambda) V(S_{t+1}) + \lambda G_{t+1}^\lambda\Big), \tag{69}$$

which is computed by a reverse scan over time. Here $D_{t+1} \in \{0, 1\}$ is the done flag, so the factor $(1 - D_{t+1})$ disables bootstrapping after termination.

**Adapting to the distributional setting.** The distributional critic outputs $K$ particles that represent a sample-based approximation of the return distribution. In our case, these $K$ particles are predicted jointly by a neural network and cannot be assumed to be perfectly i.i.d., so their ordering should not be used implicitly when forming bootstrapped targets.

| Environment settings | Values for Maze | Values for Atari |
|---|---|---|
| Stack size | 1 | 4 |
| Frame skip | 1 | 4 |
| One-frame observation shape | $(84, 84, 3)$ | $(84, 84)$ |
| Agent's observation shape | $(84, 84, 3)$ | $(84, 84, 4)$ |
| $\gamma$ | 0.99 | 0.99 |
| Reward clipping | – | false |
| Terminate on Life Loss | – | true |
| Sticky Actions | false | false |

Table 2. Environment settings for pixel-based control experiments, adapted from Zhang et al. (2021). In contrast to Zhang et al. (2021), we do not use reward clipping for Atari.

**Algorithm 3** Distributional multivariate PQN with $\lambda$-returns (adapted from Algorithm 1 in (Gallici et al., 2024)). Here $I$ is the number of parallel environments, $T$ is the rollout length, $K$ is the number of particles, and $\alpha \in \mathbb{R}^d$ is the fixed scalarization weight vector used to define $Q_\phi^\alpha(s, a) := \alpha^\top\big(\frac{1}{K}\sum_{k=1}^K Z_\phi(s,a)_{:,k}\big)$.

---

**Initialise** distributional critic parameters $\phi$ for $Z_\phi : \mathcal{S} \times \mathcal{A} \to \mathbb{R}^{d \times K}$
**Sample** initial states $s_0^i \sim P_0$ for all environments $i \in \{0, \ldots, I-1\}$
$t \leftarrow 0$
**for** each episode **do**
 **for** each $i \in \{0, 1, \ldots, I-1\}$ **(in parallel) do**
  $a_t^i \sim \pi_{\text{Explore}}(s_t^i)$                 (e.g. $\epsilon$-greedy w.r.t. $Q_\phi^\alpha$)
  $r_t^i \sim P_R(s_t^i, a_t^i), \quad s_{t+1}^i \sim P_S(s_t^i, a_t^i), \quad d_{t+1}^i \sim P_D(s_t^i, a_t^i)$
 **end for**
 $t \leftarrow t + 1$
 **if** $t \bmod T = 0$ **then**
  **Compute greedy actions from scalarized values**
  **for** $k = t - T$ **to** $t$ **do**
   $a_k^{\star,i} \leftarrow \arg\max_{a'} Q_\phi^\alpha(s_k^i, a')$
  **end for**
  **Compute $\lambda$-return targets (mixture implementation for particle targets)**
  $G_t^{\lambda,i} \leftarrow Z_\phi(s_t^i, a_t^{\star,i}) \in \mathbb{R}^{d \times K}$
  **for** $k = t - 1$ **down to** $t - T$ **do**
   $G_k^{\lambda,i} \leftarrow r_k^i + \gamma(1 - d_{k+1}^i)\,\text{Mix}_\lambda\big(Z_\phi(s_{k+1}^i, a_{k+1}^{\star,i}), G_{k+1}^{\lambda,i}\big)$
  **end for**
  **for** number of epochs **do**
   **for** number of minibatches **do**
    Sample minibatch $B$ of size $b \leq I{\cdot}T$ from $\{t - T, \ldots, t - 1\} \times \{0, \ldots, I-1\}$
    **Distributional update**
    $\phi \leftarrow \phi - \dfrac{\eta_t}{b}\,\nabla_\phi \sum_{(k,i) \in B} \mathcal{L}_{\text{dist}}\big(Z_\phi(x_k^i), G_k^{\lambda,i}\big)$
   **end for**
  **end for**
 **end if**
**end for**

---

In Q-learning, TD-$\lambda$ combines the one-step bootstrap and the continuation through a convex combination via (69). For distributional return learning, we reinterpret the same recursion at the level of return laws:

$$G_t^\lambda \;\overset{\mathcal{D}}{=}\; R_{t+1} + \gamma(1 - D_{t+1})\Big((1 - \lambda)\,Z_{t+1} + \lambda\,G_{t+1}^\lambda\Big), \qquad Z_{t+1} := Z(S_{t+1}, a^{\text{policy}}). \tag{70}$$

When approximating these laws by particles, we implement the convex combination as a *mixture* over the two components. Let $\{z_{t+1}^{(k)}\}_{k=1}^K$ denote the particles predicted at $(S_{t+1}, a^{\text{policy}})$ and $\{G_{t+1}^{\lambda,(k)}\}_{k=1}^K$ the particles of the continuation target. For each transition and each $k$, sample $b_t^{(k)} \sim \text{Bernoulli}(\lambda)$ and set

$$c_t^{(k)} := \underbrace{\mathbf{1}\{b_t^{(k)} = 0\}\,\tilde{z}_{t+1}^{(k)}}_{\text{one-step bootstrap}} + \underbrace{\mathbf{1}\{b_t^{(k)} = 1\}\,\tilde{G}_{t+1}^{\lambda,(k)}}_{\text{continuation target}}. \tag{71}$$

Second, before applying this mixture we independently *shuffle* the $K$ particles within each of the two sets $\{z_{t+1}^{(k)}\}_{k=1}^K$ and $\{G_{t+1}^{\lambda,(k)}\}_{k=1}^K$, and then define $\tilde{z}_{t+1}^{(k)}$ and $\tilde{G}_{t+1}^{\lambda,(k)}$ as the particles obtained after shuffling. This breaks any systematic index pairing while preserving the empirical distribution of each set.

| Component | Specification used in our experiments |
|---|---|
| Input | $84 \times 84$ pixel observation (stacked as in Table 2) |
| Preprocess | divide by 255 |
| Conv1 | 32 channels, $8 \times 8$ kernel, stride 4, VALID |
| Norm & act. | LayerNorm + SiLU |
| Conv2 | 64 channels, $4 \times 4$ kernel, stride 2, VALID |
| Norm & act. | LayerNorm + SiLU |
| Conv3 | 64 channels, $3 \times 3$ kernel, stride 1, VALID |
| Norm & act. | LayerNorm + SiLU |
| Flatten | reshape to a vector |
| FC | Dense(512) |
| Norm & act. | LayerNorm + SiLU |
| Output head | Dense($A \cdot K \cdot d$), reshaped into $K$ particles per action in $\mathbb{R}^d$ |

*Table 3.* Neural network architecture used in our pixel-based control experiments. We follow Gallici et al. (2024) for the CNN trunk and LayerNorm placement, replace ReLU with SiLU, and modify the final layer to predict $K$ particles per action.

| Hyper-parameters | Values |
|---|---|
| Number of frames | $5 \times 2^6$ |
| Number of environments | 256 |
| Rollout length (steps per environment, per update) | 1 |
| Number of epochs (per update) | 2 |
| Number of minibatches (per epoch) | 32 |
| TD-$\lambda$ | 0.0 |
| Learning rate | $1.0 \times 10^{-5}$ |
| Optimizer | RAdam (Liu et al., 2019) |
| Number of particles | 200 |
| Slicing: number of projection directions | 64 |
| Multiquadric kernel bandwidth | 100.0 |
| Evaluation : number of Monte Carlo rollouts | 16 |

*Table 4.* Default hyper-parameter settings used for the maze experiments.

## H.3. Experimental setup

We base most of our implementation on PQN (Gallici et al., 2024). The distributional extension used in this work is summarized in Algorithm 3. Environment parameters shared by our pixel-based setups are reported in Table 2, and the default hyper-parameters for the training setup are reported in Table 5. As discussed in Section F, we use the powered sliced objectives for all base metrics.

**Neural network architecture.** We keep most of the Q-network of Gallici et al. (2024) and apply two changes: we replace ReLU with SiLU, and we change the final linear layer to output $K$ particles per action. The resulting architecture is summarized in Table 3.

## H.4. Maze

We faithfully reimplemented the maze environments introduced by Zhang et al. (2021). The environments were rewritten entirely in JAX (Bradbury et al., 2018), including both the environment dynamics and the rendering pipeline. This choice was made to ensure full compatibility with PQN, which is itself implemented in JAX, and to support large-scale parallel execution with 256 environments in parallel. Unless stated otherwise, the training settings for the maze experiments are reported in Table 4.

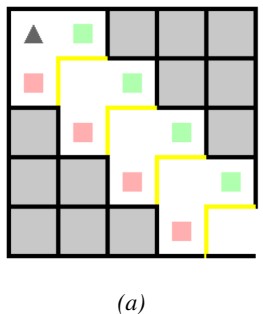
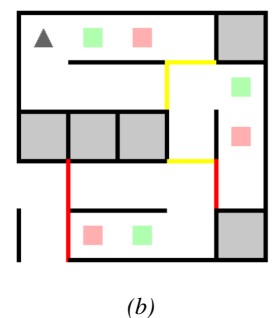
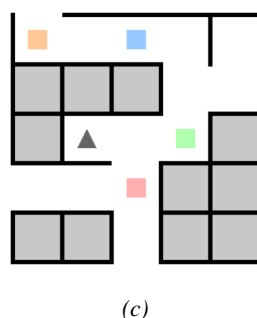

*(a)*  *(b)*  *(c)*

*Figure 6.* Maze environments reproduced from (Zhang et al., 2021). Initial observations are shown for three variants: (a) MAZE-EXCLUSIVE, with two mutually exclusive reward sources; (b) MAZE-IDENTICAL, with two positively correlated reward sources; and (c) MAZE-MULTIREWARD, with four correlated reward sources.

**Environment.** All maze variants are fixed grid environments with deterministic movement dynamics and pixel observations. The agent observation is an $84 \times 84 \times 3$ RGB rendering of the environment. This observation fully specifies the current environment configuration relevant to control, including the maze layout, the agent position, and the reward locations. The agent occupies a single cell in the grid and can take one of four discrete actions corresponding to moving up, down, left, or right. If the target cell is free, the agent moves to that cell, otherwise the action has no effect and the agent remains in its current position. Moves also have no effect when the agent is trying to return to a previous position. Colored walls can only be passed in one direction. Yellow walls can be passed from the top or from the left. Red walls can be passed from the bottom or from the right. Episodes start from a fixed initial position shared across all maze variants and run for a fixed horizon. At each step, the environment emits a vector-valued reward.

Reward locations are indicated by colored squares in the maze. Each color corresponds to one reward component. When the agent reaches such a square, it receives a stochastic scalar reward in the corresponding component and the reward cannot be collected again within the same episode. The number of reward components and their distributions depend on the maze variant described next.

**Maze variants and reward functions.** Figure 6 illustrates the three maze variants and the placement of colored reward sources. The reward dimensionality is $d = 2$ for MAZE-EXCLUSIVE and MAZE-IDENTICAL, and $d = 4$ for MAZE-MULTIREWARD. The reward distributions follow (Zhang et al., 2021, Appendix A.2).

- **MAZE-EXCLUSIVE and MAZE-IDENTICAL ($d = 2$).** Each position with a green square yields a reward $r \sim \mathcal{U}(0.2, 0.6)$ in the first reward component. Each position with a red square yields a reward $r \sim \mathcal{U}(0.4, 0.9)$ in the second reward component.

- **MAZE-MULTIREWARD ($d = 4$).** Each orange square yields a reward $r \sim \mathcal{U}(0.2, 0.4)$ in the first reward component. Each blue square yields a reward $r \sim \mathcal{U}(0.8, 1.0)$ in the second reward component. Each green square yields a reward $r \sim \mathcal{U}(0.3, 0.5)$ in the third reward component. Each red square yields a reward $r \sim \mathcal{U}(0.5, 0.7)$ in the fourth reward component.

**Evaluation protocol.** To evaluate the learned return distributions, we generate Monte Carlo (MC) rollouts from the reset distribution $S_0 \sim P_0$ using a uniform random policy $\pi(a \mid s) = \frac{1}{|\mathcal{A}|}$, so the rollouts capture both environment stochasticity and policy stochasticity. For a fixed horizon $H$, each rollout yields a discounted return

$$G_0^{(n)} := \sum_{t=0}^{H-1} \gamma^t R_{t+1}^{(n)} \in \mathbb{R}^d, \qquad \widehat{\mu}_{\mathrm{MC}} := \frac{1}{N} \sum_{n=1}^{N} \delta_{G_0^{(n)}}.$$

We then query the distributional critic at the start state for all actions and aggregate its predicted particles using the same uniform policy $\pi$, yielding an empirical critic-implied return distribution $\widehat{\mu}_\phi$. Finally, we compare the two empirical distributions using the exact optimal-transport estimator $\widehat{\mathbf{W}}_2(\widehat{\mu}_{\mathrm{MC}}, \widehat{\mu}_\phi)$ computed with POT (Flamary et al., 2021).

**H.5. Atari**

**Reward decomposition.**  We reproduce the Atari multi-reward setup of (Zhang et al., 2021, Appendix A.2). Concretely, we consider six environments, AIRRAID, ASTEROIDS, GOPHER, MSPACMAN, UPNDOWN, and PONG, and decompose each primitive reward into a fixed-dimensional vector while preserving the total reward (the sum of components equals the original scalar reward). We list the exact mappings below.

- **AirRaid.** Primitive rewards take values in $\{100, 75, 50, 25, 0\}$ and are mapped to

$$100 \mapsto [100, 0, 0, 0], \quad 75 \mapsto [0, 75, 0, 0], \quad 50 \mapsto [0, 0, 50, 0], \quad 25 \mapsto [0, 0, 0, 25], \quad 0 \mapsto [0, 0, 0, 0].$$

- **Pong.**

$$-1 \mapsto [-1, 0], \qquad +1 \mapsto [0, 1], \qquad 0 \mapsto [0, 0].$$

- **Asteroids.** Let $r$ be the primitive reward and output $[r_1, r_2, r_3]$ defined by

$$r_1 = \begin{cases} 20, & (r - 20) \bmod 50 = 0, \\ 0, & \text{otherwise}, \end{cases} \qquad r_2 = \begin{cases} 50, & (r - r_1 - 50) \bmod 100 = 0, \\ 0, & \text{otherwise}, \end{cases} \qquad r_3 = r - r_1 - r_2.$$

- **MsPacman.** Let $r$ be the primitive reward and output $[r_1, r_2, r_3, r_4]$ defined by

$$r_1 = \begin{cases} 10, & (r - 10) \bmod 50 = 0, \\ 0, & \text{otherwise}, \end{cases} \qquad r_2 = \begin{cases} 50, & (r - r_1 - 50) \bmod 100 = 0, \\ 0, & \text{otherwise}, \end{cases}$$

$$r_3 = \begin{cases} 100, & (r - r_1 - r_2 - 100) \bmod 200 = 0, \\ 0, & \text{otherwise}, \end{cases} \qquad r_4 = r - r_1 - r_2 - r_3.$$

- **Gopher.** Let $r$ be the primitive reward and output $[r_1, r_2]$ defined by

$$r_1 = \begin{cases} 20, & (r - 20) \bmod 100 = 0, \\ 0, & \text{otherwise}, \end{cases} \qquad r_2 = r - r_1.$$

- **UpNDown.** Let $r$ be the primitive reward and output $[r_1, r_2, r_3]$ defined by

$$r_1 = \begin{cases} 10, & (r - 10) \bmod 100 = 0, \\ 0, & \text{otherwise}, \end{cases} \qquad r_2 = \begin{cases} 100, & (r - r_1 - 100) \bmod 200 = 0, \\ 0, & \text{otherwise}, \end{cases} \qquad r_3 = r - r_1 - r_2.$$

**Scalarization rule.**  Following Zhang et al. (2021), we derive a scalar control signal from the multi-dimensional return by simple summation of its components. Concretely, letting $Z(s, a) \in \mathbb{R}^d$ denote the vector-valued return, we define the scalar action-value used for control as

$$Q(s, a) = \mathbb{E}\left[\mathbf{1}^\top Z(s, a)\right],$$

where $\mathbf{1} \in \mathbb{R}^d$ is the all-ones vector.

**Evaluation protocol.**  Alongside the training environments, we run a separate pool of $8$ evaluation environments in parallel, using the same evaluation implementation as in the public codebase of Gallici et al. (2024). During evaluation, the agent follows a purely greedy policy (no exploration noise). This design choice is deliberate for performance: evaluation runs continuously without interrupting training. We additionally report *normalized scores* to aggregate results across environments. We compute

$$\text{Norm}(\pi) := 100 \times \frac{J(\pi) - J(\pi_{\text{rand}})}{J(\pi_{\text{ref}}) - J(\pi_{\text{rand}})},$$

where $J(\cdot)$ is the undiscounted evaluation return. The constants $J(\pi_{\text{rand}})$ and $J(\pi_{\text{ref}})$ are the random and reference scores used for normalization. We do not interpret these as *human-normalized* scores because our Atari setup does not use sticky actions (we follow (Zhang et al., 2021)). We therefore refer to them simply as *normalized scores*.

| Hyper-parameters | Default values |
|---|---|
| Number of frames | $5 \times 10^7$ |
| Number of environments | 128 |
| Rollout length (steps per environment, per update) | 32 |
| Number of epochs (per update) | 2 |
| Number of minibatches (per epoch) | 32 |
| TD-$\lambda$ | 0.65 |
| Learning rate | $2.5 \times 10^{-4}$ |
| Optimizer | RAdam (Liu et al., 2019) |
| Gradient-norm clipping | 10.0 |
| $\epsilon$-start | 1.0 |
| $\epsilon$-finish | 0.001 |
| $\epsilon$-decay ratio (of total updates) | 0.1 |
| Number of particles | 200 |
| Slicing: number of projection directions | 128 |
| Multiquadric kernel bandwidth | 100.0 |

*Table 5.* Default hyper-parameter settings used in our pixel-based control experiments.

# I. Chain environment

Section I.3 describes the standard one-sampled distributional TD regime (Algorithm 4). Section I.4 describes the near-exact variant that explicitly constructs the transition-mixture Bellman target (Algorithm 5). Unless stated otherwise, distributional accuracy is evaluated at the start state using the same Monte Carlo protocol as for Maze (Section H.4).

## I.1. Environment description

We reproduce and generalize the chain environment of Rowland et al. (2019a) as used by Nguyen-Tang et al. (2021). *This setup is used purely for policy evaluation.* The MDP has states $\{s_0, \ldots, s_{K-1}\}$ with initial state $s_0$ and terminal (absorbing) state $s_{K-1}$. A schematic is provided in Fig. 1. For any nonterminal state $s_i$ with $i \le K-2$ and action $a \in \{\texttt{fwd}, \texttt{bwd}\}$, the next state satisfies

$$S_{t+1} = \begin{cases} s_{i+1}, & \text{with probability } p(a), \\ s_0, & \text{with probability } 1 - p(a), \end{cases} \qquad p(\texttt{fwd}) = 0.9, \;\; p(\texttt{bwd}) = 0.1,$$

and if $S_t = s_{K-1}$ then $S_{t+1} = s_{K-1}$. Episodes terminate upon first entering $s_{K-1}$, and we do not emit further rewards nor bootstrap beyond termination. In the original scalar chain, the reward is $+1$ upon entering the terminal state, $-1$ on transitions that reset to $s_0$, and 0 otherwise.

In our multivariate variant, the transition dynamics are unchanged and only the reward is modified. Here the return dimension is $d = K$. We represent each state $s_i$ by a one-hot vector $e_i \in \mathbb{R}^K$ and emit the positional reward vector $R_{t+1} = e_{S_{t+1}}$, which is the one-hot embedding of the entered next state. This yields a $K$-dimensional return that tracks discounted state occupancies in the spirit of successor features (Barreto et al., 2017) while preserving the original chain's stochasticity. In our experiments we use a scalar discount $\gamma$ and instantiate the matrix discount as $\Gamma = \gamma I_d$.

## I.2. Empirical setup

To model the return distribution we follow Nguyen-Tang et al. (2021) and, for each state–action pair, parametrize an empirical distribution with a fixed set of particles. Since the chain is tabular and small, we do not use a neural network. We initialize the particles randomly and update them using a distributional divergence, as described in the two subsequent sections. Default hyper-parameters are reported in Table 6.

**Evaluation protocol.** We follow the same Monte Carlo evaluation protocol as for the Maze environments (Section H.4). Concretely, we generate Monte Carlo (MC) rollouts starting from the initial state $S_0 = s_0$ and unroll the evaluation policy ($\pi(s) \equiv \texttt{fwd}$) until termination at $s_{K-1}$. Each rollout yields a discounted return vector, and repeating this procedure

produces an empirical return distribution at the start state. We then query the distributional critic at $(s_0, \texttt{fwd})$ to obtain its predicted particle distribution and compare the two, using the same distributional metric as in the Maze evaluation.

### I.3. One-sampled distributional TD

In distributional RL, the (population) Bellman target at $(s, a)$ is a mixture law induced by transition and policy randomness. In most implementations, however, TD bootstrapping replaces this mixture by a one-sample instantiation based on a single sampled successor $(S_{t+1}, A_{t+1})$. This is the setting discussed in Sec. 4.4.

We begin with this standard *one-sampled* distributional TD bootstrapping regime in our tabular chain setting. The purpose of this subsection is only to specify the target distribution produced by a single sampled successor, which we will contrast in the next subsection with a closer approximation of the full (mixture) distributional Bellman target.

For clarity, we describe this mode for the evaluation policy that always goes right, i.e. $\pi(s) \equiv \texttt{fwd}$ for all nonterminal states. At each update, we only modify the particles at the *current* tabular entry $(S_t, \texttt{fwd})$ using a single observed transition.

Consider one on-policy transition $(S_t, \texttt{fwd}, R_{t+1}, S_{t+1})$, where $S_{t+1} \sim P(\cdot \mid S_t, \texttt{fwd})$, and take $A_{t+1} = \texttt{fwd}$. We form a one-sample distributional target by bootstrapping from the single successor pair $(S_{t+1}, \texttt{fwd})$:

$$\hat{Z}_{t+1} \;:=\; R_{t+1} \;+\; \Gamma\, Z_\phi(S_{t+1}, \texttt{fwd}),$$

where $\Gamma \in \mathbb{R}^{d \times d}$ is the (possibly matrix) discount. In our particle implementation, $\hat{Z}_{t+1}$ is represented by the target particle set

$$\hat{z}_{t+1}^{(n)} \;:=\; R_{t+1} \;+\; \Gamma\, z_\phi^{(n)}(S_{t+1}, \texttt{fwd}), \qquad n = 1, \ldots, N,$$

and no gradient flows through the bootstrap particles $z_\phi^{(n)}(S_{t+1}, \texttt{fwd})$ when updating the particles at $(S_t, \texttt{fwd})$. The TD update at time $t$ then matches the empirical distribution at the current tabular entry $(S_t, \texttt{fwd})$ to the empirical distribution defined by $\{\hat{z}_{t+1}^{(n)}\}_{n=1}^N$ using a distributional divergence.

### I.4. Near exact distributional TD

Because the chain dynamics are known and finite, the (population) distributional Bellman target can be approximated directly, rather than instantiated from a single sampled successor as in Sec. I.3. We exploit the fact that, for each state–action pair $(s, a)$, the next state has only two possible outcomes, so the mixture induced by the transition kernel can be constructed explicitly and represented with a fixed particle budget.

For clarity, we describe this mode for the evaluation policy that always goes right, i.e. $\pi(s) \equiv \texttt{fwd}$ for all nonterminal states. In this regime, all nonterminal states are updated simultaneously in a batch sweep.

Fix a nonterminal state $s = s_i$ with $i \leq K - 2$ and take $a = \texttt{fwd}$. Then the next state is $s_{\text{succ}} = s_{i+1}$ with probability $p = 0.9$ and $s_{\text{fail}} = s_0$ with probability $1 - p = 0.1$. Let $R(s, s') \in \mathbb{R}^d$ denote the (vector) reward emitted on transition $s \to s'$. We form the two successor-backed-up target distributions

$$Z_{\text{succ}} \;:=\; R(s, s_{\text{succ}}) \;+\; \Gamma\, Z_\phi(s_{\text{succ}}, \texttt{fwd}), \qquad Z_{\text{fail}} \;:=\; R(s, s_{\text{fail}}) \;+\; \Gamma\, Z_\phi(s_{\text{fail}}, \texttt{fwd}),$$

| Hyper-parameters | Default values |
|---|---|
| Optimizer | RAdam (Liu et al., 2019) |
| Learning rate | $2.5 \times 10^{-3}$ |
| Discount factor $\gamma$ | 0.9 |
| Number of particles $N$ | 500 |
| Gradient-norm clipping | 0.1 |
| Chain forward success probability $p(\texttt{fwd})$ | 0.9 |
| Chain length | 10 |
| Particle initialization | $\mathcal{N}(-1, 0.08^2)$ |

*Table 6.* Default hyper-parameter settings used in our tabular chain policy-evaluation experiments.

**Algorithm 4** One-sampled distributional TD (`fwd`-only policy).

> **Initialise** tabular particles $\{z_\phi^{(n)}(s, \texttt{fwd})\}_{n=1}^N$ for all $s \in \{s_0, \ldots, s_{K-1}\}$
> Fix evaluation policy $\pi(s) \equiv \texttt{fwd}$ and discount matrix $\Gamma \in \mathbb{R}^{d \times d}$
> Sample initial state $S_0 \sim P_0$
> **for** $u = 0, 1, \ldots, U-1$ **do**
>     Observe $(S_{u+1}, R_{u+1}) \sim P(\cdot, \cdot \mid S_u, \texttt{fwd})$
>     *Form target particles*
>     **for** $n = 1, 2, \ldots, N$ **do**
>         $\hat{z}^{(n)} \leftarrow R_{u+1} + \Gamma\, \mathbf{sg}(z_\phi^{(n)}(S_{u+1}, \texttt{fwd}))$
>     **end for**
>     $g_u \leftarrow \nabla_\phi \mathcal{D}\Big(Z_\phi(S_u, \texttt{fwd}), \frac{1}{N}\sum_{n=1}^N \delta_{\hat{z}^{(n)}}\Big)$
>     $\phi \leftarrow \phi - \eta_u\, g_u$
>     $S_u \leftarrow S_{u+1}$
> **end for**

**Algorithm 5** Near-exact distributional TD (`fwd`-only policy).

> **Initialise** tabular particles $\{z_\phi^{(n)}(s, \texttt{fwd})\}_{n=1}^N$ for all $s \in \{s_0, \ldots, s_{K-2}\}$
> Fix evaluation policy $\pi(s) \equiv \texttt{fwd}$ and discount matrix $\Gamma \in \mathbb{R}^{d \times d}$
> **for** $u = 1, 2, \ldots, U$ **do**
>     **Construct mixture targets for all states (in parallel)**
>     **for each** $s = s_i, i \in \{0, \ldots, K-2\}$ **do**
>         $s_{\text{succ}} \leftarrow s_{i+1}, \; s_{\text{fail}} \leftarrow s_0, \; p \leftarrow 0.9$
>         $T_{\text{succ}} \leftarrow R(s, s_{\text{succ}}) + \Gamma\, \mathbf{sg}(Z_\phi(s_{\text{succ}}, \texttt{fwd}))$
>         $T_{\text{fail}} \leftarrow R(s, s_{\text{fail}}) + \Gamma\, \mathbf{sg}(Z_\phi(s_{\text{fail}}, \texttt{fwd}))$
>         $\{\hat{z}^{(n)}(s)\}_{n=1}^N \leftarrow \text{Resample}\Big(p\, T_{\text{succ}} + (1-p)\, T_{\text{fail}}\Big)$
>     **end for**
>     $g_u \leftarrow \nabla_\phi \sum_{i=0}^{K-2} \mathcal{D}\Big(Z_\phi(s_i, \texttt{fwd}), \frac{1}{N}\sum_{n=1}^N \delta_{\hat{z}^{(n)}(s_i)}\Big)$
>     $\phi \leftarrow \phi - \eta_u\, g_u$
> **end for**

*Figure 7.* Two tabular policy-evaluation regimes for the chain environment under the `fwd`-only policy. Left: standard one-sampled bootstrapping from a single observed successor. Right: a near-exact approximation of the mixture Bellman target obtained by explicitly constructing the two-outcome transition mixture for every state. Here $\mathbf{sg}(\cdot)$ denotes a stop-gradient operator.

and define the corresponding mixture target

$$Z_{\text{targ}} := p\, Z_{\text{succ}} + (1-p)\, Z_{\text{fail}}.$$

In our particle implementation, $Z_{\text{targ}}$ is represented by drawing $N$ target particles from this mixture: we pool the particles from $Z_{\text{succ}}$ and $Z_{\text{fail}}$ and resample $N$ of them with mixture weights $p$ and $1-p$. As in Sec. I.3, no gradient flows through the bootstrap particles when updating the particles at $(s, \texttt{fwd})$.

This yields a close approximation of the full mixture Bellman target while keeping the update fully tabular and using a fixed particle budget per state.

## J. More results and ablations

This section collects additional results and diagnostics that complement the main experiments. Section J.1 reports update-time scaling with the number of particles. Section J.2 provides additional chain results, including a max-slicing sanity check and Monte Carlo versus predicted return visualizations. Section J.3 reports a Maze ablation on the choice of Cramér estimator, together with Monte Carlo versus predicted return visualizations. Section J.4 reports per-environment Atari learning curves and studies the effect of the number of projection directions.

### J.1. Computational cost

In Figure 8 we compare the wall-clock time of a single update iteration of Algorithm 5 across the objectives considered in this work, while varying the number of particles used to model the return distribution. As expected, MMD and sliced-MMD exhibit a much steeper increase in runtime as the number of particles grows, which is consistent with the quadratic complexity of standard MMD estimators ($\mathcal{O}(n^2)$). In contrast, the other sliced variants scale more mildly with the number of particles, which is consistent with a near $\mathcal{O}(n \log n)$ cost when their one-dimensional base computations admit efficient sorting-based implementations.

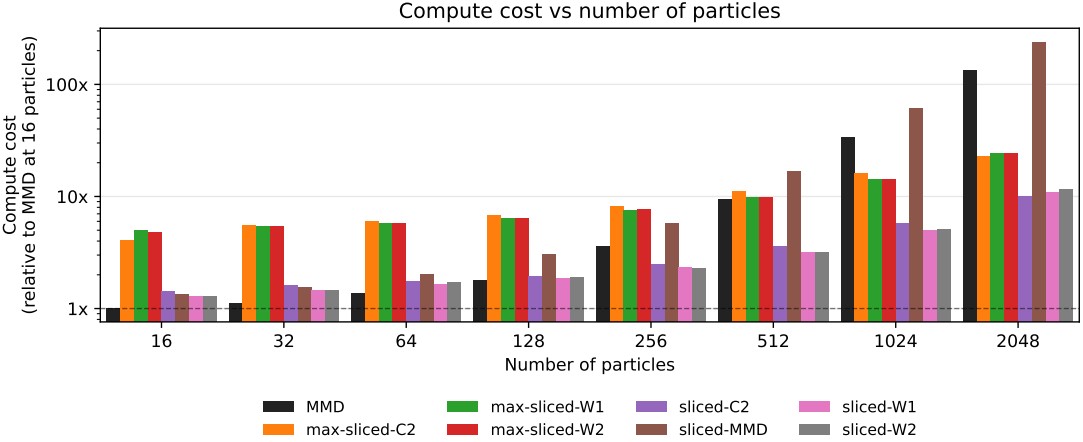

*Figure 8.* Wall-clock time per update as a function of the number of particles, normalized to MMD at the smallest particle count.

## J.2. Chain environment

### J.2.1. TESTING GRADIENT BIAS FROM MAX SLICING

We now isolate the selection bias induced by max slicing by considering a chain setting in which the multivariate return distribution is effectively one-dimensional. We do so by enforcing that all rewards are colinear: there exists a fixed direction $u \in \mathbb{R}^d$ such that

$$R_{t+1} = r_{t+1}\, u \qquad \text{for all transitions,}$$

for some scalar reward process $(r_t)_{t\geq 0}$. In our experiments, we take $u = \mathbf{1}_d$ and replicate the scalar chain reward across all $d$ coordinates, which is a direct generalization of the scalar chain setting of Nguyen-Tang et al. (2021).

**Consequence for max slicing.** When the return distribution is supported on the one-dimensional subspace $\mathrm{span}\{u\}$, the max-sliced objective reduces to a constant multiple of a one-dimensional divergence and the selection mechanism behind Lemma D.13 simply doesn't apply. Indeed, since $Z = \sum_{t\geq 0} \gamma^t R_{t+1} = Y\, u$ for some scalar random variable $Y$, for any slicing direction $\theta \in S^{d-1}$ we have

$$\langle \theta, Z \rangle = \langle \theta, u \rangle\, Y.$$

Thus the projected distributions are scaled versions of the same scalar law. For the divergences considered in this work, this scaling induces a deterministic multiplicative factor: for Wasserstein, the standard homogeneity property gives $\mathbf{W}_p((S_s)_{\#}\mu, (S_s)_{\#}\nu) = s\, \mathbf{W}_p(\mu, \nu)$, and for Cramér we have $\mathbf{C}_2((S_s)_{\#}\mu, (S_s)_{\#}\nu) = s\, \mathbf{C}_2(\mu, \nu)$, shown in Proposition A.4. Consequently, for all $\theta$,

$$\Delta\big((P_\theta)_{\#}\mu,\, (P_\theta)_{\#}\nu\big) = |\langle \theta, u \rangle|\, \Delta\big(\mu^{(1)}, \nu^{(1)}\big),$$

where $\mu^{(1)}, \nu^{(1)}$ are the corresponding scalar return distributions and $\Delta \in \{\mathbf{W}_p, \mathbf{C}_2\}$. Taking the supremum over $\theta$ gives

$$\sup_{\theta \in S^{d-1}} \Delta\big((P_\theta)_{\#}\mu,\, (P_\theta)_{\#}\nu\big) = \|u\|_2\, \Delta\big(\mu^{(1)}, \nu^{(1)}\big),$$

which depends only on $u$ and not on the sample used to form the empirical sliced losses.

**Empirical verification.** Figure 9 reports, for each objective, the empirical Wasserstein–2 distance between Monte Carlo returns and the learned critic distribution at the initial state, averaged over 5 random seeds (with 95% confidence intervals). In this degenerate setting, max-sliced-$\mathbf{C}_2$ performs on par with the objectives that satisfy (U), consistent with the fact that the max-slicing selection mechanism (and the associated gradient bias) does not apply here.

### J.2.2. MONTE CARLO RETURNS VS PREDICTED RETURNS

Figure 10 provides a qualitative companion to the chain results reported in Section 5 (Fig. 2). For the same evaluation setting, we directly visualize the learned return particles at the initial state alongside Monte Carlo rollouts from the true

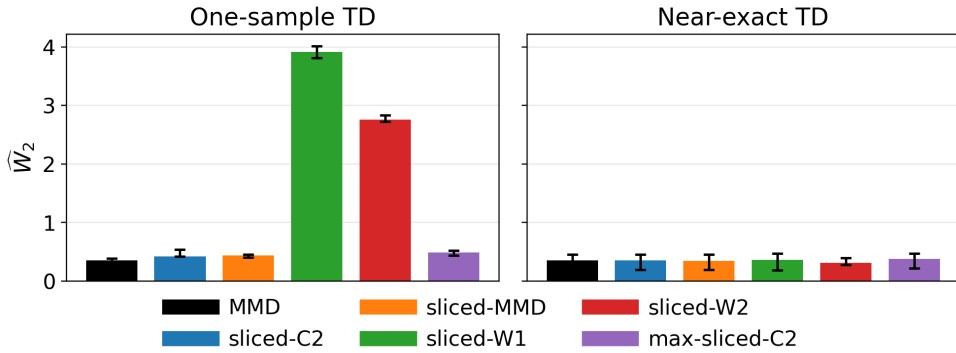

*Figure 9.* Empirical Wasserstein–2 distance between predicted and Monte Carlo return distributions in the chain environment with identical reward coordinates. The reward is zero everywhere except at the terminal state, yielding an effectively one-dimensional return distribution. Bars report the median $\widehat{W}_2$ across 5 random seeds (initial state); error bars show $95\%$ bootstrap CIs (10k resamples). Left: one-sampled distributional TD. Right: near-exact distributional TD.

environment. Notably, although the Wasserstein–2 objective can collapse to a degenerate return distribution in this setting, its predicted mean remains qualitatively comparable to that of objectives that fit the distribution well.

### J.3. Maze

#### J.3.1. CHOICE OF CRAMÉR ESTIMATOR

As discussed in Section A.2.1, the Cramér–2 objective admits multiple empirical estimators. In addition to our default estimator, we further tested the CRPS-based alternative while keeping the rest of the training and evaluation protocol unchanged. Figure 11 shows that this choice has no noticeable effect on distributional matching accuracy across the three environments considered, and does not change the conclusions drawn from the main maze results.

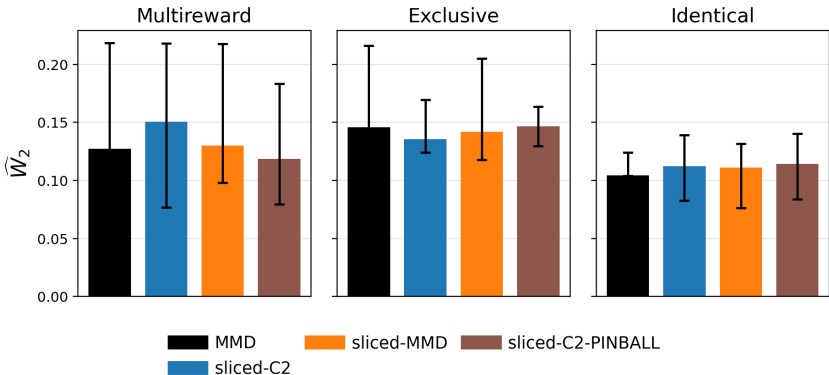

*Figure 11.* Effect of the Cramér–2 empirical estimator choice on distributional matching accuracy in three maze environments. We compare the default Cramér estimator to the CRPS-based alternative (Section A.2.1) while keeping the training pipeline fixed, including the same bootstrap resampling scheme. Bars report the median metric value across 5 random seeds (initial state); error bars show $95\%$ bootstrap CIs (10k resamples). Across all three environments, the estimator choice has no noticeable impact.

J.3.2. MONTE CARLO RETURNS VS PREDICTED RETURNS

Figure 12 provides a qualitative companion to the Maze policy-evaluation results reported in Section 5 (Fig. 3). For each of the three Maze variants, we visualize the critic's learned return particles at a fixed reference state alongside Monte Carlo return samples from the true environment, plotting two return coordinates (dimensions 0 and 1). Blue points denote the particles predicted by the critic, while orange points are Monte Carlo returns. The figure illustrates that objectives satisfying (U) (notably sliced Cramér and sliced-MMD) closely match the Monte Carlo return distribution, whereas objectives that violate (U) can exhibit visible distributional mismatch.

## J.4. Atari

J.4.1. NUMBER OF PROJECTION DIRECTIONS

Figure 13 complements the aggregated normalized-score results in Figure 5 by showing per-environment learning curves for the same runs. We vary the number of projection directions used by the sliced objectives and visualize how this choice affects raw control performance (undiscounted evaluation returns) over training. In this experiment, performance is remarkably robust to the number of projection directions, with strong results already obtained using a single projection direction.

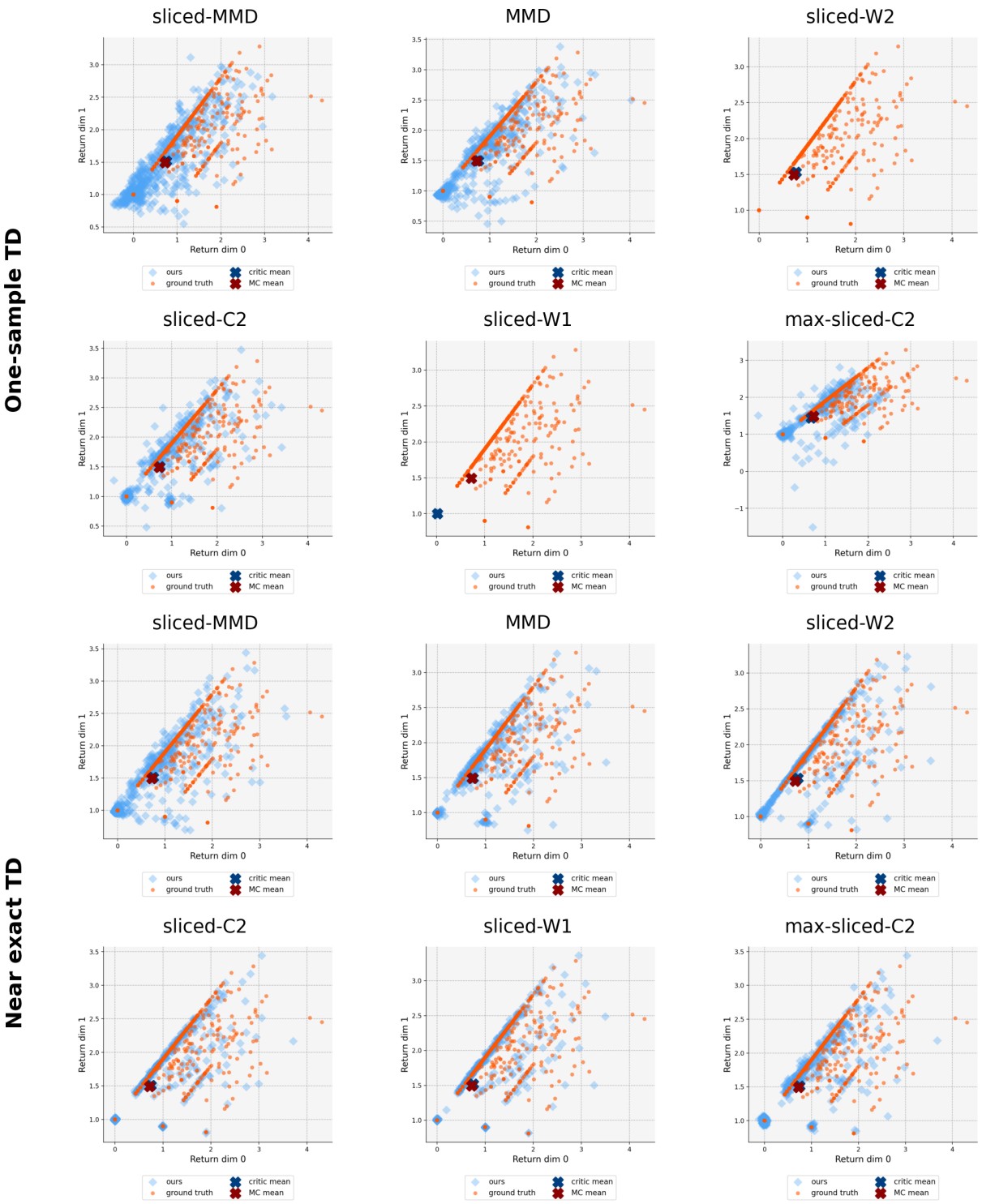

*Figure 10.* Comparison between learned and ground-truth return distributions in the multivariate chain environment. Each panel shows the joint distribution of two return coordinates (dimensions 0 and 1) at the initial state (the start of the chain) under the `fwd-only` evaluation policy. Orange dots correspond to Monte Carlo rollouts approximating the true distributional Bellman target, while blue dots are the particles learned by distributional TD using the indicated objective. Rows correspond to the two training regimes described in Section I: (top) standard one-sampled distributional TD (Algorithm 4), and (bottom) near-exact distributional TD using an explicit mixture construction (Algorithm 5). Markers indicate the empirical mean of the learned critic distribution and the Monte Carlo mean.

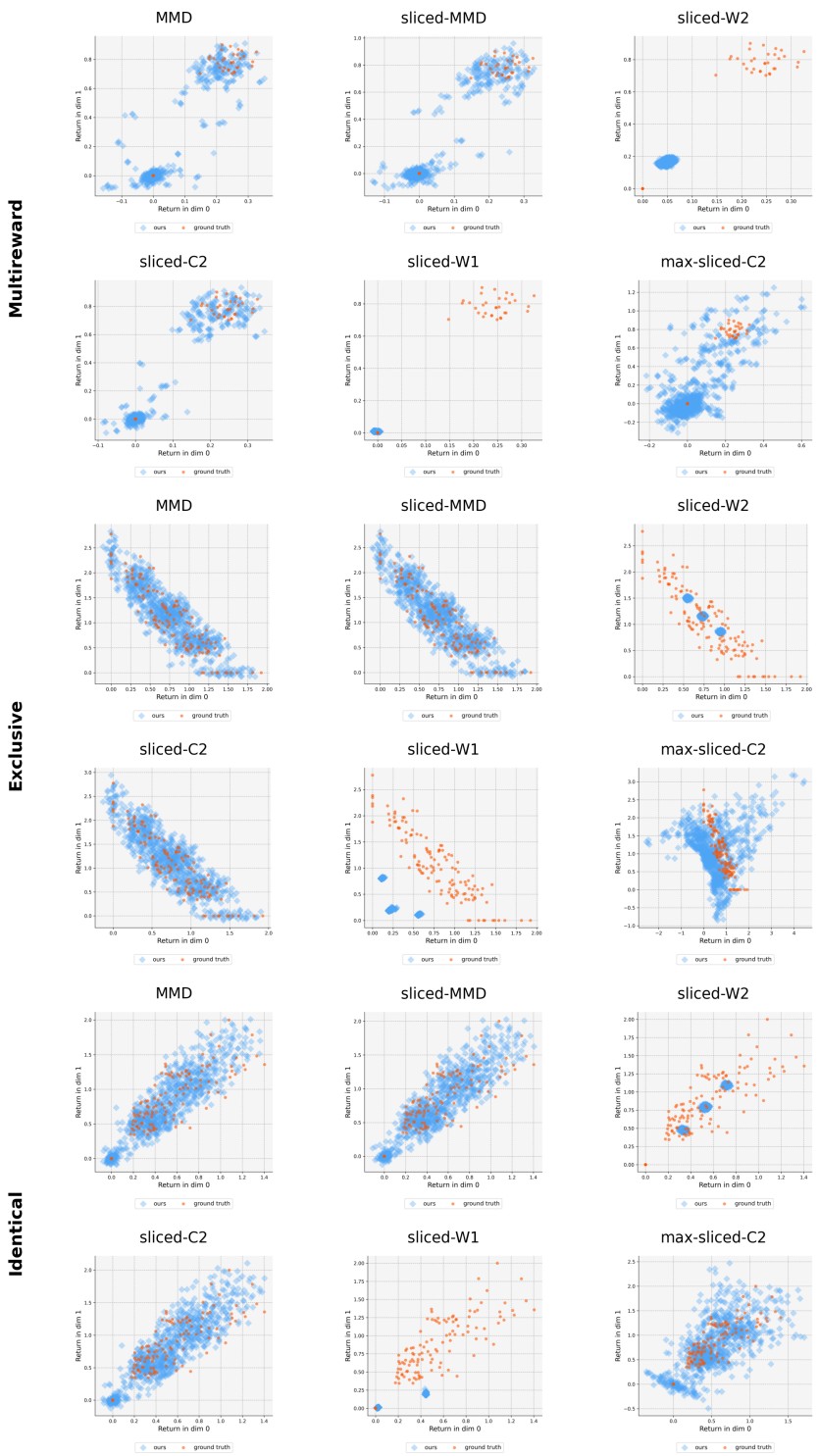

*Figure 12.* Monte Carlo returns versus critic-predicted return particles in the Maze environments. Each panel shows the joint distribution of two return coordinates (dimensions 0 and 1) at the start state, under the uniform random policy used for evaluation. Orange dots are Monte Carlo discounted returns obtained from rollouts of the true environment, while blue dots are the critic-implied particles aggregated across actions using the same policy. Rows correspond to the three Maze variants (MAZE-MULTIREWARD, MAZE-EXCLUSIVE, and MAZE-IDENTICAL), and columns to the distributional objectives used to train the critic.

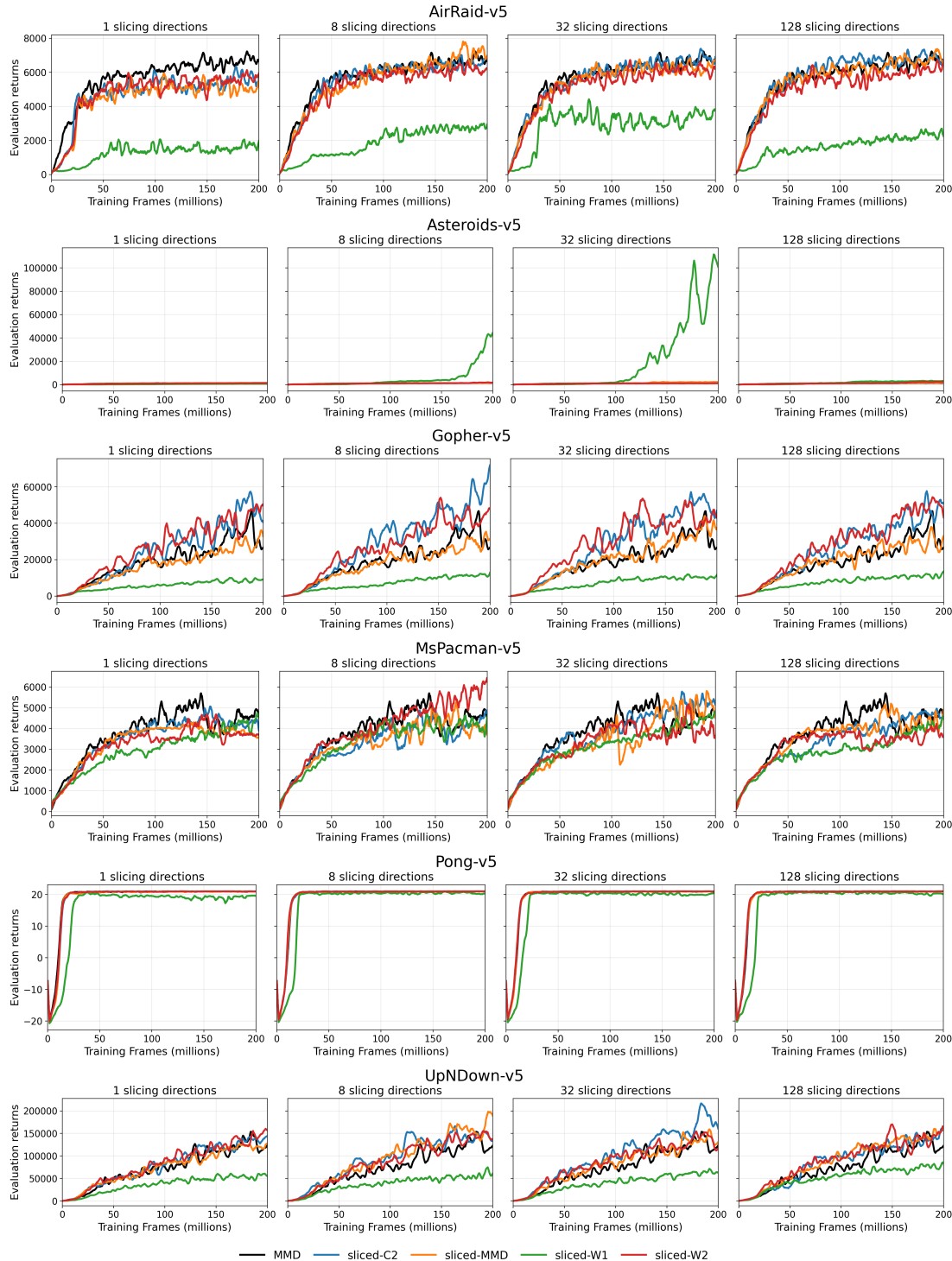

*Figure 13.* Learning curves on six Atari environments with decomposed reward signals. All methods are evaluated using the same protocol as in Gallici et al. (2024), following a greedy policy during evaluation and running evaluation environments continuously alongside training. We report raw undiscounted evaluation returns (sum of rewards) rather than normalized scores, and smooth curves by window averaging with window size 151. Each row corresponds to a different environment; curves show the median across 5 random seeds. All methods build on the PQN architecture and training pipeline of Gallici et al. (2024), following the Atari reward decomposition setup of Zhang et al. (2021), and differ only in the distributional objective used to train the critic.

## K. LLM Usage

We used an LLM-based assistant to support the preparation of this paper. In particular, it was employed to (i) rephrase draft paragraphs for clarity and suggest alternative framings of related work, (ii) format proofs, explore directions, and verify intermediate steps, (iii) assist in debugging code, (iv) suggest LaTeX equation formatting, and (v) help identify relevant theoretical results in preceding works. All core research contributions, including the development of theoretical results, algorithms, and experiments, were carried out by the authors.

