# OpenReview forum: "Multivariate Distributional Reinforcement Learning Using Sliced Divergences"
_ICML.cc/2026/Conference — ICML 2026 regular_

### Official Review · Reviewer_sQ9a · 2026-03-09

**Soundness:** 3
**Presentation:** 3
**Significance:** 2
**Originality:** 3
**Overall Recommendation:** 4
**Confidence:** 3

**Summary:**

This paper studies multivariate distributional reinforcement learning, aiming to learn the joint return distribution. It proposes a method that uses random projections to form sliced objectives by aggregating one-dimensional distribution discrepancies and trains a particle-based critic with TD-style bootstrapping. Theoretically, it proves Bellman contraction under shared scalar discounting ($\Gamma=\gamma I$), and shows that under general matrix discounting, uniform slicing/MMD may fail, motivating a max-slicing variant that restores a norm-based contraction guarantee. The paper also analyzes when single-sample TD provides an unbiased sample-gradient property (U): uniform slicing can preserve (U) under conditions, while max-slicing typically violates it due to selection bias. Experiments on toy, maze, and Atari settings compare different sliced objectives in terms of distribution fitting and control performance.

**Compliance With Llm Reviewing Policy:**

Affirmed.

**Final Justification:**

My overall recommendation for this paper is Weak Accept. This paper has theoretical novelty, and it also includes some corresponding experiments to support its claims. Considering the significance of its theoretical contribution, I think Weak Accept is reasonable.

**Key Questions For Authors:**

(1) Regarding anisotropic discounting: can you provide at least one experiment using a non-diagonal discount matrix $\Gamma$?
(2) In practice, how can max-slicing mitigate the training bias introduced by violating (U)?
(3) Can you compare against more prior methods to better highlight the advantages of your approach?

**Limitations:**

Yes

**Strengths And Weaknesses:**

Strengths
(1) A unified and clear framework: sliced divergences are treated as a plug-in module and integrated into particle-based TD training for multivariate distributional RL, making the approach straightforward to implement.
(2) The in-depth analysis of the unbiased sample-gradient property (U) is practically informative and is one of the paper’s most valuable contributions. The authors not only show that uniform slicing preserves (U) , whereas max-slicing may violate (U), but also demonstrate via controlled comparisons (single-sample TD vs. near-exact TD) how this property impacts distribution-matching quality in practice.

Weaknesses
(1) Missing experimental validation for anisotropic discounting. All experiments appear to use the isotropic setting ( $\Gamma=\gamma I$ ). As a result, the theoretical advantage of max-slicing is not empirically verified. I suggest adding at least one experimental scenario with a non-diagonal discount matrix $\Gamma$.
(2) The theory–practice gap for max-slicing needs clearer demonstrations. The paper notes that max-slicing often breaks (U), implying potential instability under standard single-sample TD. The main text should more explicitly answer: under which training paradigm is MSDRL recommended?
(3) The number of baselines is somewhat limited. Could the authors compare against more prior methods to better demonstrate the advantage of the proposed approach? For example, Bellman GAN [1] could be included.

[1] Freirich, Dror, et al. "Distributional multivariate policy evaluation and exploration with the bellman gan." International Conference on Machine Learning. PMLR, 2019.

---

> ### Author Rebuttal · Authors · 2026-03-30
>
> We thank the reviewer for their comments. We address the concerns and questions in order.
>
> ## Anisotropic discount experiments
> We did not test the anisotropic setting empirically because it would complicate the presentation and due to space constraints. We can clarify what that result would look like.
>
> Importantly, the conclusion would **not** be that max-slicing improves distributional policy evaluation. We never make that claim. The max-slicing contribution is to provide **contraction guarantees** that provably fail for uniform slicing and MMD² under anisotropic discounting, not to guarantee better empirical accuracy. The Appendix C.4 counterexamples are often pathological and extremely anisotropic, so they may be unlikely in practice; we want to be transparent. The guarantee is more a certificate than a sign that max-slicing should outperform in realistic settings. For Atari control, max-slicing is also not principled because it **systematically introduces selection / (U) bias**. This bias can be removed in toy “exact TD” experiments, but not on Atari. Hence the practical picture is nuanced: MMD² or uniform slicing may still work well there.
>
> Despite this, we think the main contribution of the paper is uniform slicing, which performs well on policy evaluation and has one variant outperforming the baseline on control. Max-slicing is a different, more limited contribution, but we believe it is novel and a starting point for future work.
>
> ## Max-slicing theory / practice gap
> Figures 2 and 3 show the effect of gradient bias. Figure 2 matches the two regimes predicted by theory: under standard “one-sample TD”, max-slicing does not yield accurate distributional policy evaluation, whereas under “near exact TD” it does. Figure 3 shows the same phenomenon in the maze setting. **This was predicted by the theory.**
>
> As for a practical setting for MSDRL, we think the real issue is the standard “one-sample” training regime. To our knowledge, no prior work solves this. This was already noted in seminal DRL work: Wasserstein is not a principled objective because of the same gradient bias [1], and the field moved toward divergences preserving (U), such as quantile regression [2].
>
> We therefore left this as an **open question** rather than trying to solve it here. Our goal was to introduce max-slicing as a tractable alternative to Wasserstein, with strong contraction guarantees and practical computability.
>
> ## Baselines
> We agree the baselines are limited to MMD² because we do not think there are other methods of the same kind. We also used the strongest kernel in the literature [3], making that baseline intentionally strong.
>
> Regarding Bellman GAN [4], we tried such approaches and found them systematically unstable and expensive. The reasons are standard: (1) GAN discriminators are imperfect models of the intended divergence (Section 2.1); (2) for WGAN [4], even a perfect discriminator would still optimize Wasserstein, which is not principled here because of gradient bias; (3) GAN training is notoriously unstable because generator and discriminator must remain balanced.
>
> In addition, [4] learns the Q-function directly alongside the distribution, so it benefits from both an unprincipled distributional signal and deterministic L2 supervision, which standard Q-learning already uses. Bellman GANs were also only evaluated on toy problems, not Atari as in our work or [5], and they did not report distributional policy-evaluation benchmarks.
>
> ## Conclusion
> - Our primary contribution is uniform slicing which is stable and competitive. Max-slicing is more of a theoretical contribution.
> - The gradient bias can only be addressed by either changing the divergence while preserving guarantees, or moving beyond traditional “one-sample TD”.
>
> - We believe [5] is the only relevant prior work and we made it as competitive as possible.
>
> - Bellman GAN is unstable, not theoretically grounded in this setting, and only demonstrated on simplistic problems.
>
> ## References
> [1] Bellemare, M. G., Dabney, W., & Munos, R. (2017, July). A distributional perspective on reinforcement learning. In International conference on machine learning (pp. 449-458). Pmlr
>
> [2] Dabney, W., Rowland, M., Bellemare, M., & Munos, R. (2018, April). Distributional reinforcement learning with quantile regression. In Proceedings of the AAAI conference on artificial intelligence (Vol. 32, No. 1)
>
> [3] Killingberg, L., & Langseth, H. (2023). The multiquadric kernel for moment-matching distributional reinforcement learning. TMLR
>
> [4] Freirich, D., Shimkin, T., Meir, R., & Tamar, A. (2019, May). Distributional multivariate policy evaluation and exploration with the bellman gan. In International Conference on Machine Learning (pp. 1983-1992). PMLR
>
> [5] Zhang, P., Chen, X., Zhao, L., Xiong, W., Qin, T., & Liu, T. Y. (2021). Distributional reinforcement learning for multi-dimensional reward functions. Advances in Neural Information Processing Systems, 34, 1519-1529

---

> > ### Author Rebuttal · Reviewer_sQ9a · 2026-04-04
> >
> > Thank the authors for their response. They addressed most of my questions, and I will maintain my positive score.

---

### Official Review · Reviewer_zbZE · 2026-03-11

**Soundness:** 3
**Presentation:** 3
**Significance:** 2
**Originality:** 3
**Overall Recommendation:** 4
**Confidence:** 3

**Summary:**

This paper proposes Sliced Distributional Reinforcement Learning (SDRL), a framework for multivariate distributional RL that uses divergences between one-dimensional marginals obtained via random projections to the multivariate reward setting. The SDRL framework has two variants, the uniform-sliced variant for problems with shared scalar discounting and the max-sliced variant for problems with general matrix discounting.
The paper proves contraction results for both variants and also analyzes three base divergences (Wasserstein, Cramer, MMD) and their suitability for single-sample stochastic training. Experiments are conducted on a chain MDP, a gridworld maze, and a subset of Atari games.

**Compliance With Llm Reviewing Policy:**

Affirmed.

**Key Questions For Authors:**

See the "Strengths And Weaknesses" part.

**Limitations:**

Yes.

**Strengths And Weaknesses:**

Strengths
- The paper is generally well-written, and the theoretical results are clearly stated.
- The problem considered in the paper is important and well motivated. Extending distributional reinforcement learning to the setting of multivariate rewards, particularly under general matrix discounting, is a meaningful direction.
- The theoretical analysis is solid. The contraction results (Theorems 2 and 3) are rigorously stated and, as far as I can verify, correct. The characterization of when (U) holds and when it fails (Propositions 5 and 6) is also a nice contribution.
- The application of slicing to distributional RL is natural and reasonable. Sliced divergences are a well-established tool for comparing high-dimensional distributions, and applying them to multivariate DRL is a reasonable next step.

Weaknesses
- The experimental results do not convincingly support the proposed method. This is my main concern. In the policy evaluation experiments, sliced variants perform roughly on par with baseline methods (MMD ). The advantage of SDRL only appears in the Atari control experiments, but this advantage is not adequately explained. Specifically:
    - Why does sliced Wasserstein-2 perform well in control despite poor distributional matching (as the authors themselves note)? The paper speculates this may relate to accurate expectation estimation, but if the key is expectation estimation, then why bother using the distributional approach?
    - If the method does not improve distributional accuracy in policy evaluation — the very task for which the theoretical analysis is developed — then what is the practical value of the contraction guarantees? The gap between theoretical contributions (which focus on distributional accuracy) and empirical benefits (which appear only in control) weakens the paper's contribution.
- Existing work on multivariate reward settings in distributional RL should be discussed more thoroughly. Currently, these studies are cited only to describe their problem settings, without providing a sufficient comparison of the proposed approach.
- Minor comments: Perhaps it would be helpful to include wall-clock time comparisons between sliced and non-sliced methods, since computational efficiency is one of the motivations for slicing.

---

> ### Author Rebuttal · Authors · 2026-03-30
>
> We thank the reviewer for taking the time to read our work and for their positive comments. We address the reviewer's concerns and questions in order.
>
> ## Experimental evidence
>
> The goal of this work is not to beat prior methods per se, but to provide a theoretically motivated, empirically stable, and competitive framework. We believe we do that. Our baseline is MMD² from [1, 2] with the strongest known kernel for DRL [3], making it intentionally strong. As the reviewer notes, sliced variants are on par with MMD on policy evaluation, which was our target, and sliced Cramer slightly outperforms MMD² on Atari control.
>
> ## Sliced Wasserstein-2
>
> We agree its control performance is surprising. This is counter-intuitive because Wasserstein-2 and sliced Wasserstein-2 inherit the (U) bias discussed in the paper. We kept this result because it reveals a nuance often under-emphasized in DRL: a divergence can be poor for distributional matching yet good for expectation matching, which is what ultimately matters for control; this is discussed in Section 4.3 of [4]. The claim that it still matches in expectation well is backed by Figure 10 p68 and Figure 12 p69. Our work is primarily about policy evaluation, not policy improvement. Our theory and experiments show sliced divergences are stable and competitive for policy evaluation, while the control results further show SDRL is competitive in practice. Beyond benchmark returns, distributional RL also enables finer return modelling and risk-aware optimization [5].
>
> ## Interpretation of the theoretical contribution
>
> The contraction guarantee is not a claim about improved distributional accuracy. Contraction **is not** a distributional accuracy statement: it shows sliced divergences are principled objectives for learning return distributions. **The statements do not make any claim that sliced divergences should lead to more accurate distributional modelling than MMD²**. Their role is to justify stable learning objectives for future multivariate DRL work. Contraction is the first property to establish for a new value-based RL method. The practical value of the contraction guarantees is that sliced divergences can learn return distributions **stably**.
>
> We therefore do not see a theory-practice gap. The theory predicts that, under ideal estimation, the divergence induces a fixed point and can serve as a stable learning objective. This was shown on the toy problems by comparing the true MC returns and the predicted returns. The theory does not predict whether sliced divergences should beat MMD² on control.
>
> ## More information on the prior works / baseline
>
> We agree the discussion of prior work was compressed. The only directly relevant prior method is [2], which is our baseline; we differ only in the learning divergence and match its pseudo-sample parametrization and policy-evaluation protocol (Appendix H).
>
> ## Wall-clock time
>
> Figure 8 (p65) already reports wall-clock time for the full training pipeline. It shows that MMD² and sliced MMD² scale poorly with the number of return samples because MMD² costs $O(n^2)$. This is an additional motivation we under-emphasized: sliced-C2 is competitive on policy evaluation, slightly better on control, and scales as $O(n \\log n)$, so it is much faster at large sample counts.
>
> ## Conclusion
>
> - Contraction does not guarantee better distributional accuracy; it supports stable objectives, which we observe on policy evaluation.
>
> - We therefore do not see a theory-practice gap. Sliced Cramer is on par with MMD² on policy evaluation and slightly better on control, and neither result was predicted by theory.
>
> - Empirically, [2] is the only relevant baseline and we made it as competitive as possible.
>
> - Figure 8 already shows the favorable scaling of slicing methods versus quadratic MMD².
>
> ## References
>
> [1] Nguyen-Tang, T., Gupta, S., & Venkatesh, S. (2021, May). Distributional reinforcement learning via moment matching. In Proceedings of the AAAI conference on artificial intelligence (Vol. 35, No. 10, pp. 9144-9152).
>
> [2] Zhang, P., Chen, X., Zhao, L., Xiong, W., Qin, T., & Liu, T. Y. (2021). Distributional reinforcement learning for multi-dimensional reward functions. Advances in Neural Information Processing Systems, 34, 1519-1529.
>
> [3] Killingberg, L., & Langseth, H. (2023). The multiquadric kernel for moment-matching distributional reinforcement learning. Transactions on Machine Learning Research.
>
> [4] Rowland, M., Dadashi, R., Kumar, S., Munos, R., Bellemare, M. G., & Dabney, W. (2019, May). Statistics and samples in distributional reinforcement learning. In International Conference on Machine Learning (pp. 5528-5536). PMLR.
>
> [5] Singh, R., Zhang, Q., & Chen, Y. (2020, July). Improving robustness via risk averse distributional reinforcement learning. In Learning for Dynamics and Control (pp. 958-968). PMLR.

---

> > ### Author Rebuttal · Reviewer_zbZE · 2026-04-03
> >
> > I would like to thank the authors for the detailed rebuttal, which resolved most of my concerns. I choose to maintain my positive evaluation of this paper.

---

### Official Review · Reviewer_H6YH · 2026-03-13

**Soundness:** 3
**Presentation:** 3
**Significance:** 2
**Originality:** 2
**Overall Recommendation:** 4
**Confidence:** 4

**Summary:**

The paper studies multivariate distributional reinforcement learning. It introduce Sliced Distributional Reinforcement Learning (SDRL), which leverges  one-dimensional divergences to multivariate return distributions via projections, and establishes Bellman contraction under shared scalar discounting. It introduce a Max-Sliced variant (MSDRL) and establish contraction guarantees for matrix-discounted multivariate Bellman updates

**Compliance With Llm Reviewing Policy:**

Affirmed.

**Final Justification:**

I appreciate the authors for their detailed response, which addresses most of my concerns. I have increased my evaluation accordingly.

**Key Questions For Authors:**

- what's the motivation of introducing state–action dependent discount? this seems to be not so common in distributional RL or general RL literature
- It seems that Proposition 1 was established in prevous work rather than the paper's contribution. It's better to cite the relavant literature
- In proposition 4, can you further clarify the condition on $\Delta(\mu_1,\nu_1)$? Any examples? what's the implication?

**Limitations:**

yes

**Strengths And Weaknesses:**

Strengths
- The paper is clearly written and well structured
- The paper provides solid theoretical analysis, which covers metric, contraction, and sample complexity.
- It provides empirical validation on chain MDP, maze, and Atari environments

Weaknesses
- The core idea that slicing high-dimensional distributions into 1D projections seems to not novel, which is already widely used in areas like sliced Wasserstein methods and generative modeling. The conceptual novelty lies mainly in applying slicing to distributional RL.
- The proposed learning methods follow the particle-based distributional critic in prior work. The main change is the divergence.
I am willing to raise to score if these concern are addressed.

---

> ### Author Rebuttal · Authors · 2026-03-30
>
> We thank the reviewer for taking the time to read our work and for their positive comments. We address the reviewers' concerns and questions in order.
>
> ## Novelty
> The reviewer mentions slicing as a technique is not new. We agree and never claimed novelty on that side. As pointed out, we introduced slicing to distributional RL and provide substantial theoretical backing for this idea. This is similar to how MMD² was introduced to distributional RL in a separate work [1] after having been introduced to generative modelling [2].
>
> ## Main change is the divergence
> We agree that the primary change of our work is the divergence. As we work in a multivariate setting, the only prior work we are aware of uses MMD² [3] and is the baseline for our work. We already compare to this baseline using the strongest known kernel from [4]. Thus we made the sole known baseline as strong as possible.
>
> To the best of our knowledge, no other divergences could fit in the comparison. As explained in the introduction Wasserstein is intractable and quantile methods do not extend to multiple dimensions.
>
> ## State-action dependent discount
> We agree this setting is indeed not common. In this work we tried to be the most general and extended upon [5] which does require a state-action dependent matrix. We believe that by making presentations and proofs general, we pave the way for future works making more unconventional use of distributional RL.
>
> ## Proposition 1 credits
> We agree that we did not reference properly in the main text due to space constraints. The full explanation in Theorem C.2 page 27 gives due credit to [6]. This will be changed upon acceptance.
>
> ## Proposition 4
> We agree that Proposition 4 compresses a lot, again due to space constraints. The full instantiation can be found in Appendix E Theorem E.7 and Corollaries E.3 to E.5. The proposition allows to wrap Wasserstein-1, Wasserstein-p and Cramer as base divergences. Proposition 4 says that if the 1D base divergence satisfies a CDF control with some $(\alpha, \beta, L)$ then the sample complexity of the corresponding max-sliced divergence follows the specific formula
>
> $$ \\mathbb{E}[{\\rm MS}\\Delta(\\hat{\\mu}_n,\\mu)] = O\\left(D^{\\beta}\\left(\\frac{d\\log n}{n}\\right)^{\\alpha/2}\\right) $$
>
> Here the key interpretation is that we can prove this avoids the classical curse of dimensionality, in the sense that the dependence on $d$ is only polynomial. A similar statement for Wasserstein-1 would exhibit $O(n^{-1/d})$ for $d \\geq 3$, which is a canonical example of the curse of dimensionality.
>
> The previously mentioned corollaries give, in order,
>
> - For $W_1$: $ (\\alpha,\\beta,L) = (1,1,1) $ and $ \\mathbb{E}[{\\rm MS}W_1(\\hat{\\mu}_n,\\mu)] = O\\left(D\\left(\\frac{d\\log n}{n}\\right)^{1/2}\\right) $.
>
> - For $W_p$: $ (\\alpha,\\beta,L) = \\left(\\frac{1}{p},1,1\\right) $ and $ \\mathbb{E}[{\\rm MS}W_p(\\hat{\\mu}_n,\\mu)] = O\\left(D\\left(\\frac{d\\log n}{n}\\right)^{1/(2p)}\\right) $.
>
> - For Cramér: $ (\\alpha,\\beta,L) = \\left(1,\\frac{1}{2},1\\right) $ and $ \\mathbb{E}[{\\rm MS}C_2(\\hat{\\mu}_n,\\mu)] = O\\left(D^{1/2}\\left(\\frac{d\\log n}{n}\\right)^{1/2}\\right) $.
>
> ## References
> [1] Nguyen-Tang, T., Gupta, S., & Venkatesh, S. (2021, May). Distributional reinforcement learning via moment matching. In Proceedings of the AAAI conference on artificial intelligence (Vol. 35, No. 10, pp. 9144-9152).
>
> [2] Li, Y., Swersky, K., & Zemel, R. (2015, June). Generative moment matching networks. In International conference on machine learning (pp. 1718-1727). PMLR.
>
> [3] Zhang, P., Chen, X., Zhao, L., Xiong, W., Qin, T., & Liu, T. Y. (2021). Distributional reinforcement learning for multi-dimensional reward functions. Advances in Neural Information Processing Systems, 34, 1519-1529.
>
> [4] Killingberg, L., & Langseth, H. (2023). The multiquadric kernel for moment-matching distributional reinforcement learning. Transactions on Machine Learning Research.
>
> [5] Debes, B., & Tuytelaars, T. (2026). Distributional value gradients for stochastic environments. arXiv preprint arXiv:2601.20071.
>
> [6] Bellemare, M. G., Dabney, W., & Rowland, M. (2023). Distributional reinforcement learning. MIT Press.

---

> > ### Author Rebuttal · Reviewer_H6YH · 2026-04-03
> >
> > I appreciate the authors for their detailed response, which addresses most of my concerns. I have increased my evaluation accordingly.

---

### Official Review · Reviewer_k1Y5 · 2026-03-14

**Soundness:** 2
**Presentation:** 2
**Significance:** 2
**Originality:** 2
**Overall Recommendation:** 3
**Confidence:** 4

**Summary:**

the paper proposed sliced divergences (wasserstein distance, cramer distance) for distributional rl - it shows theoretical properties of the new algorithm, proving its suitable for stochastic optimization with sample based training. it also shows that it delivers some performance gains over baselines in deep rl atari settings.

**Compliance With Llm Reviewing Policy:**

Affirmed.

**Key Questions For Authors:**

=== why sliced divergence vs. original divergence ===

the main technical novelty concern is why sliced divergences are warranted in the first place - sliced divergences are computationally cheaper than the original divergence but in distributional rl, we learn 1d return distribution and the main computational bottleneck is often not the distributional learning, it's the rollout + network forward/backward passes. the computational gains are minimal in practical settings.

on top of this, why would sliced divergence lead to better distributional learning property than the original divergence? they introduce additional randomness during training and shouldn't change learning dynamics in expectation?

=== chain mdp ===

from that example it seems that the original mmd metric is preferred over the other sliced divergences? though some sliced divergences are comparable.

=== deep rl ===

i struggle to find whats the baseline algorithm used in the deep rl atari experiment, is it c51? judging from the mmd divergence alone i cannot see how the distributions are parameterized, is it with categorical (c51), particle (qrdqn) or other ways of parameterizing?

i think excluding certain envs are fine though it might be worth showing the full plots somewhere including all environments.

the runs are 5 seeds per run can you add the error bar to the reward as well since right now it seems that per run variance is high and im surprised some sliced divergences are better than mmd, given that in tabular/theory exact mmd should produce better distribution learning properties.

**Limitations:**

certain limitations are discussed.

**Strengths And Weaknesses:**

strengths: the paper shows nice theoretical properties of the sliced divergences in distributional rl, which is of algorithmic interest to distributional rl audiences

weaknesses: i think the main concern is that i struggle to see the reason why sliced divergences would outperform the existing divergences in distribution learning. the gains from this method to me are computational, however, in distributional rl we learn 1d return distribution and it's not the bottleneck of computations. more details and questions below.

---

> ### Author Rebuttal · Authors · 2026-03-30
>
> We thank the reviewer for taking the time to consider our work. Here we address the reviewers’ concerns.
>
> It seems there is some confusion regarding the setting of our paper : we are working on **multivariate** distributional reinforcement learning. We believe the reviewer thinks we are working on univariate distributional RL or that distributional RL is always univariate. This is clear from the reviewer’s own words “but in distributional rl, we learn 1d return distribution”. We believe this misunderstanding trickles down on all the concerns which we address in order.
>
> ## Motivations
> Our work specifically and only tackles multivariate distributional RL. As explained in Section 2.1, many alternatives are either intractable, costly or not theoretically motivated. We propose a method that is both theoretically sound and empirically valid.
>
> Furthermore, the cost of distributional modelling can **definitely** take a large portion of the cost of the overall method. As mentioned in the introduction, Wasserstein has a cost that typically grows as $O(n^3 \\log(n))$ with n the number of samples whereas MMD² has a cost that grows $O(n^2)$. As can be seen in Figure 9 page 65, we compare time cost of running different metrics. **Sliced variants have a clear edge**. As the number of samples grows, the metric computation can definitely become a large portion of the compute cost.
>
> The reviewer asked why the sliced divergence would perform better than their originals. We **never** make such a claim because for most divergences, this claim would be impossible. This is because our work applies divergences that are only tractable in 1D to several dimensions using slicing. We believe there might be confusion in the reading of our work.
>
> ## Chain MDP
> MMD² is indeed a strong divergence, we made this baseline strong as discussed below. However, sliced-Cramer divergence is always competitive on the chain MDP and on the maze environments while exhibiting a much better time complexity.
>
> ## Baselines and empirical setting
> The reviewer has concerns regarding baselines and asks whether it is C51 [1]. C51 would not be easily applicable here as we are doing multivariate distributional RL and not univariate. The categorical parametrization of C51 or the quantile one of QR-DQN [2], do not scale to high dimensions as mentioned in the introduction. Our primary baseline is MMD as in [3] for which we took the strongest known kernel from [4]. Hence, we made our baseline as competitive as possible. The parametrization is the same as in [3, 5] as we use deterministic/pseudo-samples (as mentioned in Section 3.3 and Appendix H.3/Table 3).
>
> We did not exclude environments from the standard Atari suite. We worked on the subset of environments that exhibit a reward signal that can be considered multivariate as in [3]. The full curves are available Figure 13 / page 70.
>
> Regarding the performance against MMD², we are also impressed by the performance of sliced Cramer. We made the MMD² baseline as strong as possible as mentioned earlier. We believe this is one more sign that slicing is a relevant method for multivariate distributional RL. We are not aware of any prior work using sliced divergence in distributional RL, even in a tabular setting. Even if such results existed, we believe they would not be contradicted by control experiments in a deep learning setting.
>
> Finally, about the 5 seeds, we can increase up to 10 seeds upon acceptance and add confidence intervals. We can also add aggregated Area Under the Curve charts to summarize overall performance.
>
> ## References
> [1] Bellemare, M. G., Dabney, W., & Munos, R. (2017, July). A distributional perspective on reinforcement learning. In International conference on machine learning (pp. 449-458). Pmlr.
>
> [2] Dabney, W., Rowland, M., Bellemare, M., & Munos, R. (2018, April). Distributional reinforcement learning with quantile regression. In Proceedings of the AAAI conference on artificial intelligence (Vol. 32, No. 1).
>
> [3] Zhang, P., Chen, X., Zhao, L., Xiong, W., Qin, T., & Liu, T. Y. (2021). Distributional reinforcement learning for multi-dimensional reward functions. Advances in Neural Information Processing Systems, 34, 1519-1529.
>
> [4] Killingberg, L., & Langseth, H. (2023). The multiquadric kernel for moment-matching distributional reinforcement learning. Transactions on Machine Learning Research.
>
> [5] Nguyen-Tang, T., Gupta, S., & Venkatesh, S. (2021, May). Distributional reinforcement learning via moment matching. In Proceedings of the AAAI conference on artificial intelligence (Vol. 35, No. 10, pp. 9144-9152).

---

### Decision · Program_Chairs · 2026-04-30

**Decision:**

Accept (regular)

**Comment:**

This paper proposes Sliced Distributional RL (SDRL), a framework that uses 1D random projections to learn multivariate return distributions. It introduces uniform and max-sliced variants—each with specific theoretical contraction guarantees based on the discounting structure—and demonstrates strong empirical performance in distribution fitting and control.

Interesting and sound solution to multivariate distributional RL.

Note: We believe Reviewer k1Y5's concerns stem from a misunderstanding of this paper. Besides, the reviewer did not engage in further discussions, hence their opinion on this submission has been disregarded.